# Models optimized for real-world tasks reveal the task-dependent necessity of precise temporal coding in hearing

Mark R. Saddler [1,2,3] ✉ & Josh H. McDermott [1,2,3,4] ✉

Neurons encode information in the timing of their spikes in addition to their firing rates. Spike timing is particularly precise in the auditory nerve, where action potentials phase lock to sound with sub-millisecond precision, but its behavioral relevance remains uncertain. We optimized machine learning models to perform real-world hearing tasks with simulated cochlear input, assessing the precision of auditory nerve spike timing needed to reproduce human behavior. Models with high-fidelity phase locking exhibited more human-like sound localization and speech perception than models without, consistent with an essential role in human hearing. However, the temporal precision needed to reproduce human-like behavior varied across tasks, as did the precision that benefited real-world task performance. These effects suggest that perceptual domains incorporate phase locking to different extents depending on the demands of real-world hearing. The results illustrate how optimizing models for realistic tasks can clarify the role of candidate neural codes in perception.

Sensory systems encode information about the environment in the spiking activity of neurons. Decades of experiments have clarified how stimulus properties are represented at different stages of neural processing, but less is known about how this information gives rise to complex human behavior.

In perceptual science, ideal observer models have long been used to analyze which features of a neural code contribute to behavior[1–3]. An ideal observer is the statistically optimal solution to a perceptual task given the information available at some stage of neural processing[4]. Since evolutionary pressures drive biological perceptual systems in the direction of optimal performance for tasks that are important in the natural environment, comparisons of an organism's behavior to that of optimal task solutions under candidate biological constraints (e.g., a type of neural code) can reveal whether the organism is also operating under those constraints. This approach has produced rigorous computational accounts of some aspects of vision[5–9] and hearing[2,10–14]. However, ideal observers are limited to tasks for which provably optimal solutions can be derived (i.e., for which probability

distributions of the generative parameters can be specified), precluding most real-world behaviors. Because real-world tasks are the ones that biological systems are likely to have been optimized for, ideal observers have had limited applicability in domains where they might otherwise be most useful.

Here, we propose machine learning as an alternative approach to link neural coding to behavior. Contemporary machine learning models are expressive functions that can be optimized to perform real-world tasks with natural stimuli, "learning" solutions from empirical distributions of stimuli and labels rather than mathematical descriptions of a task. In contrast to analytically derived optimal solutions, the solutions found via an optimization process are not guaranteed to be optimal (for instance, the optimization procedure could get stuck in local optima, and/or the model class being optimized could be sub-optimal for the problem). However, optimization drives a model towards better performance, such that the resulting model may nonetheless reveal the characteristics of a system optimized for a problem under particular constraints. In this way, machine learning

[1]Department of Brain and Cognitive Sciences, MIT, Cambridge, MA, USA. [2]McGovern Institute for Brain Research, MIT, Cambridge, MA, USA. [3]Center for Brains, Minds, and Machines, MIT, Cambridge, MA, USA. [4]Program in Speech and Hearing Biosciences and Technology, Harvard, Cambridge, MA, USA. ✉e-mail: msaddler@mit.edu; jhm@mit.edu

offers an alternative to the traditional ideal observer approach for real-world perception problems that can only be specified empirically. Previous work has shown that human-like behavior can emerge in deep artificial neural networks optimized for natural tasks[15–20], consistent with the idea that humans are shaped by optimization for such tasks. We propose that comparing the behavior of models optimized to perform tasks using different neural representations can reveal the aspects of neural coding necessary for human behavior, and the ecological pressures that drive their use.

Here, we apply this general approach to the problem of temporal coding. Neurons transmit information in the precise timing of their spikes[21] in addition to their time-averaged firing rates. Temporal coding has been identified across multiple sensory modalities[22–26], but spike timing is arguably most precise in the auditory nerve, where action potentials align to the temporal structure of sound with sub-millisecond precision. This precise spike timing plausibly helps to encode the "fine structure" of a sound waveform (i.e., individual pressure oscillations; Fig. 1a). Mammalian auditory nerve fibers phase lock to sound frequencies as high as 3 to 5 kHz[27–29], such that the auditory system in principle has access to this information from the outset. However, whether this information is actually used by the brain remains among the most debated issues in hearing science[30–32].

The one aspect of hearing widely believed to rely on high-fidelity phase locking is sound localization, which depends in part on microsecond-level timing differences between the two ears. Neural circuits for extracting these timing differences from stimulus fine structure are found in many non-human animals. However, for other aspects of hearing there is no consensus.

The issue has remained unresolved for several reasons. First, there is no conclusive evidence for monaural circuits that could extract the information in spike timing. The precision of temporal coding degrades with each synapse along the ascending auditory pathway[33,34], such that physiological mechanisms for extracting information from high-frequency phase locking are likely to be situated early. Yet despite considerable effort, no such mechanisms for extracting phase locking monaurally have been discovered[31,32]. Second, causal manipulations of phase locking are impractical due to the difficulty of manipulating the nerve in vivo, and because non-human animals do not exhibit many human auditory behaviors. Third, attempts to address the issue psychophysically have been inconclusive[32,35], despite widespread proposals that phase locking is critical for hearing in noise[36–40]. The issue is important to resolve for the design of cochlear implants, which at present generally do not reproduce the phase locking seen in the normal ear, and for understanding how different forms of hearing loss (some of which are argued to affect the fidelity of temporal coding[41]) affect behavior.

In addition to not knowing whether and when temporal coding is used by the auditory system, it has also remained unclear why it would or would not be used. Even for sound localization, this remains unsettled, as the upper frequency limit of interaural time difference judgments is lower than the presumptive limit of phase locking in the nerve[42–44]. Although there are physiological correlates of this limit (cells that provide input to brainstem binaural circuits exhibit degraded synchrony above 1 kHz[45]), it is not well understood in normative terms. This question has remained difficult to answer because until recently it was infeasible to model real-world auditory behavior, leaving it unclear to what extent precise timing in the input was computationally important for audition. If phase locking is not needed to obtain good performance in natural auditory tasks, and/or if the circuits to extract it monaurally are prohibitively expensive to implement, it might be discarded by the downstream auditory system.

Classical ideal observers have been applied to aspects of this problem, but were restricted to simple tasks with artificial stimuli (e.g., discrimination of single frequencies[2,12]). The resulting models generally overestimate human performance[2], plausibly because human perceptual systems are not optimized for such simple tasks and artificial stimuli. Our approach was to instead investigate the issue using models optimized for ecological tasks, by training machine learning systems to perform such tasks using simulated auditory nerve representations as input. To ask whether the information encoded via peripheral spike timing is necessary to account for behavior, we separately optimized models with altered nerve phase locking and compared the models' behavior to that of human listeners (Fig. 1b). The results provide new evidence for the importance of high-fidelity temporal coding in perception, and provide a way to understand why it is used, by showing where it is critical for real-world task performance.

## Results

### Auditory nerve model stage

We hard-coded the model input representation to approximate the information the ear sends to the brain. We used a phenomenological model[46] of the auditory periphery to simulate instantaneous firing rate responses of a population of auditory nerve fibers whose frequency tuning and sensitivity was intended to match that of the human ear. We simulated the 3 canonical auditory nerve fiber types found in mammals, which have different spiking thresholds and spontaneous activity[47]. High-spontaneous-rate fibers have low thresholds but narrow dynamic ranges, such that their firing rates saturate at conversational sound levels. Medium- and low-spontaneous-rate fibers have higher thresholds and broader dynamic ranges but are less numerous in the ear. The resulting frequency-by-time-by-fiber-type array of instantaneous firing rates was then converted to an array of sampled spike counts, representing the population response of 32000 individual auditory nerve fibers per ear (60% high-spontaneous-rate, 25% medium-spontaneous-rate, and 15% low-spontaneous-rate), which served as the input representation to the networks. To our knowledge, our models are the first to perform naturalistic tasks using a near-realistic representation of the information from a sensory receptor organ.

### Temporal coding manipulation

The fidelity of temporal coding in the mammalian ear is limited by the capacitance and ion channel properties of the hair cell membrane[29] as well as the hair-cell-to-nerve-fiber synapse[48], both of which act as low-pass filters. The upper limit of phase locking was altered in silico by changing the cutoff frequency of the low-pass filter governing the inner hair cell potential in the peripheral model. We optimized machine models with different cutoffs to ask whether phase locking was necessary to obtain human-like behavior.

In one training condition, this low-pass cutoff was set to a default value of 3000 Hz, which produces phase locking similar to that seen in electrophysiological recordings from non-human animals (Fig. 1c). This upper limit is presumed to be shared by humans[49–52] but is not directly measurable. To investigate the behavioral relevance of temporal coding, we also trained models with each of three lower cutoff values: 1000 Hz, 320 Hz, and 50 Hz. Lowering this cutoff degrades the fidelity of temporal coding, progressively blurring the auditory nerve representation along the time-axis (Fig. 1d and e). The lowest cutoff eliminates essentially all phase locking to temporal fine structure in natural sounds (as well as to envelope modulations above the cutoff). However, as expected, the manipulation had very little effect on the pattern of firing rates across the cochlear frequency axis (Fig. 1f): nerve fibers with low phase locking limits encode high frequency sounds in their firing rates, just not with precise spike timing. We separately optimized neural networks operating on auditory nerve representations with these four different cutoff frequencies. We note that the cutoff determines the frequency at which phase locking precision rolls off, not the upper limit of all detectable phase locking, which could be slightly higher. For simplicity, we refer to different models by their cutoff frequencies (e.g., the 3000 Hz phase locking model).

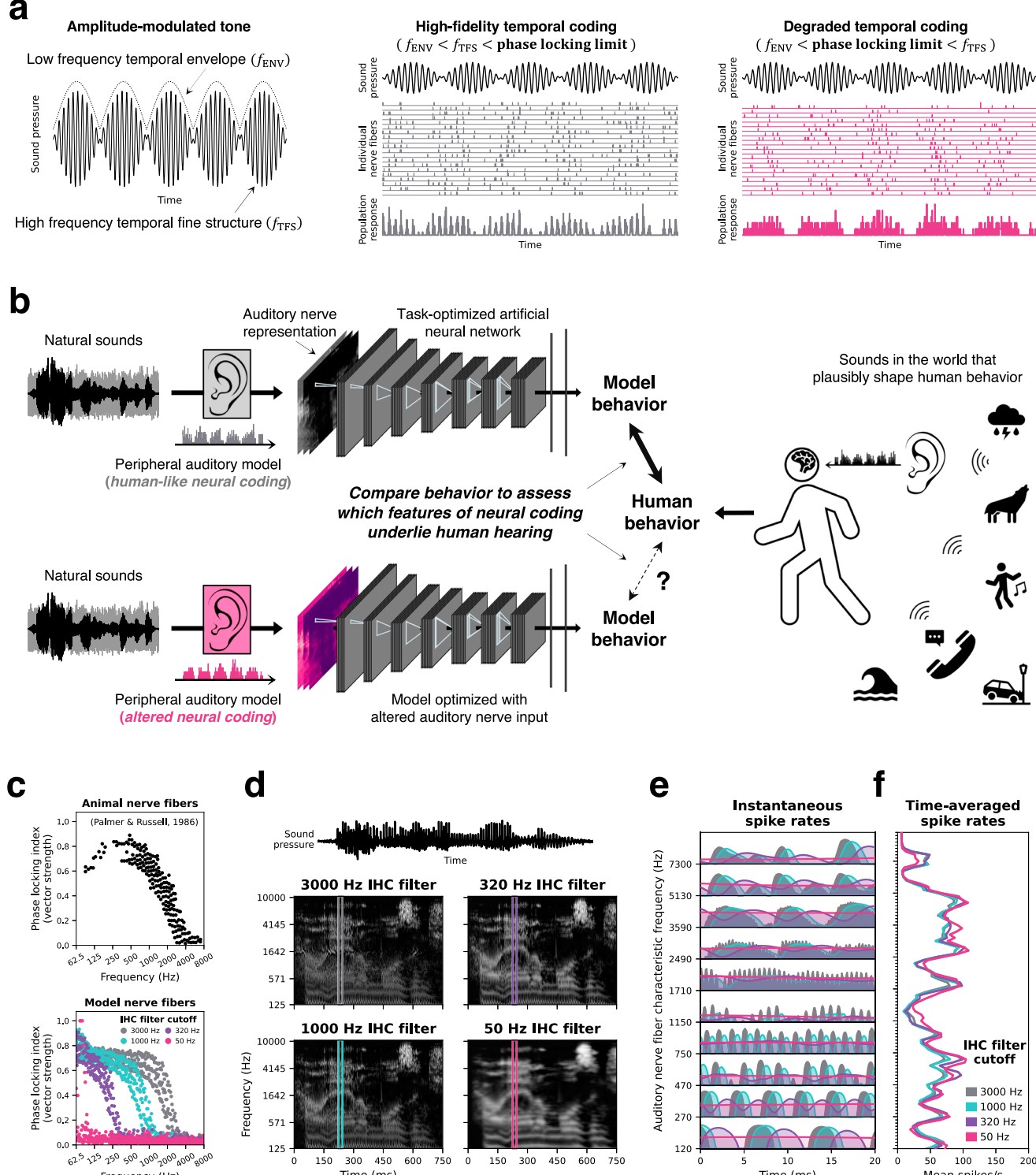

**Artificial neural network model stages and training**

The neural network portion of each model consisted of a feedforward series of stages instantiating linear convolution, nonlinear rectification, normalization, and pooling. The parameters of these model stages were optimized to perform auditory tasks via supervised machine learning. Each task was operationalized as a classification problem with a single ground-truth label per stimulus.

The performance of a neural network depends on both the weights (that are optimized via gradient descent for training task performance) and the hyperparameters that define the network architecture (e.g., the number of layers and the size and shape of

convolutional filter kernels). To ensure that these hyperparameters were also optimized for the tasks, we used the top 10 best-performing network architectures previously identified in large-scale random architecture searches conducted for each task (Supplementary Tables 1–2)[18,19,53]. Results for each task and cochlear model configuration are presented as the average of these 10 network architectures, allowing us to provide uncertainty estimates and marginalize across the idiosyncrasies of any single network architecture.

Our approach relied on optimizing models for "natural" tasks, on the grounds that these are likely to have shaped the nervous system's strategies. As such, we define natural tasks to be those that humans

**Fig. 1 | Overview of approach. a** Sound waveforms carry information in their amplitude envelope as well as their individual pressure oscillations (the "temporal fine structure" or "TFS"). The envelope and fine structure are encoded with phase-locked spike timing in the auditory nerve. As temporal coding is degraded, auditory nerve spikes no longer phase lock to the fine structure, encoding only the slower envelope fluctuations. **b** Schematic of the approach. Human auditory behavior is shaped by the ears and the acoustic environment. Models optimized to perform naturalistic tasks might reproduce human-like behavior if optimized for the auditory nerve information used by the human auditory system. **c** Top: The strength of phase locking as a function of frequency, measured in the auditory nerve fibers of guinea pigs. Data are re-plotted from Ref. 29. Bottom: The roll-off in phase locking strength is determined by the low-pass filter characteristics of the inner hair cell.

Manipulating the hair cell low-pass filter cutoff in model auditory nerve fibers changes the upper frequency limit of phase locking. The 3000 Hz cutoff best approximates the guinea pig data and is commonly used to model the human auditory nerve. **d** Simulated auditory nerve representations of the same speech waveform with four different configurations of the auditory nerve model. Configurations differed in the inner hair cell low-pass filter cutoff. **e** Instantaneous firing rates from example auditory nerve fibers illustrate the degradation of precise spike timing as the phase locking limit is lowered. Note the rapid oscillations in firing that are present for higher phase locking limits, but absent when the limit is lowered. **f** Time-averaged firing rates across the 25 ms window depicted in (**e**) illustrate that lowering the phase locking limit does not disrupt "place" cues in the overall pattern of excitation across the cochlear frequency axis.

perform in their daily lives and that have likely been important for survival: recognizing and localizing everyday sounds in everyday conditions. We contrast such tasks with those often used in laboratory experiments, where both the behavioral judgment and stimuli can be artificial (e.g. discriminating synthetic tones or noise signals). Accordingly, training stimuli were compiled from large-scale corpora of natural sounds (speech and recordings of auditory scenes) and were meant to approximate the "auditory diet" that likely shaped biological hearing systems over the course of evolution and development.

Models were optimized to perform three different auditory tasks: sound localization, voice recognition, and word recognition. For each task, we separately trained models with four different auditory nerve phase locking limits and then compared their behavior to that of humans. The models featured here build on previous models of human sound localization[19] and word recognition[16,54] but were improved in several respects to enable a strong test of the importance of temporal coding. Specifically, they operated on more realistic input representations (incorporating spikes and multiple types of auditory nerve fibers), were trained on more realistic datasets, and were evaluated with an expanded set of psychoacoustic experiments. We emphasize that the models were not fit to match human data and were optimized only for task performance. Any similarity to human behavior is thus a consequence of optimization for the task given the constraints of the simulated auditory nerve input and model architecture.

### Logic of approach and aggregate results

We begin by outlining the logic of the approach along with a summary of the results that illustrates how the overall results relate to this logic and the conclusions that follow from it. We then present the results of individual experiments in each of the three task domains, which illustrate the specific effects that underlie the overall results.

### Effect of temporal coding on naturalistic task performance

For each of the three tasks, we first evaluated models on naturalistic stimuli in noise, asking whether phase locking is necessary for good performance. Because lowering the phase locking limit removes information from the model's input, there are two main qualitative possibilities. Performance could worsen for phase locking limits below a critical value, which would provide evidence that fine-grained temporal information from phase locking up to that value is beneficial for the task, and thus might have driven its role in perception. Alternatively, performance could remain similar across phase locking limits. This result would indicate that fine-grained temporal information is not needed for the task in question.

It was a priori likely that high-fidelity phase locking would matter to some extent for sound localization, where microsecond-level timing differences between the two ears can plausibly only be conveyed via spike timing, and where corresponding neural circuitry has been documented. However, it was unclear whether the benefit of phase locking would cap out below the presumptive upper limit of the

auditory nerve. It was also unclear what to expect for word and voice recognition.

Figure 2 shows the effect of the phase locking cutoff on overall performance for each of the three tasks in noise (solid lines; left $y$-axes). As expected, sound localization (Fig. 2a) was worse for lower cutoffs, with mean absolute localization error increasing by 6.4° as the cutoff was lowered from 3000 to 50 Hz ($p < 0.001$, evaluated by bootstrapping across 10 neural network architectures). However, the effect was driven by cutoffs below 1000 Hz. Voice recognition (Fig. 2b) showed a comparably large effect of phase locking, with accuracy dropping by 10.5% between the 3000 and 50 Hz conditions ($p < 0.001$). By contrast, the effect on word recognition (Fig. 2c) was modest, with accuracy dropping only 2.1% across conditions ($p < 0.001$). These results indicate that high-fidelity temporal coding aids localization and voice recognition in natural conditions but in absolute terms is less critical for word recognition.

### Effect of temporal coding on human-model behavioral similarity

For each task we then simulated a battery of psychoacoustic experiments measuring the effect of different cues on perception (described in detail in subsequent sections). We asked whether phase locking was necessary for a model to exhibit human-like behavior as assessed via the pattern of performance across the battery of experiments. The most diagnostic result would be that one of the phase locking limits produces more human-like behavior than the others. Such a result would indicate that phase locking up to that limit contributes to that behavior (and thus must be extracted by the auditory system). The maximally human-similar model could in principle occur for a low phase locking limit, if high-frequency phase locking is discarded by the auditory system for that task.

Figure 2 also shows human-model behavioral similarity in each of the three task domains for each phase locking cutoff (dotted lines; right $y$-axes). The results implicate phase locking in all three types of behavior, but to different extents. For sound localization the 1000 Hz cutoff produces most human-like behavior (as evaluated by the correlation between human and model results; see Supplementary Fig. 1 for comparable results using root-mean-squared error as the measure of similarity). This result implicates phase locking up to but not above 1000 Hz in human sound localization. By contrast, for both voice and word recognition, the three highest cutoffs produced comparably human-like behavior and only the 50 Hz cutoff produced an appreciably worse match to humans. This result provides evidence that phase locking above 50 Hz is used in human perception in these domains, with no evidence that phase locking above 320 Hz is used.

Insight into why phase locking is used when it is may be obtained by comparing the dotted and solid lines in Fig. 2. For sound localization (Fig. 2a), the finding that localization performance does not improve above 1000 Hz (Fig. 2a, solid line) provides a normative explanation for why phase locking above 1000 Hz does not appear to influence human perception (Fig. 2a, dotted line). If phase-locked spike-timing information above 1000 Hz does not aid localization in naturalistic

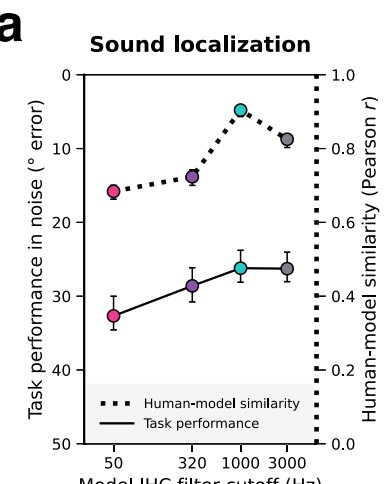
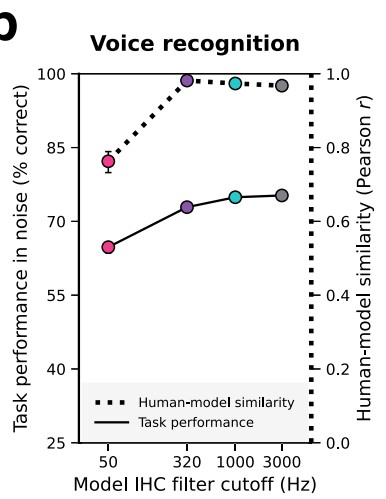
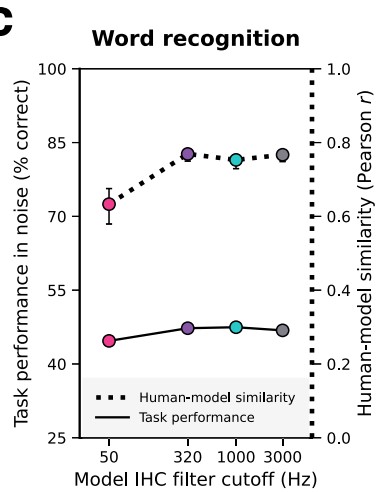

**Fig. 2 | Models with access to phase-locked spike timing have better and more human-like hearing.** Each panel corresponds to a different task and summarizes the effect of auditory nerve phase locking limit on naturalistic model task performance and overall human-model behavioral similarity. Naturalistic task performance is quantified as a single number averaged across noise conditions shown in later figures (left y-axes; solid lines). Overall human-model behavioral similarity is quantified as the Pearson correlation between analogous human and model data points, averaged across all experiments for each model task (right y-axes; dotted lines). Individual experiments are described in subsequent results sections and figures. Error bars indicate 95% confidence intervals of the mean (bootstrapped across 10 network architectures for each model). **a** Sound localization. The left y-axis plots mean absolute error for the sound localization model and is inverted so that better model performance corresponds to higher positions on the y-axis. **b** Voice recognition. Here and in (**c**) the left y-axes plot percent correct for the model when tested on speech in noise. **c** Word recognition. Source data are provided as a Source Data file.

conditions, then there would be little evolutionary pressure to extract it. Other species with smaller heads might require higher-fidelity temporal coding of time differences for good localization performance; see Discussion.

A similar correspondence is evident for voice recognition: recognition performance improves as the cutoff is raised from 50 – 320 Hz (Fig. 2b, solid line: improvement of 8.13%), but not much beyond that (e.g., from 320 Hz – 1000 Hz: improvement of 2.02%), roughly mirroring the effect of the phase locking cutoff on human-model similarity (Fig. 2b, dotted line). These results are consistent with the idea that phase locking is being used by the brain to the extent that it is advantageous within a domain for task performance.

The results for word recognition (Fig. 2c) are qualitatively similar: an improvement in performance was again evident from 50 – 320 Hz (improvement of 2.58%), but not beyond (no significant change from 320 Hz to 3000 Hz), and this mirrored the effect on human-model similarity. However, it is less clear that the modest improvement in performance would be enough to drive the incorporation of phase locking into the perceptual strategy (see below).

In the following sections, we consider each task in turn, showing the effect of phase locking on performance at different noise levels, and on a large set of psychophysical assays.

**Model optimization – sound localization**
To assess sound localization behavior, models were tasked with reporting the location of target sound sources in naturalistic auditory scenes rendered as binaural audio using a virtual acoustic room and head simulator (Fig. 3a). Each training scene consisted of a target source rendered at a single location in a room in the presence of spatially diffuse texture-like background noise. Background noises were selected to be more temporally homogeneous than the targets (e.g., the sound of running water rather than a single splash; see Methods) to ensure the task was well-defined. The model classified the azimuth and elevation of the target source relative to the simulated listener's head. The model operated on auditory nerve responses from the simulated listener's left and right ears, and thus had access to the same monaural and binaural cues as a human listener in the same

scene (Fig. 3b). Models optimized for this task with access to high-fidelity temporal coding in the peripheral representation have previously been shown to replicate characteristics of human sound localization[19], including the frequency-dependent use of interaural time and level differences for azimuth judgments[55] and the use of ear-specific spectral cues for elevation judgments[56,57]. However, it was unclear what would happen if models were optimized without high-fidelity temporal coding. Specifically, it was unclear how impaired temporal coding would affect the use of different localization cues, and whether the upper limit of time difference encoding evident in humans[42–44] could be explained by what is needed for natural task performance.

**Degraded temporal coding impairs sound localization**
We compared human and model sound localization accuracy for a set of 460 natural sounds presented in different levels of background noise (Fig. 3c). Humans were asked to report which of 95 loudspeakers (spanning −90° to 90° azimuth and 0° to 40° elevation) produced the target sound. This task was intended to tap into some of the challenges (background noise, many possible locations) that accompany localization in real-world settings. On each trial threshold-equalizing noise[58] was played diffusely from 9 other randomly selected loudspeaker locations. Models performed the same task in a virtual rendering of the loudspeaker array room. Overall task performance was quantified with mean absolute localization error as a function of SNR. Although human listeners outperformed all models at the lowest SNRs (plausibly because the models occasionally report the location of the noise rather than the target), models with access to high-frequency phase locking produced the best match to human behavior (Fig. 3d). Models with 3000 and 1000 Hz phase locking limits exhibited near-human-level robustness to noise, while models with degraded temporal coding made progressively larger localization errors as the phase locking limit was lowered. Degraded temporal coding impaired localization performance in both azimuth and elevation, though the effect was larger for azimuth (Fig. 3d, middle vs. right). These results confirm that precise spike timing is important for localizing natural sounds, particularly in noisy environments.

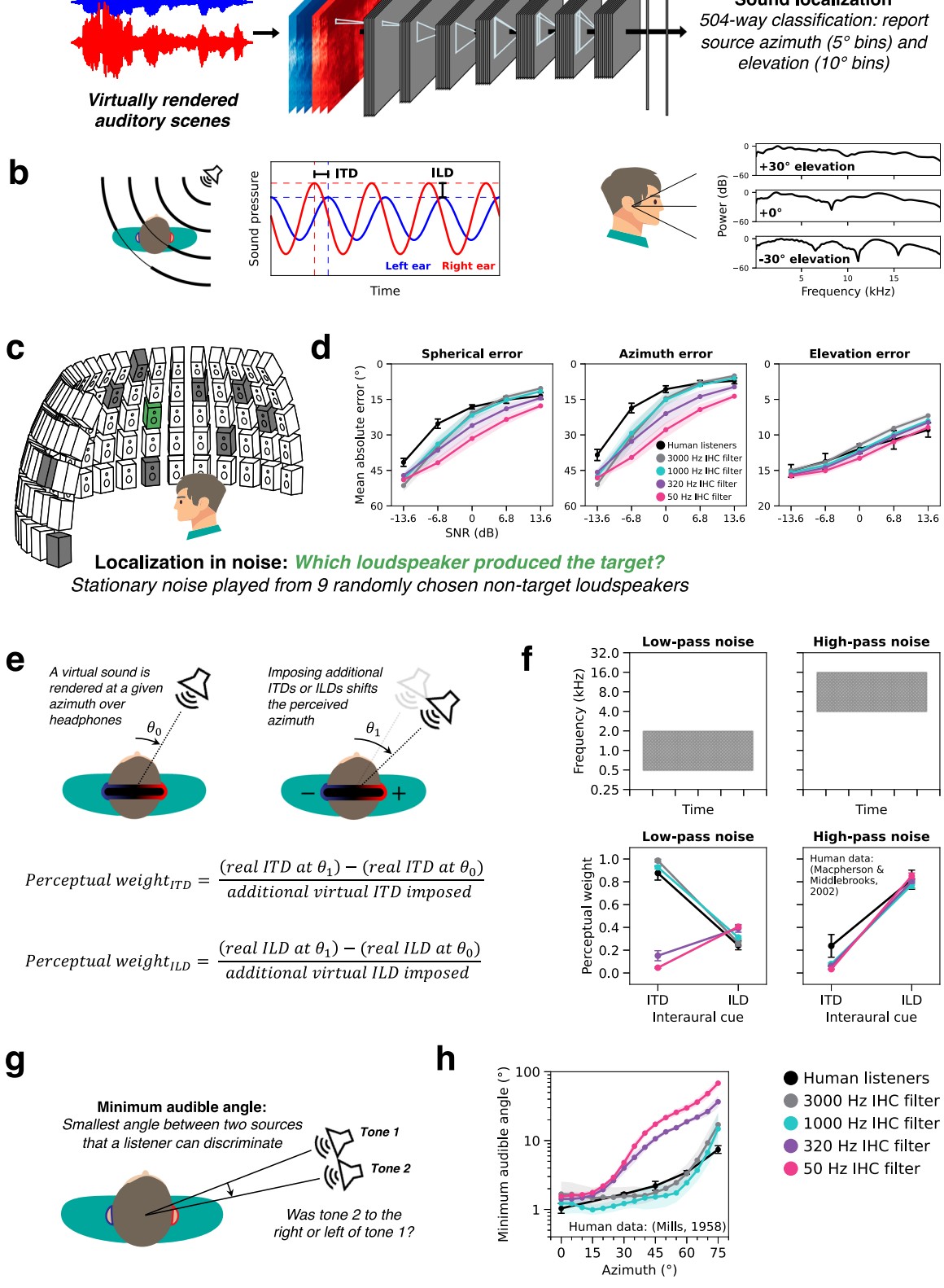

**Localization in noise:** *Which loudspeaker produced the target?*
*Stationary noise played from 9 randomly chosen non-target loudspeakers*

$$Perceptual\ weight_{ITD} = \frac{(real\ ITD\ at\ \theta_1) - (real\ ITD\ at\ \theta_0)}{additional\ virtual\ ITD\ imposed}$$

$$Perceptual\ weight_{ILD} = \frac{(real\ ILD\ at\ \theta_1) - (real\ ILD\ at\ \theta_0)}{additional\ virtual\ ILD\ imposed}$$

## Auditory nerve phase locking is critical for ITD-based sound localization

Biological sound localization relies on three main cues. Small time and level differences between sounds at the two ears provide cues to a source's location in the azimuthal plane (Fig. 3b, left). In addition, before impinging on the ear drum, a sound waveform is altered by the pinna, head, and torso, which boost some frequencies and attenuate others. This anatomical filtering is direction-specific, providing a third cue to a source's location (Fig. 3b, right). Humans rely on these "spectral" cues to judge elevation[59,60]. To investigate the contribution of temporal coding to each of these localization cues, we simulated a set of classic psychoacoustic experiments on the models.

**Fig. 3 | Sound localization is impaired in models with degraded auditory nerve spike timing. a** Localization model schematic. Deep artificial neural networks optimized for sound localization operated on binaural auditory nerve representations of virtually rendered auditory scenes. Nerve representations from the left and right ear were supplied as distinct channels to the first neural network stage. **b** Sound localization cues available to human listeners. Left: interaural time and level differences (ITDs and ILDs) are shown for pure tones recorded at the left and right ear. Right: spectral differences in the anatomical transfer function provide a monaural cue to elevation. **c** Schematic of the sound localization in noise experiment. **d** Mean absolute error for humans ($n = 11$) and models localizing natural sounds in noise are plotted as a function of SNR. The three axes separately plot spherical, azimuth, and elevation errors. $Y$-axes are inverted so that better performance is higher. **e** Schematic of the ITD / ILD cue weighting experiment. The perceptual weights measure the extent to which added ITDs or ILDs shift the perceived azimuth of a virtual sound presented over headphones. **f** ITD and ILD perceptual weights measured with low-pass and high-pass noise from humans ($n = 13$) and models. Note that the noise is the signal to be localized, rather than serving as a masker. **g** Schematic of minimum audible angle experiment. **h** Minimum audible angles plotted as a function of azimuth for human and model listeners. Model error bars always indicate ±2 standard errors of the mean across 10 network architectures per phase locking condition. In (**d**, **f**) human error bars indicate ±2 standard errors of the mean across participants. In (**h**) human error bars indicate ±2 standard errors from 1 listener averaged across 4 different pure tone frequencies (250, 500, 750, and 1000 Hz). Human data in (**f**, **h**) are re-plotted from the original studies[55,68]. Listener schematics in (**b**–**g**) adapted from Francl & McDermott, Nature Human Behaviour, Volume 6, January 2022, reproduced with permission from SNCSC. Source data are provided as a Source Data file.

Humans rely more on interaural time differences (ITDs) at low frequencies and interaural level differences (ILDs) at high frequencies[61]. One demonstration of this comes from measurements of human sensitivity to interaural cue manipulations with virtual sounds[55] (Fig. 3e). In the original experiment, sounds were rendered at different azimuths using a virtual acoustic simulator. Interaural cue sensitivity was inferred from how much a sound's perceived location appeared to shift as additional ITDs or ILDs were added to the binaural waveforms. Shifts in perceived azimuth were mapped back to units of ITD or ILD (specified as the ITD or ILD change corresponding to an actual shift in azimuth by the same amount), allowing interaural cue sensitivity to be quantified as a dimensionless weight: the slope of the response cue value relative to the imposed cue value. For low-frequency sounds, the ITD weight in humans is much larger than the ILD weight. The reverse is true for high-frequency sounds.

Although the encoding of ITDs is thought to make use of phase locking, it was a priori not entirely clear what to expect from the models with altered phase locking limits. The ITDs of natural sounds are present in amplitude envelopes in addition to the fine structure within frequency channels[62–64]. Because envelope modulation rates are usually low, interaural envelope delays should in principle be detectable even if the effective sampling rate of cochlear transduction is lowered via the phase locking limit, and could potentially produce ITD sensitivity without high-fidelity phase locking.

To investigate the contribution of phase locking to this frequency-specific cue dependence, we simulated this experiment on our models (Fig. 3f). Models with high phase-locking limits replicated human behavior, exhibiting high ITD sensitivity only for low frequencies and high ILD sensitivity only for high frequencies. Models with degraded temporal coding (320 and 50 Hz phase locking limits) deviated from human behavior, progressively losing ITD sensitivity at all frequencies and gaining superhuman ILD sensitivity at low frequencies. These results suggest that phase-locked spike timing up to 1000 Hz is necessary for human-like dependence on binaural cues, implicating temporal coding in this aspect of perception.

## Azimuth dependence of human localization requires phase locking

The non-human-like cue dependence under degraded phase locking was also evident in the dependence of localization acuity on azimuth. Human sound localization is best near the midline and becomes less accurate toward the periphery[65–67], as can be quantified by minimum audible angle thresholds[68] (the smallest detectable angular distance between two sources) (Fig. 3g). We simulated an experiment measuring minimum audible angle thresholds for pure tones. Thresholds measured from the 3000 and 1000 Hz phase locking models resembled those of human listeners (Fig. 3h). By contrast, the 320 and 50 Hz phase locking models exhibited a qualitatively different dependence on azimuth, with much higher thresholds away from the midline. These results suggest that ITD cues conveyed by precise spike timing are particularly important for accurate localization away from the midline. This idea is consistent with findings that ILDs are less reliable at lateral azimuths by virtue of varying nonmonotonically with azimuth[69], which might make ITDs critical for lateral localization.

## Physiological model architecture constraints improve predictions of ITD sensitivity

Human listeners are remarkably sensitive to ITDs at low frequencies, but this sensitivity deteriorates at higher frequencies[43]. In principle this sensitivity could be limited by the upper limit of phase locking in the auditory nerve. However, human sensitivity instead declines rapidly above 1 kHz and is fully lost by 1.5 Hz[44]—well below the presumptive 3-5 kHz phase locking limit of the auditory nerve. To better understand this discrepancy, we studied the frequency limits of ITD sensitivity in our models.

ITD sensitivity as a function of frequency has been characterized with ΔITD thresholds with pure tones (single frequencies; Fig. 4a). In such experiments, listeners judge which of two lateralized tones (each with a different ITD) appears further to the right. The ΔITD threshold is the smallest change in ITD needed to reliably discriminate tones in this way. We simulated one such previously published experiment[44], measuring model ΔITD thresholds as a function of frequency. Model ΔITD thresholds were unmeasurably high for frequencies above a model's phase locking limit, as expected (Fig. 4b). Thresholds measured from the 1000 Hz phase locking network produced the closest match to human behavior. The 3000 Hz phase locking model in fact exhibited superhuman ITD sensitivity, with thresholds on the order of 20 μs even up to 2.5 kHz.

This discrepancy with humans is consistent with the known anatomy of the binaural system. Because ITD estimation requires a comparison of input from the two ears, perceptual sensitivity to high-frequency ITDs requires temporal coding at that frequency to be maintained in the auditory system until the stage at which this comparison is made[70,71]. The lower limit of ITD sensitivity in humans is plausibly due to anatomical constraints that force information from each ear to pass through additional synapses before being compared, with some loss of temporal precision at these synapses[45]. But this explanation in turn raises the question of why the auditory system would not have evolved a way to make the comparison happen earlier.

Because the models developed here can be tested in naturalistic conditions, they provide an answer to this question. When tested on naturalistic auditory scenes (natural sounds in noise), the 1000 Hz phase locking model localized just as well as the 3000 Hz model (Figs. 2a and 3d). And across all other psychoacoustic experiments we simulated, there was no significant difference in human-model similarity between the 1000 and 3000 Hz phase locking models (Fig. 2b and Supplementary Fig. 2). These results suggest that temporal coding above about 1000 Hz provides little adaptive benefit.

To test the idea that "early" interaural comparisons accounted for our model's superhuman ITD sensitivity, we altered the neural network

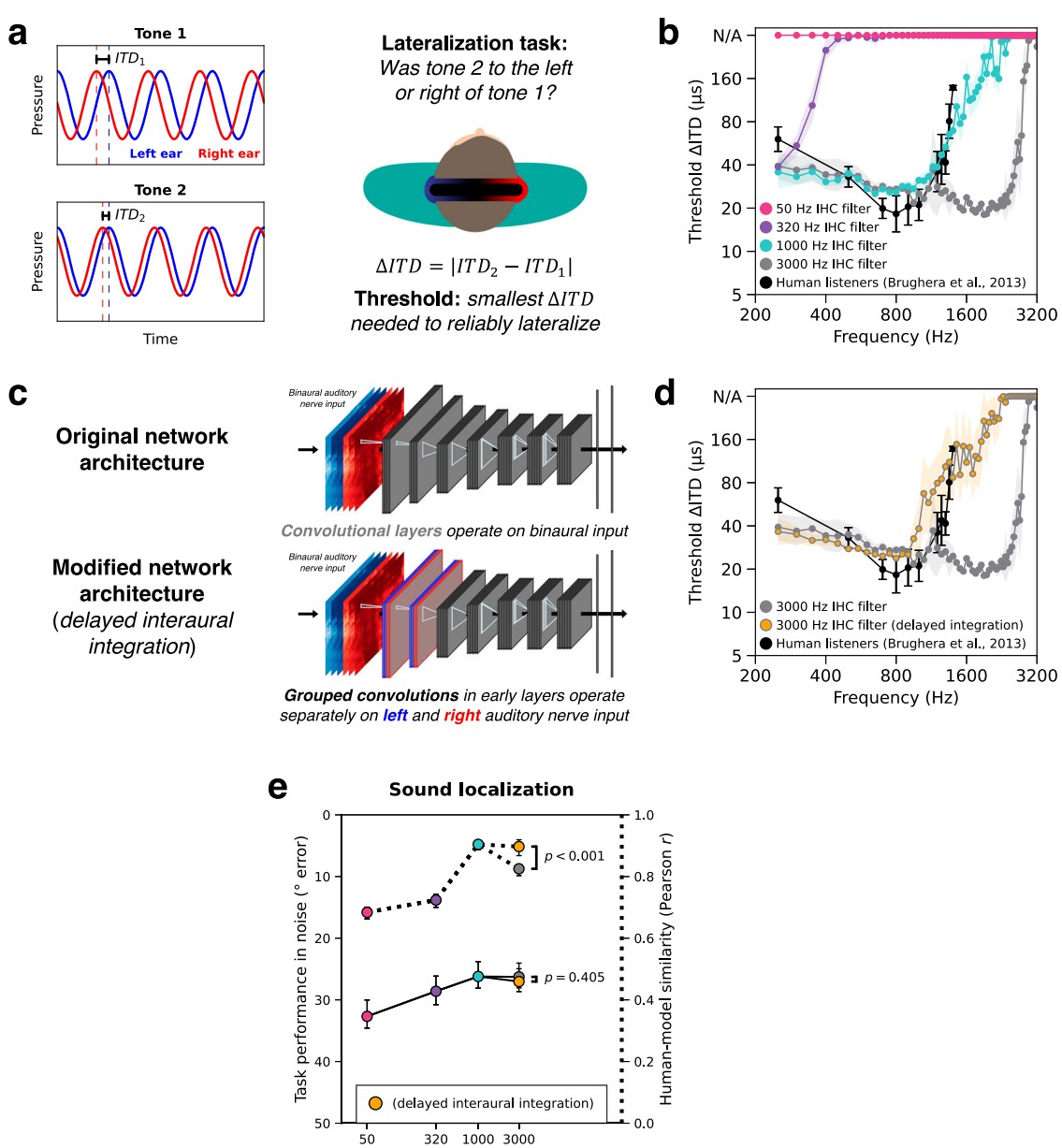

**Fig. 4 | Upper frequency limit of interaural time difference sensitivity.**
**a** Schematic of experiment used to measure ITD sensitivity as a function of frequency. On each trial, listeners heard a pair of pure tones with two different ITDs and judged whether the second tone was located to the right or left of the first. **b** ITD lateralization thresholds measured as a function of frequency from humans ($n = 4$) and models. **c** Schematic of neural network architecture modification to delay binaural integration. Replacing the first two convolutional layers with grouped convolutions (1 group for each ear) forces models to process the ears separately (and to downsample in time due to the inclusion of pooling operations, which reduce the fidelity of temporal coding, analogous to the loss of fidelity that occurs at each synapse in the auditory system) before binaural integration occurs in the first standard convolutional layer. Blue and red represent information from the left and right ears, respectively. **d** ITD lateralization thresholds measured as a function of frequency from humans and models with and without the modified network architectures (both models had the same 3000 Hz phase locking limit in their auditory nerve representation). Error bars in (**b**–**d**) indicate ±2 standard errors of the mean across 4 human participants or 10 network architectures. Human data are re-plotted from the original study[44]. **e** Effect of phase locking limit on sound localization in noise (left y-axis, solid lines) and human-model behavioral similarity (right y-axis, dotted lines). The data is re-plotted from Fig. 2a but now includes the delayed interaural integration model. The statistical significance of differences between models with and without delayed interaural integration was assessed by two-tailed paired comparisons (p-values indicate the probability of obtaining a more extreme score than the delayed model under a null distribution bootstrapped from the non-delayed model). Error bars indicate 95% confidence intervals of the mean bootstrapped across network architectures. Listener schematic in (**a**) adapted from Francl & McDermott, Nature Human Behaviour, Volume 6, January 2022, reproduced with permission from SNCSC. Source data are provided as a Source Data file.

architectures slightly to delay interaural integration (Fig. 4c). We replaced the standard convolution operations in the earliest neural network stages with "grouped" convolution operations (with one group for each ear; see Methods), such that the resulting models must initially process information from the left and right auditory nerve separately. Reasoning that synapses introduce temporal jitter that effectively imposes low-pass filtering[72], interaural integration was only allowed to occur in the models after early temporal pooling layers that downsample in time, reducing temporal fidelity. We note that this is a relatively weak biological constraint in the context of the detailed models of binaural processing stages[70,71,73–75] that are used elsewhere in our field. We note also that there is evidence for enhancement of the

precision of low-frequency phase locking in the brainstem[45], in addition to the loss of higher frequency phase locking that we modeled here. It nonetheless seemed useful to assess whether a minimalistic biologically inspired constraint would be sufficient to replicate human behavior.

We trained these modified neural network architectures with 3000 Hz phase locking auditory nerve input and evaluated them on the full set of sound localization experiments. Consistent with our hypothesis, the models lost sensitivity to high-frequency ITDs (like humans; Fig. 4d) but were otherwise unaffected (Supplementary Fig. 2). Delaying interaural integration thus increased the overall human-model similarity score (which aggregates results across all experiments) for the 3000 Hz model ($p < 0.001, d = 12.0$, evaluated by bootstrapping across 10 neural network architectures; Fig. 4e, dotted line). These results indicate that additional physiological constraints can in some cases produce better matches to human behavior. Moreover, delaying interaural integration did not impair localization performance in noise ($p = 0.405, d = 0.783$, comparing localization error between the delayed and non-delayed 3000 Hz models; Fig. 4e, solid line). This latter result provides a normative explanation for the solution that biological auditory systems have arrived at over evolution, as it suggests there is little cost to real-world behavior when integration is delayed. We note that the results are equally consistent with the possibility that the cutoff of phase locking in humans is substantially lower than 3000 Hz (i.e., lower than in other mammals, as some have argued[32]), and would also provide a normative justification for such a lower cutoff from the standpoint of sound localization.

### Not all localization phenomena are inextricably linked to phase-locked spike timing

Although removing phase locking caused pronounced discrepancies with human behavior (Supplementary Fig. 2a–f), some behaviors were relatively unaffected. All models exhibited the "precedence effect", in which localization judgments are dominated by the initial part of a sound[76] (Supplementary Fig. 2g; evidently the models without phase locking learned to prioritize ILD cues from the onset of a sound in order to localize accurately amid reflections). All models also exhibited human-like dependencies of localization accuracy on bandwidth[77] (Supplementary Fig. 2h) and of elevation accuracy on high-frequency spectral cues[56,57,78] (Supplementary Fig. 2i). However, we did find that models without access to phase locking became abnormally dependent on spectral cues for azimuthal localization (Supplementary Fig. 3), evidently to make up for the impaired binaural information that results from impaired phase locking. This latter result provides further evidence for the importance of phase locking to human spatial hearing.

### Model optimization – word and voice recognition

To model speech perception, we optimized models to recognize words and voices using the Word-Speaker-Noise dataset[79] (Fig. 5a), consisting of 2 s speech excerpts superimposed on real-world background noise. Training on this dataset has previously been shown to produce models that yield the best current predictions of auditory cortical responses[80]. Models were jointly optimized to classify stimuli according to the word that appeared in the middle of the excerpt (794-way word recognition task) and the talker that produced the utterance (433-way voice recognition task). These tasks were intended to capture some of the challenges of everyday speech and voice recognition (background noise, large numbers of classes, high degree of variability), subject to practical constraints of dataset generation and model optimization (see Methods). We also trained models on each task individually and found similar results (Supplementary Figs. 4 and 5). We present results from the joint-task model here.

### Phase locking improves voice recognition more than word recognition in real-world noise

We first measured human and model word recognition performance as a function of SNR in different types of background noise: recorded auditory scenes, speech babble, instrumental music, and stationary speech-shaped noise (Fig. 5b). Lowering the phase locking limit produced modest deficits for model word recognition accuracy in some of the conditions, with no detectable effect in others. To the extent that there was a benefit from phase locking, it occurred between the 50 Hz and 320 Hz conditions.

We next measured voice recognition performance in the same models on the same stimuli (Fig. 5c); it was not possible to run human participants in this experiment (see below). At low SNRs, models with access to phase locking performed better than models without. As with word recognition, almost all the benefit from phase locking occurred below 320 Hz, suggesting phase locking up to but not above the F0 of most human speech improves voice recognition in noise.

To further search for naturalistic noise conditions in which phase-locked spike timing might contribute to word recognition, we measured human and model word recognition in each of 43 different real-world auditory textures[81] (Fig. 5d; Supplementary Fig. 6). At a fixed SNR of −3 dB, these different textures produced a wide range of human word recognition scores (25% to 80% correct; this variation was highly reliable, with Spearman-Brown corrected split-half reliability = 0.968). Models tested on the same stimuli produced similar word recognition scores as humans, accounting for about 95% of the explainable variance in the human data. This similarity again held regardless of the phase locking limit (Fig. 5e). However, a scatter plot comparing word recognition scores between the 50 and 3000 Hz phase locking models (Fig. 5f, left) shows a small benefit of phase locking for word recognition (mean benefit = +2.3%; standard deviation = 2.3%), and inspection of results for individual textures reveals that larger benefits (5–8%) occurred for a few types of noise (Supplementary Fig. 6a). This result extends previous findings that neural networks trained to recognize speech can replicate patterns of human speech intelligibility[16,82–84], and further underscores a small benefit of high-fidelity phase locking for word recognition in noise (for monaural conditions in which localization cues cannot aid performance).

An analogous scatter plot of model voice recognition scores measured with the same stimuli (Fig. 5f, right) shows a considerably larger benefit of phase locking than for word recognition (mean benefit = +14.9%; standard deviation = 5.7%), with effects as large as 22–29% for particular types of noise. Comparing absolute voice recognition performance to that of humans is practically challenging because listeners do not all recognize the same voices and overall accuracy depends strongly on listener familiarity with voices. Instead, we asked whether the qualitative characteristics of human voice recognition were shared by the models, and whether this depended on the phase locking limit.

### The dependence of human voice recognition on absolute pitch requires phase locking

Humans rely in part on absolute pitch to recognize voices. When a familiar talker's voice is F0-shifted or made inharmonic (by frequency-jittering its harmonic components to be inconsistent with any single F0) (Fig. 6a), the voice is less recognizable[85]. To assess whether these characteristics were shared by the models and if they depended on phase locking, we measured human and model voice recognition with F0-shifted (Fig. 6b, closed symbols) and inharmonic speech (Fig. 6c, closed symbols). Models were tested on familiar voices (but held-out speech utterances) from the training set. Humans were tested on celebrity voices, as in prior work[85].

There is no way to match the relative familiarity of test voices between human and model participants, confounding comparisons of absolute performance. However, models with access to phase-locked

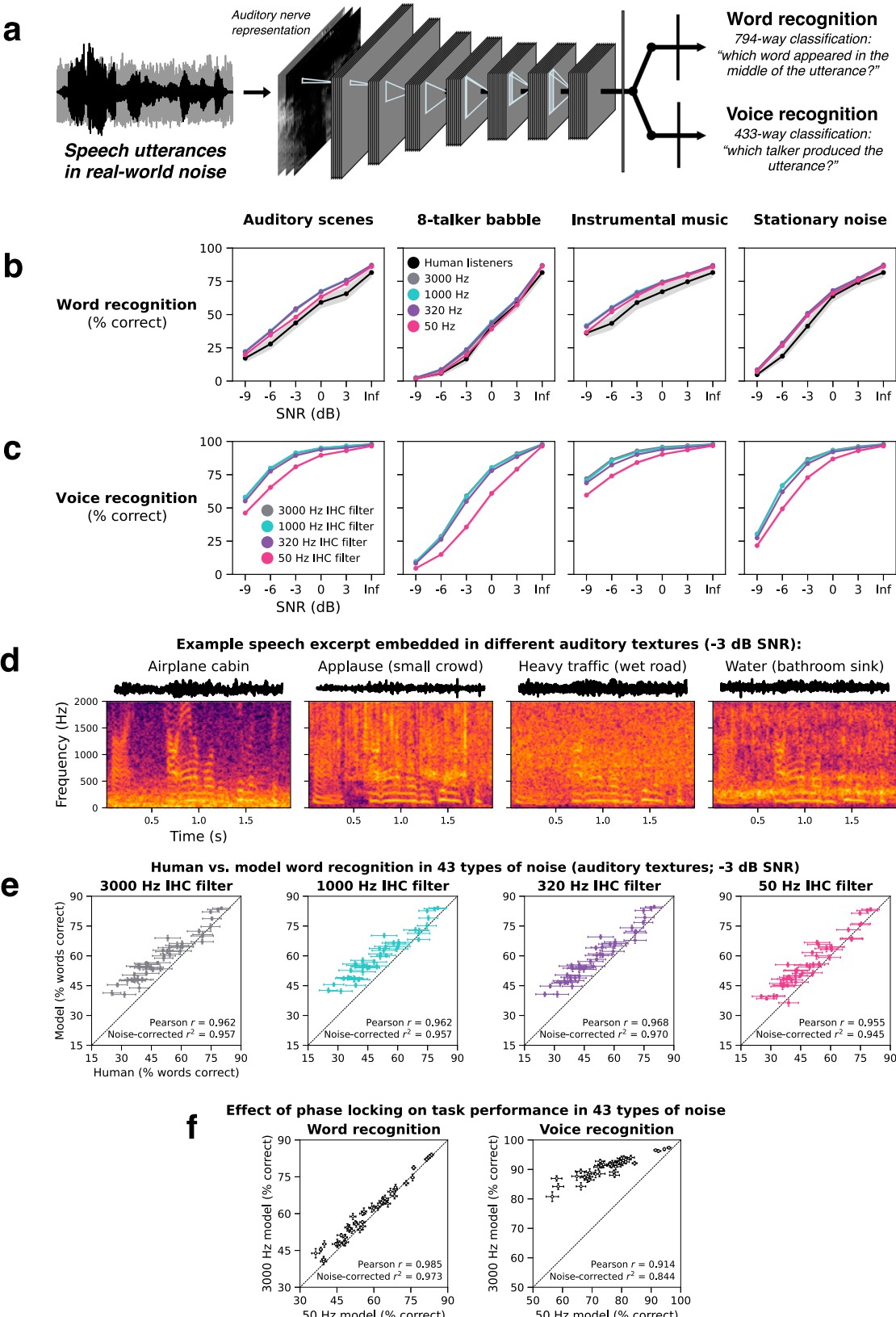

**a** Speech utterances in real-world noise → Auditory nerve representation → **Word recognition** *794-way classification: "which word appeared in the middle of the utterance?"* / **Voice recognition** *433-way classification: "which talker produced the utterance?"*

**b** Word recognition (% correct) — Auditory scenes, 8-talker babble, Instrumental music, Stationary noise (Human listeners, 3000 Hz, 1000 Hz, 320 Hz, 50 Hz; SNR (dB): -9 -6 -3 0 3 Inf)

**c** Voice recognition (% correct) — (3000 Hz IHC filter, 1000 Hz IHC filter, 320 Hz IHC filter, 50 Hz IHC filter; SNR (dB): -9 -6 -3 0 3 Inf)

**d** Example speech excerpt embedded in different auditory textures (-3 dB SNR): Airplane cabin, Applause (small crowd), Heavy traffic (wet road), Water (bathroom sink)

**e** Human vs. model word recognition in 43 types of noise (auditory textures; -3 dB SNR)
- 3000 Hz IHC filter: Pearson r = 0.962, Noise-corrected $r^2$ = 0.957
- 1000 Hz IHC filter: Pearson r = 0.962, Noise-corrected $r^2$ = 0.957
- 320 Hz IHC filter: Pearson r = 0.968, Noise-corrected $r^2$ = 0.970
- 50 Hz IHC filter: Pearson r = 0.955, Noise-corrected $r^2$ = 0.945

**f** Effect of phase locking on task performance in 43 types of noise
- Word recognition: Pearson r = 0.985, Noise-corrected $r^2$ = 0.973
- Voice recognition: Pearson r = 0.914, Noise-corrected $r^2$ = 0.844

spike timing best replicated the qualitative properties of human behavior. Human voice recognition was best for voices at their natural F0 and fell off with progressively larger shifts in either direction, as in prior work[85]. Human performance was also impaired by making voices inharmonic. Models with the 50 Hz phase locking limit exhibited superhuman robustness to these F0 manipulations, suggesting the reliance on absolute pitch evident in humans only emerges with the aid of phase locking. The presumptive explanation is that pitch cues from phase locking aid performance in noise, such that models with phase locking learn a recognition strategy that uses these cues. This strategy leaves them dependent on these cues and thus produces worse performance when F0 is altered. By contrast, the model without phase

**Fig. 5 | Auditory nerve spike timing improves voice recognition more than word recognition in real-world noise. a** Speech model architecture and task. Deep artificial neural networks were jointly optimized to recognize words and voices from simulated auditory nerve representations of speech in noise. The two tasks shared all model stages up to the final task-specific output layers. **b** Human (*n* = 44) and model word recognition as a function of SNR. Each panel plots task performance in a different naturalistic noise condition[16]. **c** Model voice recognition as a function of SNR. It was not possible to run humans in this experiment as human participants would not be familiar with the specific voices the model was trained to recognize. **d** Spectrograms of the same speech excerpt embedded in different

auditory textures. **e** Human (*n* = 47) vs. model word recognition scatter plots for speech embedded in each of 43 distinct auditory textures at −3 dB SNR. Each data point represents the human and model word recognition score for a single auditory texture. **f** Effect of phase locking on model word and voice recognition in 43 distinct auditory textures. The left scatter plot compares word recognition performance for the 50 and 3000 Hz IHC filter models. The right scatter plot compares voice recognition performance for the 50 and 3000 Hz IHC filter models. All error bars indicate ±2 standard errors of the mean across human participants or 10 network architectures. Source data are provided as a Source Data file.

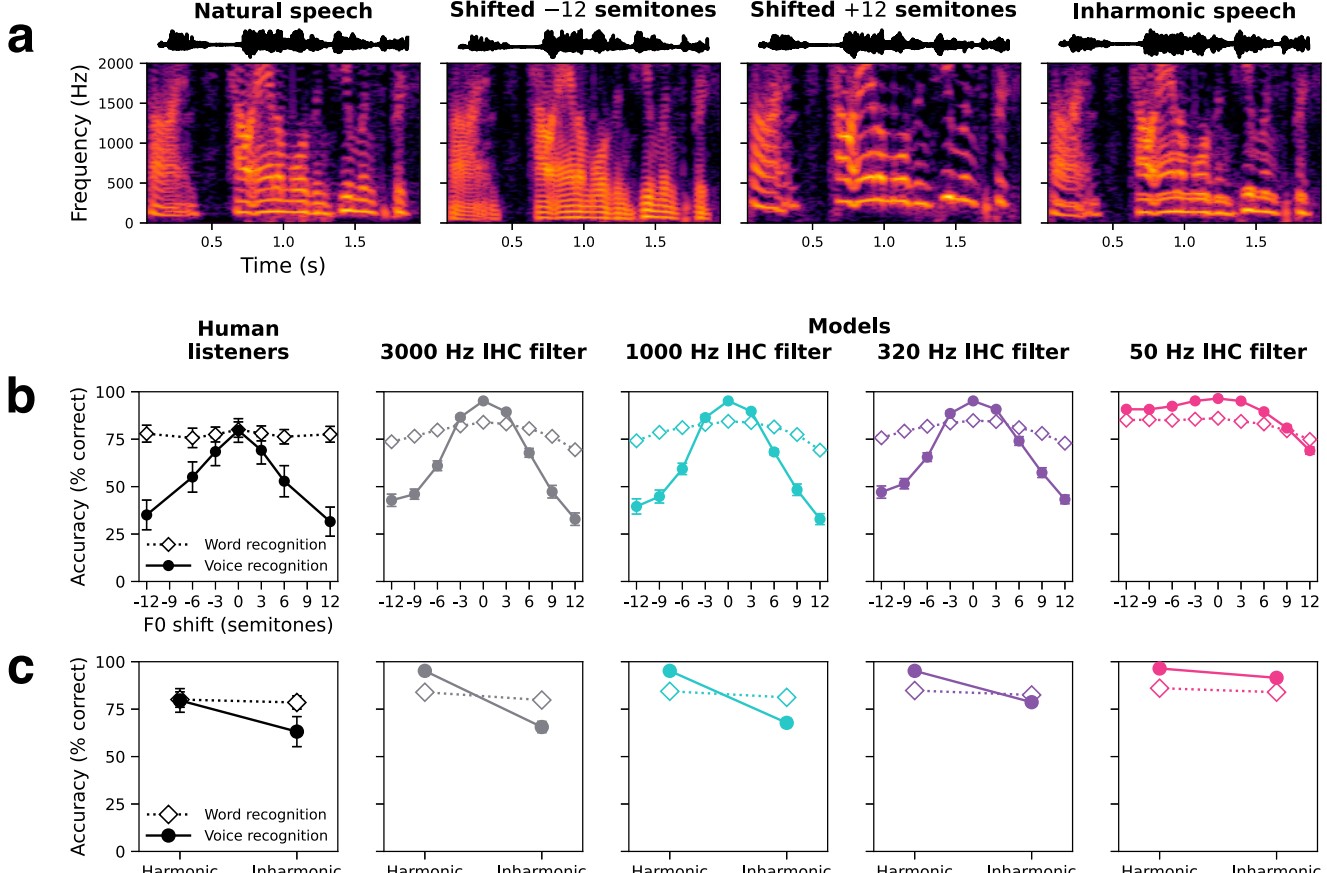

**Fig. 6 | Auditory nerve spike timing is critical for human-like voice recognition. a** Stimuli for F0-altered word and voice recognition experiments. Spectrograms show the same speech excerpt resynthesized in four different pitch conditions: unmodified (natural), F0-shifted down 12 semitones, F0-shifted up 12 semitones, and inharmonic. In the inharmonic condition, harmonic frequency components were randomly frequency-shifted such that they were no longer integer multiples

of a common F0 and were no longer linearly spaced in frequency. **b** Word and voice recognition accuracy for humans and models tested on F0-shifted speech. **c** Word and voice recognition accuracy for humans and models tested on harmonic and inharmonic speech. All error bars indicate ±2 standard errors of the mean across human participants (*n* = 22 for word recognition; *n* = 95 for voice recognition) or 10 network architectures. Source data are provided as a Source Data file.

locking does not use this cue and so is more robust to its alteration. These results provide additional evidence for a role of phase locking (up to about 320 Hz) in human voice recognition and pitch perception.

We also measured human and model word recognition with F0-shifted and inharmonic speech (Fig. 6b, c, open symbols). In contrast to the results for voice recognition, human word recognition was unaffected by these F0 manipulations. Model performance was similarly robust regardless of phase locking limit, remaining comparable to that for humans in all conditions.

### Phenomena previously linked to phase locking – effect of tone vocoding

In addition to general proposals that phase locking aids speech perception in noise, phase locking has been linked to two specific effects

on human speech recognition. The first is the effect of tone vocoding—a signal manipulation intended to remove information conveyed by phase locking, which in some conditions produces deficits in speech intelligibility[38]. The second is the benefit of spatial separation between sound signals, which exhibits individual differences across human listeners, especially in reverberant listening conditions[39,86]. These individual differences have been proposed to be mediated by the integrity of temporal coding, as is presumably dependent on the extent of nerve fiber survival. We asked whether the models would exhibit these effects, and whether this depended on access to high-fidelity phase locking.

The tone vocoding manipulation first decomposes a speech waveform into frequency bands, using a simulated cochlear filter bank[38] (Fig. 7a). The temporal envelopes of each band are extracted

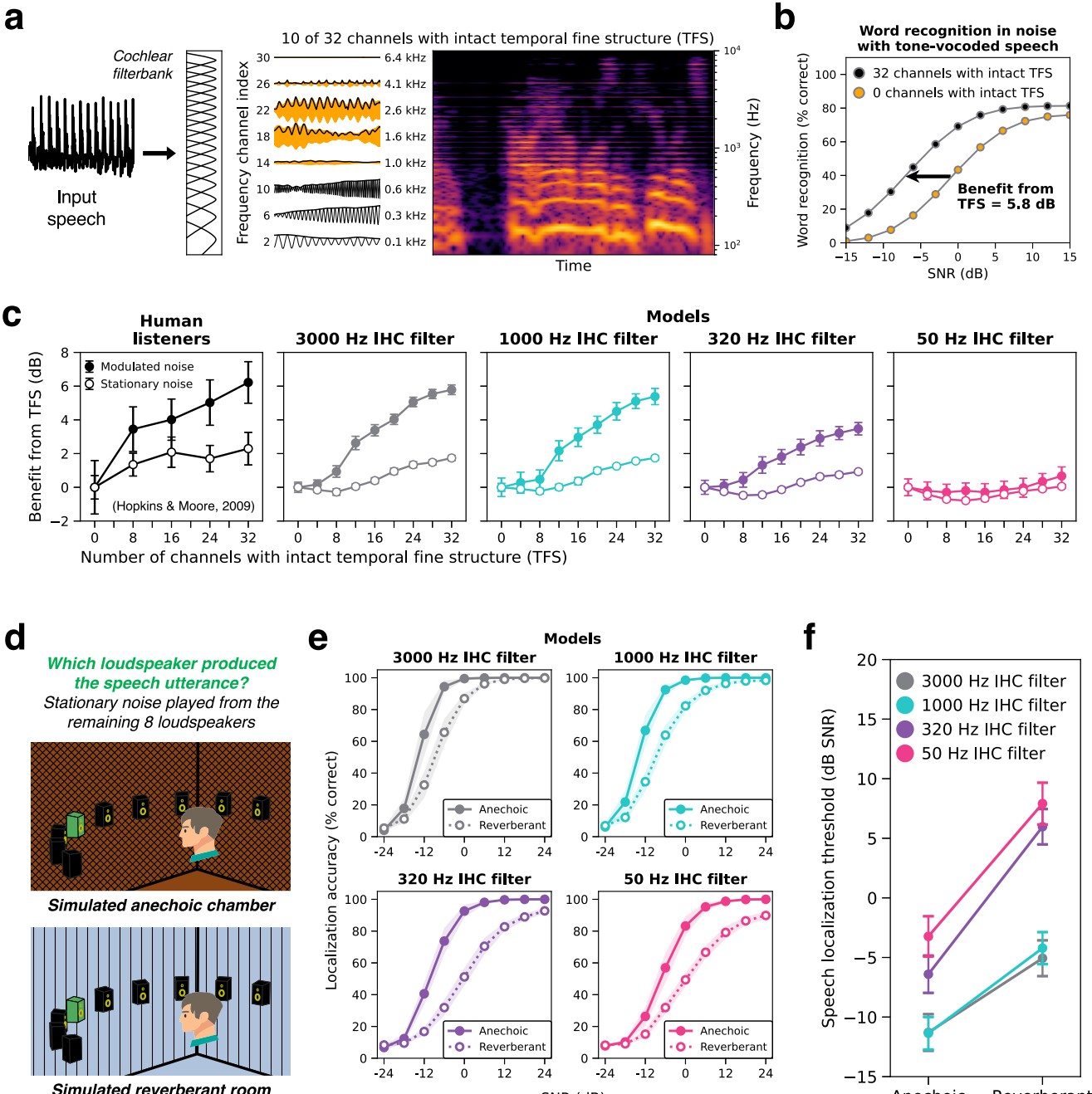

**Fig. 7 | Auditory nerve phase locking is needed to account for phenomena previously linked to temporal fine structure. a** Schematic of tone-vocoding stimulus manipulation with a "cutoff channel" of 10. A speech waveform was separated into 32 frequency bands by a band-pass filter bank that mimics the cochlea's frequency tuning. Frequency channels up to and including the cutoff channel were left intact. In frequency channels above the cutoff, temporal fine structure (TFS) was disrupted by replacing the band with a pure tone carrier at the channel's center frequency, amplitude modulated by the envelope of the original band. **b** The benefit from temporal fine structure was quantified by plotting word recognition accuracy vs. SNR and measuring leftward shifts in these psychometric functions as the cutoff channel (i.e., the number of channels with intact temporal fine structure) was increased. All shifts were computed relative to performance with fully tone-vocoded speech (0 channels intact, orange circles). **c** Tone vocoding results. The benefit from temporal fine structure—measured from humans and models—is plotted as a function of the number of channels with intact temporal fine structure. Open circles plot the benefit in stationary noise and closed circles plot the benefit in

amplitude-modulated noise. Human data in (**c**) is re-plotted from the original study[38] and errors bars indicate ±1 standard error of the mean across 10 participants. **d** Schematic of the speech localization experiment in anechoic and reverberant conditions. **e** Model sound localization accuracy as a function of SNR and reverberation. Panels plot performance in a simulated anechoic (solid symbols) and reverberant (open symbols) room for each phase locking model. Although the qualitative effects shown here have been documented in humans, the experiment we used to measure the effects in our model had not been conducted in human listeners, and so we do not have an explicit comparison to human data. **f** The effect of phase locking and reverberation condition on speech localization thresholds measured from the psychometric functions in (**e**). Model error bars in (**c**–**f**) indicate ±2 standard errors of the mean across 10 network architectures. Listener schematics in (**d**) adapted from Francl & McDermott, Nature Human Behaviour, Volume 6, January 2022, reproduced with permission from SNCSC. Source data are provided as a Source Data file.

and imposed on pure tone carriers at the center frequency of each channel that are then summed. This procedure produces a new waveform with similar envelope cues to the original but less informative fine structure (because the tone carriers are constant over time and fixed across stimuli). Hopkins and Moore investigated the contribution of fine structure to speech intelligibility in noise by tone vocoding all frequency channels above a given cutoff. The authors increased this cutoff from 0 (all channels vocoded) to 32 (no channels vocoded), progressively increasing the upper frequency limit of fine structure information preserved in the stimulus. Speech reception thresholds were measured as a function of this cutoff in two types of noise (stationary and modulated; Fig. 7b). In stationary noise, thresholds improved somewhat as the cutoff increased. However, this improvement was more pronounced in modulated noise, with significant improvements up to 24 channels (corresponding to 4102 Hz; Fig. 7c, leftmost panel). This result was taken to suggest that humans benefit from the information in the monaural fine structure of sound waveforms, potentially upwards of 1000 Hz.

We tested our models on the same stimulus manipulation (Fig. 7c, right panels). Models with 3000 and 1000 Hz phase locking limits best replicated the human pattern of behavior, with speech reception thresholds improving as more high-frequency fine structure was preserved, particularly in modulated noise. The benefit from fine structure information was reduced in the 320 Hz phase locking model and fully eliminated by 50 Hz. This effect drove the lower overall human-model similarity for word recognition in the 50 Hz model (Fig. 2b). We note that even the 3000 Hz phase locking model showed a smaller benefit than humans for 8 vs. 0 channels, possibly because the filter bank used to vocode stimuli in the model experiment differed slightly from that used in the human experiment (see Methods).

These model results are consistent with the qualitative interpretation of the original human results[38] as implicating phase locking in this particular effect on speech intelligibility. However, they suggest that the frequency dependence of the tone vocoding manipulation is not directly related to the frequencies of phase locking used by the brain, contrary to the intention of the original manipulation. Specifically, word recognition in the models benefitted from added high frequency information beyond their respective phase locking limits. For instance, the 1000 Hz model received a benefit from frequencies well above 1000 Hz. Since the simulated auditory nerve representations cannot encode temporal fine structure so far above the phase locking limit, the performance improvement from high frequencies cannot be driven by high frequency phase-locked spike timing.

One alternative explanation is that tone vocoding interferes with harmonic frequency relationships, such that when high-numbered harmonics are vocoded, they no longer produce temporal envelope variations at the F0 (which the auditory nerve encodes via phase locking to the F0). Because pitch is an important cue for sound segregation, disrupting the encoding of F0 could produce speech recognition deficits. Consistent with this alternative explanation, model word recognition exhibited very similar deficits in noise for inharmonic speech[87] as for tone-vocoded speech (Supplementary Fig. 7), with similar interactions with phase locking.

These results suggest phase-locked spike timing is needed to comprehensively account for human word recognition behavior. We note that the phase-locking-dependent effects of tone vocoding were present even in models that were only optimized for word recognition (Supplementary Fig. 4g). This suggests that the modest benefit of phase locking on word recognition task performance (Figs. 2a, 5, and Supplementary Fig. 6) is enough to produce a strategy that incorporates phase locking to some extent. However, the magnitude of the tone vocoding effect was somewhat larger in models that were jointly optimized for word and voice recognition (5.8 dB compared to 4.1 dB for models optimized only for word recognition; Supplementary Fig. 8). This raises the possibility that the dependence of human-like word recognition on phase locking is partly a consequence of sharing machinery with tasks that benefit more from phase locking (voice recognition being one candidate).

## Phenomena previously linked to phase locking – localization of speech

Better encoding of temporal fine structure has also been proposed to be correlated with the ability to direct spatial attention to voices in challenging acoustic environments[39,86]. Although our models did not possess selective attention, we could test whether phase locking would enable better localization of speech in noisy and reverberant environments, as would be necessary to direct spatial attention. We simulated a localization-in-noise experiment in which listeners reported which of 9 loudspeakers (2 m away, spanning −80° to +80° azimuth in 20° steps) produced a speech utterance, with threshold-equalizing noise[58] played from the remaining 8 loudspeakers (Fig. 7d). We measured model performance as a function of signal-to-noise ratio in both a simulated anechoic chamber and in a moderately reverberant room (RT60 = 1 s). All models performed worse in reverberation (Fig. 7e), but the degraded 320 and 50 Hz phase locking models were particularly impaired, producing a significant interaction between phase locking and room condition ($F(3, 36) = 75.27, p < 0.001, \eta^2_{partial} = 0.86$) (Fig. 7f). These results suggest that fine-grained temporal coding should help listeners attend to individual voices in challenging acoustic environments (e.g., a cocktail party), consistent with previous proposals[39,86]. The other major cue for selective attention in cocktail party scenarios is the sound of a target talker's voice, in particular the voice F0[88,89]. Given that phase locking is needed for voice recognition and for the representation of the voice F0 (Fig. 6), our results collectively suggest that the cues for auditory attention are likely to be compromised without intact phase locking.

## Replication with simplified cochlear model

State-of-the-art cochlear models that best capture the nonlinear response properties of the auditory nerve are computationally expensive, which can limit their integration into larger-scale models of the auditory system. To investigate whether the fine-grained details of these models are critical to account for behavior, we also optimized models with a simplified cochlear front-end. This front-end consisted of a linear cochlear filter bank followed by half-wave rectification and low-pass filtering (to impose an upper limit on phase locking), the output of which was passed through sigmoid functions approximating the rate-level functions of high-, medium-, and low-spontaneous-rate fibers[46]. We repeated the temporal coding manipulation by setting the low-pass filter cutoff to 3000, 1000, 320, and 50 Hz. In addition to testing the importance of a detailed cochlear model, the simplified model also served to rule out the possibility that effects observed with the detailed cochlear model were driven by unintended nonlinear consequences of adjusting filter parameters rather than degraded temporal coding per se.

Models with the greatly simplified cochlear stage qualitatively and in most cases quantitatively replicated the results obtained with the highly detailed model of the auditory nerve (Supplementary Figs. 9, 10, and 11). This result suggests that the effects we saw with the detailed cochlear model are not due to unintended interactions between its components. The results also indicate that future work could use the simplified model in many settings without a cost.

## Comparison of machine learning models with ideal observers

The approach taken in this paper is predicated on the idea that an optimized machine learning model can approach the characteristics of an ideal observer. To test the plausibility of this assumption, we trained neural network models on a task for which provably optimal observers can be derived: frequency discrimination. We trained 120 different convolutional neural network architectures on the task and selected

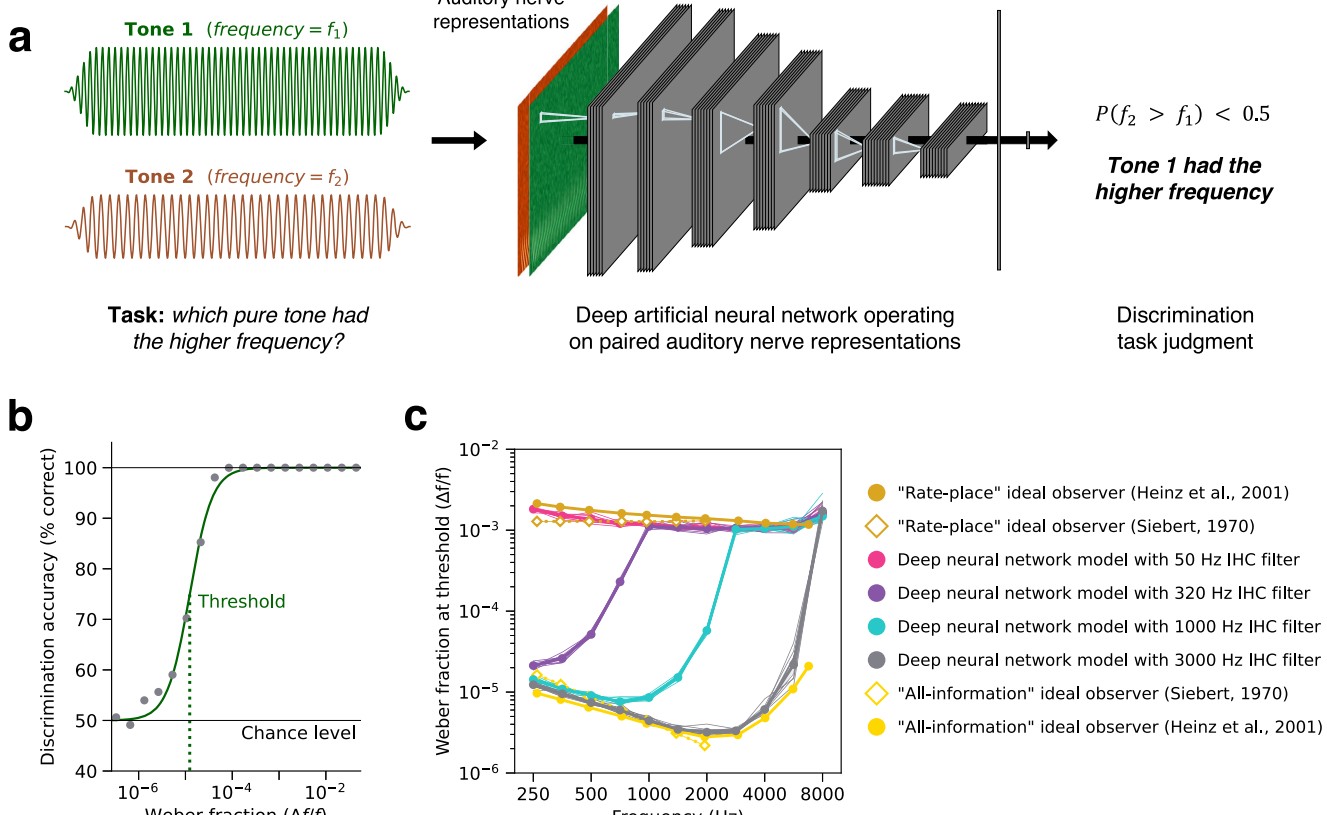

**Fig. 8 | Deep neural networks optimized for pure tone frequency discrimination closely approximate previous ideal observer models. a** Schematic of deep neural network frequency discrimination model. The two tones were passed through an auditory nerve model and then provided as input to a convolutional neural network as separate channels (shown in brown and green). The levels of the tones were varied independently. **b** Model frequency discrimination thresholds were computed from psychometric functions measuring pure tone discrimination accuracy as a function of frequency difference, expressed as the Weber fraction on a log-scale. **c** Frequency discrimination thresholds measured from previous ideal observer models and deep neural network models with different phase locking limits. Thresholds for the ideal observer models (gold and yellow markers) were re-plotted from Ref. 12. Siebert (1970) analytically and Heinz et al. (2001) computationally derived the optimal task performance of models with access to either all the available information ("all-information") or only the "rate-place" (i.e., time-averaged) information in auditory nerve representations. Deep neural network model thresholds are plotted as the mean across 10 network architectures for each phase locking conditions (thick pink, purple, blue, and grey lines; error bars indicate ±2 standard errors of the mean). Thin lines plot thresholds from individual network architectures. Source data are provided as a Source Data file.

the 10 top-performing architectures. Simulated auditory nerve representations of the two stimuli (200 ms pure tones of different frequencies) were supplied to the models as different input channels (Fig. 8a), with models separately optimized for the four phase locking cutoffs used elsewhere in this paper. We measured discrimination thresholds from psychometric functions generated from the model judgments (Fig. 8b), using the same stimulus conditions with which previously published ideal observers for this task were evaluated. As shown in Fig. 8c, the optimized neural network model with the lowest phase locking cutoff (50 Hz) closely approximated the "rate-place" ideal observer that operates exclusively on firing rates. By contrast, the model with the highest phase locking cutoff (3000 Hz) closely approximates the ideal observer that uses spike timing in addition to firing rates. The two intermediate phase locking cutoffs produce results intermediate between the two ideal observers. This result shows that machine learning models of the sort used in this paper can achieve results that are close to optimal for simple tasks, bolstering the idea that the results shown here for more complex tasks may also be indicative of characteristics of ideal observers.

## Discussion

We developed models of real-world sound localization, voice recognition, and word recognition by optimizing artificial neural networks to classify simulated auditory nerve representations of natural sounds.

The resulting models closely replicated human behavior for natural sounds as well as for synthetic experimental stimulus manipulations, despite being optimized solely for task performance with natural sounds. To investigate the perceptual role of temporal coding in human hearing, we separately optimized models with lower auditory nerve phase locking limits, measuring the effect on task performance in naturalistic conditions as well as on human-model similarity across different stimulus conditions. The phase locking manipulation impaired performance for all three tasks, though the effect was larger for sound localization and voice recognition than for word recognition. Moreover, patterns of behavioral performance deviated from those in humans in at least some experimental conditions for each of the three task domains when the phase locking limit was too low (or too high, in one case). This combination of results provides evidence that phase locking is used in human perception, and suggests that both binaural and monaural mechanisms for extracting information from spike timing must exist in the auditory system, and that models of human hearing must operate on high temporal resolution input if they are to accurately capture behavior. But the results also provide a normative perspective on why temporal coding is used, because the dependence of human-model similarity on the phase locking limit resembled that of task performance. In particular, the extent of temporal coding needed to explain human voice and word recognition was lower than that needed to explain human sound localization, and this

was reflected in the effect on task performance. This finding suggests that different domains likely use phase locking to different extents depending on its utility for natural behavior. The results also underscore that phase locking is critical for the two main attentional cues to speech (location and voice).

The comparison of machine learning models to humans also provided insight into additional biological constraints that influence human performance. Models whose neural networks had immediate access to the left and right auditory nerve representations exhibited superhuman sensitivity to interaural time differences (at much higher frequencies than is seen in humans). Simply altering the model architecture to require monaural processing stages (and thus some loss of temporal fidelity, as in the mammalian binaural system[33,34,45]) before interaural integration produced more human-like behavior, accounting for results in every experiment we considered. In addition to yielding a more comprehensive model of human localization behavior, here the main contribution of our approach was to help understand why interaural comparisons have an upper frequency limit below the presumptive upper limit of phase locking in biological auditory systems. Our results suggest that the human auditory system did not evolve to use interaural time differences much above 1000 Hz because there is little benefit to sound localization in natural settings (perhaps because time differences become ambiguous when wavelengths are short relative to head size). This set of results illustrates how our modeling approach enables normative understanding of sensory physiology.

## Relation to prior modeling work

Our approach draws inspiration from ideal observer theory, an early application of which was to investigate the role of temporal coding in hearing[2,12] in simpler settings. Siebert first derived optimal solutions to frequency discrimination given the information available in auditory nerve responses to synthetic tones. He showed that optimal observers using different features of neural coding yielded different patterns of performance, but the resulting models severely overestimated human performance. Later instantiations of this approach applied to more realistic models of the nerve encountered similar issues[12]. The overestimation of human performance in such settings is perhaps unsurprising given that there is little reason to believe that the human auditory system has been optimized for discriminating pure tones. This is a central limitation of the classical ideal observer approach: the simple tasks for which it is tractable to derive ideal observers are not those that likely drove biological optimization processes. And for the perceptual tasks that humans are plausibly optimized for (e.g., recognizing and localizing natural sounds)[90], the derivation of provably ideal observers is intractable. The present results show how the toolbox of contemporary deep learning provides an avenue to resurrect the ideal observer approach for real-world perceptual tasks. Even though the resulting models are almost surely not fully optimal for the tasks they are optimized for, they permit scientific inferences about the consequences of optimization under biological constraints.

Other previous models proposed candidate neural mechanisms for the extraction of timing information[91–93]. The main limitation of mechanistic models is that they do not make extensive behavioral predictions, and thus are difficult to test in the absence of direct neural evidence for the mechanism in question. Our approach is complementary: we employ models that make behavioral predictions, enabling the role of timing information to be tested noninvasively, but at the expense of not addressing the underlying neural circuitry. Our results nonetheless place some constraints on the underlying circuit mechanisms by revealing the range over which phase locking matters for real-world tasks. Phase-locked spike timing up to around 1000 Hz seems to be most critical for localization. Whereas for the word and voice tasks virtually all benefit of phase locking incurs between 50 and 320 Hz. These results implicate monaural mechanisms for extracting

phase locking in the range of hundreds of Hz, consistent with at least one recent mechanistic proposal[94].

The models developed here extend recent efforts to apply deep learning to auditory modeling[16,18,19,54,80,95]. In addition to providing models with fairly realistic simulations of the peripheral auditory system (with an appropriate number of spiking nerve fibers), the present experiments demonstrate substantially more extensive evidence for close matches to human behavior than were available in previous work. In particular, we show that task-optimized models replicate human localization in noise (Fig. 3d), the upper frequency limit of ITD discrimination (Fig. 4b and d), patterns of word recognition performance across a large set of natural background noises (Fig. 5e), and patterns of voice recognition performance across F0 manipulations (Fig. 6b, c). These results provide additional evidence that much of auditory perception can be accounted for with task optimization.

## Relation to prior psychophysical work

A long tradition of psychophysical research has also attempted to test the role of phase locking in perception[35–38,40,96]. Such studies have typically used stimuli intended to isolate or remove information conveyed by phase locking, often by measuring the envelope and fine structure from the output of a set of auditory filters, and then generating stimuli in which either the envelope or fine structure are rendered uninformative or otherwise altered. One challenge for these "vocoder" approaches is that if the resulting stimuli are analyzed with a filter bank that is distinct from the one used for stimulus generation, information that was limited to one stimulus component (e.g. the fine structure) during stimulus generation can appear in a different stimulus component (e.g. the envelopes) of the analysis filter bank[97,98]. It is thus difficult to know whether a stimulus that is intended to remove a particular type of information actually succeeds in doing so once the stimulus is represented in the ears of a listener.

Another challenge for these approaches is conceptual. The signal processing distinction between envelope and fine structure is well-defined at the stage of stimulus generation, but is lost at the auditory nerve, which converts the entirety of the stimulus into a single representation of spiking activity. Degradations of phase locking thus potentially affect the encoding of both the envelope and fine structure of a stimulus. For instance, the difference in performance that we observed between models with phase locking cutoffs of 50 and 320 Hz could partially reflect degradation of what would traditionally be considered envelope cues.

From our perspective, the experimental literature manipulating envelope and fine structure may be most productively treated as a set of results that a model of the auditory system should account for. Models allow us to set the interpretation of an experimental result aside and instead ask whether a model reproduces the result, i.e. whether it behaves similarly to humans under a particular stimulus manipulation. We found such experiments to aid in distinguishing models with different phase locking cutoffs (Fig. 7c), even when the interpretation of the experimental manipulation on its own is uncertain. Our results also indicate that tasks could vary in the extent to which they require fine timing, such that the conclusions derived for one task may not generalize to others[99].

## Limitations

The most obvious limitation of our approach is that there is no guarantee that current deep optimization methods and model architectures converge on optimal task solutions. Does a model without access to phase locking fail to achieve human-like performance because precise spike-timing information is strictly necessary for the task or because the model is insufficiently optimized? In principle, alternative model architectures and/or better optimization methods could lead to more human-like models without precise spike-timing, or to models that could better exploit high-fidelity timing in the input,

leading to a larger effect of the phase locking limit than we observed. We hedged against these possibilities in two ways. First, we used multiple neural network architectures for each model class, ensuring that the reported results do not reflect the idiosyncrasies of any single network architecture. Second, our conclusions do not hinge solely on differences in absolute performance. We also compared the pattern of human and model behavior across a broad range of psychoacoustic experiments, allowing us to identify qualitative differences in how models solve tasks given different types of peripheral input. The best models achieved consistently good qualitative matches across a large set of experiments. It nonetheless remains possible that the models deviate substantially from optimality, or that the architecture class and optimization method bias the models toward one of several solutions that solve the task equally well.

Our conclusions also depend on the peripheral model stage accurately capturing the information provided by the ear. Although the model we used is well validated in nonhuman animals, we have imperfect knowledge of the properties of the human ear, and inaccuracies in the model assumptions could potentially limit the validity of the conclusions. Moreover, although the nerve representation we used is, to our knowledge, more realistic than that in previous models, practical constraints nonetheless necessitated approximations (see Methods) that could in principle affect the results.

Like ideal observers, optimized machine learning models can also outperform human observers. Some of this may be attributed to human attentional lapses during experiments. Biological systems also may have sources of noise that are absent in our models. Apart from spike sampling in the auditory nerve input (see Methods), our models were deterministic. Ideal observers often posit decision stage noise to bring model performance down to the level of humans[2,12], and the same logic and approach could be applied to machine learning models in the future.

Our approach relies on optimizing models for the "right" constraints, which by hypothesis are the tasks that are critical in daily life. We optimized models for tasks that are plausibly important in this sense, but the specific instantiations of the tasks were constrained by practical considerations. For instance, the word recognition task is limited to identifying the middle word in a speech clip, with a vocabulary that is large in absolute terms (794 words) but still substantially smaller than the vocabulary of typical humans. Similarly, our sound localization task was restricted to static sources in simplified rooms, due to constraints of the head-related transfer functions and virtual acoustic simulator used. These tasks are more realistic than those used in previous generations of models, but it is possible that the demands of tasks that are even more realistic would alter the results. We also built separate models for sound localization and speech perception (again due to practical constraints). It is possible that the demands of having to perform multiple tasks with the same auditory system could affect the results. We addressed this concern to some extent by training one model on both word and voice recognition. We found results that were qualitatively similar to those obtained from separate models optimized for each of the tasks individually, but in the one experiment that showed a large effect of the phase locking limit on word recognition, the effect of the limit was stronger in the model that was also optimized for voice recognition, suggesting that interactions between task demands can influence task strategies. Particularly given proposals that the fidelity of interaural time differences relates to difficulties hearing in noise because of its effect on spatial attention[39,86], more complicated models that concurrently localize and recognize speech could provide additional insights.

Our approach is most diagnostic when there are differences in human-model similarity across model conditions, and this is a function of the experiments used to assess behavioral similarity. When human-model behavioral similarity for a task does not vary across phase locking limits, it is difficult to exclude the possibility that other behavioral assays might show a difference. Specifically, we found little difference in human-model similarity for voice and word recognition for the three highest phase locking cutoffs. There is thus no evidence in our present results that phase locking above 320 Hz is used for speech perception, but it remains possible that some other experiment could reveal a distinction (akin to that seen for sound localization). For instance, models optimized to estimate F0 require phase locking upwards of 1000 Hz to account for human pitch perception[18], raising the possibility that speech-related experiments that assess more fine-grained use of pitch could show effects of higher phase locking cutoffs. One way to address this in the future could be to use the models to derive stimulus conditions that produce different results for different phase locking cutoffs[100], and then to test humans on these derived conditions.

## Future directions

We have treated the neural network stages of our models as a black box. Our models thus offer insight into which neural cues underlie perception but do not reveal how these cues are extracted by biological neural circuits[91,94]. In principle one could probe the tuning of units within the neural network, or relate the internal representations of different model stages to those in the brain[16,80,101,102]. The absence of realistic neural components in the models presented here limits the relevance of such analyses to hypotheses for actual neural circuits for extracting temporal information. However, future models with more biological constraints have exciting potential to make progress on these questions. For instance, one finding that at present lacks a normative explanation is the "sharpening" of phase locking that occurs in some neurons in the cochlear nucleus despite having a lower overall upper limit of phase locking[45]. Task-optimized models could help evaluate the hypothesis that this sharpening aids the extraction of information, for instance by revealing whether sharpened timing emerges in intermediate stages prior to interaural comparisons. Machine learning could also be combined with specific mechanistic proposals for how brainstem circuitry may extract task-relevant cues[11,103]. Such proposals could be built into a machine learning model as an additional stage that is either fully fixed, or that has a small number of tunable parameters. Asking if the resulting model better accounts for behavior could help test mechanistic hypotheses. The representations in such models could also be compared to brain representations[16,80], or to human EEG and ABR measures proposed as diagnostics of temporal processing[104,105].

Our approach has natural extensions for modeling sensorineural hearing loss. The healthy auditory peripheral stage in our models could be altered to simulate hair cell loss[106] and/or cochlear neuropathy[107] to reveal their effects on auditory behavior. Optimizing models with different hearing loss etiologies could yield insights into the diverse behavioral outcomes of individuals with hearing loss. We found that similarly accurate predictions of human behavior were possible with a greatly simplified cochlear model stage (through which gradients can be backpropagated). This raises the possibility of directly optimizing front-end processors[53] (or even individual sounds[54,79,100]) for perceptual outcomes in the model, which could be useful for developing hearing aids and diagnostic behavioral tests.

A similar approach could be applied to cochlear implants, by substituting simulations of electrically stimulated nerve fibers[108] for the nerve model used here. Most current cochlear implant processing strategies discard phase locking to the fine structure, but also induce a number of other differences in auditory nerve responses compared to those produced by a normal ear[109–112]. It is thus not clear how much of the difficulties experienced by cochlear implant listeners (e.g. impaired sound localization, pitch perception, and speech intelligibility in noise) are primarily due to the loss of fine structure rather than to other factors. Models optimized with different types of cochlear implant processing strategies could provide

insight into these issues, and into the potential for alternative strategies.

Another natural extension would be to investigate species differences[113]. For instance, owls are known to use phase locking well above 1000 Hz for localization[114,115]. This difference with humans could plausibly be driven by the smaller interaural time differences that result from a smaller head, potentially coupled with differences in the sounds owls and humans must localize. Models trained with head-related transfer functions from an owl, and training data generated from owl-relevant sounds, could provide insight into the pressures that give rise to such species differences. The higher F0s of animal vocalizations compared to human speech also raises the possibility that non-human animal analogues of voice recognition could utilize phase locking up to higher frequencies than we found implicated for human voice recognition, which could in principle also be investigated with our modeling framework.

The general approach of investigating neural coding features with models optimized for ecological tasks is not limited to hearing. Similar analysis of tactile perception could, for instance, elucidate the perceptual role of high-fidelity temporal coding in touch[26]. More generally, the use of machine learning to reveal the consequences of optimization under constraints has widespread potential for understanding links between biology and behavior.

## Methods

### Peripheral auditory model

The Bruce et al. (2018) auditory nerve model[46] served as the primary peripheral front-end to our artificial neural networks. This model (henceforth referred to as the "detailed" auditory nerve model) was chosen because it captures many of the complex response properties of auditory nerve fibers and has been extensively validated against electrophysiological data from nonhuman animals[106,116–120]. Stages of peripheral signal processing in the model include: a fixed middle-ear filter, a nonlinear cochlear filter bank to simulate level-dependent frequency tuning of the basilar membrane, inner and outer hair cell transduction functions, and a synaptic vesicle release/re-docking model of the synapse between inner hair cells and auditory nerve fibers. Although the model's responses have only been directly compared to recordings made in nonhuman animals, the cochlear filter bandwidths in the version used in this paper were inferred from human behavioral and otoacoustic measurements[121].

The output of the auditory nerve model was a three-dimensional array of instantaneous auditory nerve firing rates with shape [N frequency channels, T timesteps, S fiber types]. Due to computational constraints, we simulated instantaneous auditory nerve firing rates at $N = 50$ points along the cochlear frequency axis. Auditory nerve fiber characteristic frequencies were spaced uniformly on an ERB-number scale[122] between 125 and 16,000 Hz for the localization model and between 125 and 8000 Hz for the speech model. The speech model's upper characteristic frequency limit was lower because some of the training data was derived from corpora with audio sampling rates of 16,000 Hz (with a corresponding Nyquist limit of 8000 Hz), reflecting the common view that speech perception is dominated by cues below 8000 Hz. The higher upper limit of the localization model reflected the established existence of localization cues above 8000 Hz[78] (making it critical to use a high audio sampling rate, and to represent high audio frequencies, in a model of localization). The use of 50 frequency channels primarily reflects computational constraints (CPU time for simulating peripheral representations, storage costs, and GPU memory during training), but is justified in part by the smoothness of the excitation pattern produced by the ear (i.e., the excitation pattern is fairly lowpass, such that it can be represented with a modest number of samples along the cochlear axis). In previous work we found that increasing the number of frequency channels tenfold had little effect on model behavior[18]. The instantaneous firing rates within each channel were downsampled from 100 kHz (the nerve model's default sampling rate to which all audio was upsampled) to 10 kHz. The localization model operated on 1 s inputs ($T = 10,000$). The speech model operated on 2 s inputs ($T = 20,000$). At each characteristic frequency, we simulated responses of $S = 3$ different auditory nerve fiber types to represent canonical high (70 spikes/s), medium (4 spikes/s) and low (0.1 spikes/s) spontaneous rate fibers[47]. Fibers with different spontaneous rates vary systematically in their thresholds and dynamic ranges. High-spontaneous-rate fibers have the lowest thresholds but smallest dynamic ranges such that their firing rates saturate at conversational speech levels.

This array of instantaneous firing rates was then converted to an array of binomially sampled spike counts representing the population response of 32,000 individual auditory nerve fibers per ear. This spike sampling is an innovation over previous deep neural network models of audition. The number of spikes occurring at each time-frequency-fiber bin was sampled from a binomial distribution with $p$ = firing rate / sampling rate and $n$ determined by the relative numerosity of different fiber types ($n$ = fraction of fibers * 32,000 total fibers / N frequency channels). We used 60% high-, 25% medium-, and 15% low-spontaneous-rate fibers[47]. To reduce the computational cost of sampling from 1.5 million ($N \times T \times S$) independent binomial distributions per ear and stimulus, we employed a Gaussian approximation. Rather than directly sampling spike counts from $Binomial(n, p)$, we instead sampled from $Normal(np, np(1 - p))$ and rounded samples to the nearest integer, yielding an approximate sample from the desired binomial distribution. We did not attempt to model refractoriness in nerve fiber responses on the grounds that summing across fibers should minimize effects of refractoriness. To test this assumption, we generated examples of an alternative set of nerve responses in which each individual nerve fiber's firing rate was set to zero for 1 ms after each spike was sampled. This resulted in a small reduction in the overall number of spikes, but otherwise produced very similar responses (the summed spike trains used as inputs to the neural networks were highly correlated to those obtained without modeling refractoriness; $r > 0.99$).

The high sampling rate of the model auditory nerve responses was intended to ensure that the information in phase locking up to 3000 Hz could be faithfully represented (the Nyquist limit for a 10 kHz sampling rate is 5 kHz, well above the highest limit used in our models). However, the discretization of time that results from this representation causes inter-spike intervals to be quantized, which might be expected to result in some loss of information, particularly at frequencies close to the upper limit of 3000 Hz. To test whether the downsampling of firing rates to 10 kHz could have limited the benefit of high-frequency phase locking, we repeated all experiments on the models with a 3000 Hz phase locking limit, instead using an auditory nerve sampling rate of 20 kHz. To keep model sizes and architectures similar after doubling the input time dimension, we modified the first two stages of each neural network to reflect the higher sampling rate. Specifically, the kernels in the first two convolutional stages in each model had twice as many taps along the time axis and the extent of temporal pooling in the second pooling stage was doubled. We reasoned that these modifications would give the models the best chance to extract information from high-frequency phase locking without doubling the number of learnable parameters in the final fully-connected layers (which would plausibly create a confound that seemed better to avoid). Despite roughly preserving the number of learnable parameters, the GPU memory footprint of these higher-sampling-rate models is considerably larger (because the output of the convolution operation contains more activations). We were able to train the models by halving the batch sizes and training for twice as many steps, thus keeping the total number of training examples constant across the two sampling rates.

The 20 kHz sampling rate models produced extremely similar results to the default 10 kHz sampling rate models (Supplementary Figs. 12, 13, and 14). For the sound localization and voice recognition models there were no statistically significant differences between the 20 kHz and 10 kHz models in overall task performance ($p > 0.07$) or human-model similarity ($p > 0.35$). For the word recognition model, there was a very small increase in overall task performance (48.0% correct compared to the 47.3%, 47.5% and 46.8% correct for the 10 kHz sampling rate models with 320, 1000, and 3000 Hz phase locking limits, $p < 0.01$) but no increase in human-model similarity ($p = 0.29$). These results suggest the 10 kHz auditory nerve sampling rate did not contribute to the lack of benefit observed for phase locking above 1000 Hz. We also note that we compared neural network models optimized for frequency discrimination to classical ideal observers for frequency discrimination (Fig. 8). One of the neural network models in this comparison used an auditory nerve model stage with a 3000 Hz phase locking limit and a 10 kHz sampling rate, and closely matched the ideal observer that used timing information (Supplementary Fig. 12d). This latter result also suggests that the 10 kHz sampling rate does not limit the timing information that can be extracted given the 3000 Hz upper limit.

### Phase locking manipulation

The phase locking manipulation was identical to that introduced in our previous work[18]. We modified the upper frequency limit of phase locking in the auditory nerve model by adjusting the cutoff frequency of the IHC low-pass filter. In the unmodified auditory nerve model, the low-pass characteristics of the IHC membrane potential were modeled as a 7th order filter with a cutoff frequency of 3000 Hz. We set this cutoff to 3000, 1000, 320 and 50 Hz.

### Simplified cochlear model

The Bruce et al. (2018) auditory nerve model[46] is computationally expensive to run, requiring peripheral representations to be pre-computed and stored on disk rather than generated on-the-fly during neural network optimization. Simulated auditory nerve representations of the training datasets alone required 12 TB (localization task) and 26 TB (speech tasks) per phase locking condition. We repeated experiments with a simplified cochlear model hard-wired into the neural network's computation graph, eliminating the need to pre-compute the nerve representation (we note that it might eventually be possible to instead approximate detailed auditory nerve models with neural networks trained for this purpose[123]). The simplified front-end consisted of a finite-impulse-response approximation to a gammatone cochlear filter bank (with impulse responses truncated to 50 ms) followed by half-wave rectification and low-pass filtering to impose the upper limit on phase locking. Simplified cochlear models operated on 50 kHz audio for the localization task and 20 kHz audio for the speech tasks (the audio training data, which were the same as for models with the detailed cochlear stage, were upsampled to these rates for numerical convenience; they made downsampling easier on the GPU). Low-pass filtering in the simplified cochlear model was performed by convolving the rectified subbands with an impulse response measured from the Bruce et al. (2018) model's IHC filter (truncated at 50 ms and then Hanning windowed). The finite-impulse-response approximation of the IHC filter ensured the frequency-dependence of phase locking in the simplified cochlear model closely matched that of the detailed model (Supplementary Fig. 9a). After low-pass filtering, cochlear representations were downsampled to 10 kHz and passed through pointwise sigmoid functions approximating the rate-level functions of high-, medium-, and low-spontaneous-rate fibers. These sigmoid functions ranged from 0 – 250 spikes/s over a dynamic range of 20, 40, or 80 dB for high-, medium-, and low-spontaneous-rate fibers with respective thresholds of 0, 12, and 28 dB SPL. These stages yielded an array of instantaneous auditory nerve firing rates with the same

dimensions as the detailed auditory nerve model. The spike sampling procedure was identical between the simplified and detailed cochlear models. We repeated the temporal coding manipulation in the simplified cochlear model by setting the IHC low-pass filter cutoff to 3000, 1000, 320, and 50 Hz.

### Artificial neural network architectures

Simulated auditory nerve representations were passed as input to deep convolutional neural networks, each consisting of a series of feedforward layers. These layers were hierarchically organized and instantiated one of several simple operations: convolution with a linear kernel, pointwise rectification, pooling, normalization, linear transformation, dropout regularization, and softmax classification. Dropout regularization helps prevent overfitting by randomly silencing network units during training, preventing learned solutions from being overly dependent on any individual unit. Softmax classification re-scales network output representations so they can be interpreted as probability distributions over output classes (the output representation for each stimulus is a non-negative vector that sums to one).

For each task, we used 10 distinct neural network architectures previously identified in large-scale random searches over architectural hyperparameters (e.g., number of layers, units per layer, convolutional kernel size and shape, and pooling extent). The individual architectures for each task are summarized in Supplementary Tables 1 and 2. For the localization model, we used the top 10 best performing architectures from Francl and McDermott (2022). These architectures implement pooling and normalization via max pooling and batch normalization operations[19]. For the speech models, we took the best-performing architecture (modified to use Hanning-weighted average pooling[79] and layer normalization operations) from Saddler and Francl et al.[53] as a starting point. We then performed a local architecture search by making 20 new architectures via single hyperparameter modifications from the starting point (e.g., adding/subtracting one model stage, or changing the convolutional kernel shape in one layer at a time). We used the 10 best-performing networks from this local architecture search for the speech model architectures.

### Localization network architecture modification

Unlike the speech models, localization models operated on binaural input. To provide this binaural input to the models, we concatenated the auditory nerve representations from the simulated left and right ear along the last axis of the input (that was also defined by the three nerve fiber types). This resulted in a single array with shape [N frequency channels, T timesteps, 2S fiber types]. Standard convolutional model stages have three-dimensional kernels, which allow the optimizable filters to integrate information across the last feature axis (i.e., between the ears in this case). Our default neural network architectures imposed no restriction on where in the model processing hierarchy interaural cues could be extracted. To test the effect of delaying interaural integration as happens in biological auditory systems, we replaced standard convolution operations in the earliest model stages with "grouped" convolutions. Grouped convolutions split their input representation along the feature axis and use a separate convolutional kernel filter for each group[124]. Setting the number of groups to 2 in the first convolutional layer separated the input for the left and right ear. Successive convolutional stages with 2 groups maintain separate monaural processing streams. Interaural integration then occurs at the first convolutional stage where the number of groups is set to 1 (i.e., standard convolution). To delay interaural integration in our networks until after significant temporal pooling had occurred, we set the number of groups to 2 for all convolutional stages prior to the point at which the representation was downsampled by a factor of at least 4 relative to the nerve model stage output (from 10 kHz to no greater than 2.5 kHz). The convolutional stages replaced

with grouped convolutions in the delayed interaural integration models are highlighted in Supplementary Table 1.

## Model optimization – overview

Artificial neural networks were optimized to perform real-world hearing tasks operationalized as classification tasks. The training datasets and individual tasks are described in the subsequent sections. In general, training stimuli were labelled with a class (one label per task) and neural network parameters were iteratively updated to minimize the softmax cross-entropy loss function via stochastic gradient descent (ADAM optimizer) with gradients computed via backpropagation. Localization models trained for 200,000 steps with a batch size of 32 and learning rate of 0.0001. Word and voice recognition models trained for 400,000 steps with a batch size of 32 and learning rate of 0.00001. The learning rates were arrived at empirically as those that worked well for the task and architectures. Classification performance on held-out validation sets was recorded after every 10,000 training steps. The neural network parameters producing the highest validation set performance during the training routine were used for the trained model. The number of steps in each model's training routine was chosen to obtain a plateau in validation set performance under all phase locking conditions. Model training times varied by architecture, but each model could be trained in 96 h on a single NVIDIA A100 GPU on the MIT OpenMind Computing Cluster. Localization models typically trained in under 48 h.

## Model optimization – sound localization

We used the sound localization task of Francl and McDermott[19] in which models classified noisy 1 s auditory scenes according to the azimuth and elevation of a target natural sound. The source location classes spanned 360° in azimuth (5° bin width) and 0 to 60° in elevation (10° bin width), yielding a total of 504 output classes (72 azimuth × 7 elevation classes). To ensure that the task was well-defined, the training scenes always consisted of a single natural sound rendered at one target location superimposed with real-world noise textures diffusely localized at 3 – 12 different distractor locations. Target sounds were taken from the Glasgow Isolated Sound Events[125] (GISE-51) subset of Freesound Dataset 50k[126] (FSD50K), which consists of variable-length recordings of individual sources spanning 51 categories of everyday sounds. We only used source clips for which the original 44.1 kHz sampling rate audio could be found in FSD50K (to ensure that spectral localization cues could be rendered faithfully). Our training and validation datasets were generated from 12465 and 1716 unique source clips, respectively. This was a substantial increase in target sources compared to the previous modeling work of Francl and McDermott[19], with the goal of increasing the robustness of the resulting model. For model evaluation and human experiments, we used 460 sounds from the GISE-51 evaluation set equally distributed across 46 sound categories (discarding 5 categories with fewer than 10 evaluation clips).

Texture-like background noise was sourced from a subset of the Audioset[127] corpus screened to remove nonstationary sounds (e.g., speech or music). The screening procedure involved measuring auditory texture statistics[81] (envelope means, correlations, and modulation power in and across cochlear frequency channels) from all recordings, and discarding segments over which these statistics were not stable in time, as in previous studies[128,129]. The screening procedure yielded 26515 and 562 unique 10 s noise clips for the training and validation datasets, respectively. Auditory scenes for the training and validation data were constructed by combining randomly sampled pairs of target sounds and texture-like noise samples (sliced into 1 s segments).

As in previous work[19], we augmented the number of unique target waveforms by applying a randomly generated band-pass filter to the target in 50% of training and validation examples. Band-pass filter center frequencies were sampled log-uniformly between 160 and 16,000 Hz. Bandwidths were sampled log-uniformly between 2 and 4 octaves and the filter order was drawn uniformly between 1 and 4. Individual target and noise sources were first spatialized and then summed together at SNRs uniformly drawn between -15 and +25 dB, except for 5% of scenes which included no noise. For all localization experiments, SNR referred to the target sound's level relative to the sum of all background noise sources.

To spatialize scenes, we used a virtual acoustic room simulator[130] to render sets of binaural room impulses responses (BRIRs) for a KEMAR in 2000 unique listener environments. The simulator used the image-source method and incorporated KEMAR's HRTFs[131]. We randomly generated 2000 unique listener environments by sampling different shoebox rooms (varying in size and wall materials) and listener positions (x, y, z coordinates and head angle) within each room. Room lengths and widths were sampled log-uniformly between 3 and 30 m and room heights were sampled log-uniformly between 2.2 and 10 m. The listener's head position was sampled uniformly in each room, subject to the constraint that the head was at least 1.45 m from every wall and no higher than 2 m from the floor. For each listener environment, we rendered BRIRs at 1008 source locations (2 distances from the listener × 72 azimuths × 7 elevations). One of the distances was 1.4 m for every BRIR. The other distance was independently sampled for each BRIR (drawn uniformly between 1 m and 0.1 m less than the distance from the listener to a wall). 1800 unique listener environments were included in the training set and the remaining 200 were used for validation. This was a substantial increase over the previous model by Francl and McDermott[19], whose data were generated from only 5 rooms. The final training and validation datasets consisted of 1,814,400 and 201,600 binaural auditory scenes, respectively. Target natural sounds were placed once at each of the 2000 × 1008 source locations to ensure the dataset was balanced across the 504 target location classes. Auditory scenes were presented to the model during training at sound levels drawn uniformly between 30 and 90 dB SPL.

## Model optimization – word and voice recognition

The same dataset was used for the word and voice recognition training tasks. We used an augmented version of the Word-Speaker-Noise dataset[79], which consists of 230,356 unique speech excerpts embedded in 718,625 unique nonspeech background noise excerpts from Audioset[127]. Randomly sampled pairs of 2 s speech and noise excerpts were combined to yield a training dataset of 5.8 million examples. A validation set of 370,000 examples was similarly constructed from speech and noise excerpts excluded from training. Each example was labeled with the talker that produced the speech utterance and the word that appeared in the middle of the utterance (i.e., that overlapped the 1 s mark of the 2 s utterance). The datasets contained 433 unique talker labels and 794 unique word labels.

We chose this closed-set word recognition task in order to facilitate supervised learning with a human-relevant task. We assume that words constitute one of the output representations of human speech recognition, and so are a good choice of model output representation given the desire to optimize for biologically relevant tasks. However, due to Zipf's law, it is difficult to obtain large numbers of examples of infrequently occurring words. As a result, if one includes most English words it is challenging to generate training sets that are balanced across word labels (with similar numbers of examples of each class, as is advantageous for supervised learning). The main alternative at present would be to build a model using methods from contemporary machine speech recognition, which typically involves training systems to map audio to characters (with subsequent stages to derive word labels from character strings). Given that character strings seem a poor candidate for the output representation of human speech recognition, we opted to instead use a word recognition task with a vocabulary size for which we could assemble a balanced training set.

Training models for voice recognition is complicated by the fact that large speech corpora are often crowd-sourced online, with individuals contributing recordings of themselves reading passages or responding to prompts. Models optimized for talker classification using such corpora may pick up on non-voice cues that predict these labels (e.g., characteristics of the recording device or environment). To help ensure that models learned robust voice representations, we applied a set of randomly sampled audio manipulations to the speech to approximate the variable conditions in which human listeners encounter the same voice. In 25% of the dataset, speech excerpts were augmented to increase natural voice variability by applying small F0 ($\pm 0.5$ semitones) and tempo shifts ($\pm 20\%$) or simulating whispering (<0.5% of examples) via the STRAIGHT algorithm[132]. In an independently drawn 25%, we applied commonly encountered audio distortions like band-pass / equalization filters, lossy audio compression / transmission, and dynamic range companding. In another independently drawn 5%, we replaced background noise with speech babble (between 12 and 36 talkers, uniformly sampled) to give the model some exposure to multi-talker situations. SNRs for the augmented speech excerpts were drawn uniformly between -10 and +10 dB and sound levels were drawn uniformly between 30 and 90 dB SPL.

We jointly optimized individual models to recognize both voices and words using this augmented dataset. Dual-task optimization was accomplished with a separate output layer for each task. All preceding network stages were shared between the two tasks and parameters were updated to minimize the sum of the softmax cross entropy loss from both tasks.

## Model evaluation – overview
For each task, we first evaluated model behavior in naturalistic conditions. Wherever possible, we tested humans on the same naturalistic tasks using the same stimuli. For each task, we then tested models on stimulus manipulations from the psychoacoustics literature and compared behavior to human results. Human-model behavioral similarity was quantified for each experiment and model by measuring Pearson correlation coefficients and root-mean-squared error between analogous human and model data points. In each experiment, we present the results averaged across the 10 model architectures.

## Human behavioral experiments – informed consent
All participants provided informed consent and the Massachusetts Institute of Technology Committee on the Use of Humans as Experimental Subjects (COUHES) approved all experiments.

## Localization model evaluation – sound localization in noise
**Human experiment.** We measured the ability of human listeners to localize natural sounds in noise using a 19-by-5 array of loudspeakers arranged on a hemisphere with 2 m radius. The array spanned 180° in azimuth (frontal hemifield) and 0°–40° in elevation (10° spacing in both azimuth and elevation) relative to the listener's head at the center. 11 normal-hearing listeners (5 female) with ages between 21 and 30 years each performed 460 trials with 460 unique target natural sounds from the GISE-51 evaluation dataset. On each trial a target natural sound was played from one of the 95 loudspeakers while threshold-equalizing noise[58] was played concurrently from 9 distinct loudspeakers. Target and noise locations were randomly sampled on each trial. The listener's task was to report which loudspeaker produced the target by entering the loudspeaker's label on a keypad. Listeners were instructed to direct their head at the loudspeaker directly in front of them for the duration of the stimulus. Once the stimulus ended, they could look at the loudspeaker where they thought the target had occurred to obtain the label. Listeners then redirected their head toward the front loudspeaker prior to the start of the next trial. Experimenter observation indicated that participants were highly compliant with these instructions. Target sounds were

presented at 60 dB SPL (A-weighted) and noise levels were determined such that the SNR of the target relative to the sum of the 9 noise sources was -13.6, -6.8, 0, +6.8, or +13.6 dB. All stimuli were sampled at 44.1 kHz and were 1 s in duration, including 15 ms onset and offset ramps (Hanning window).

**Model experiment.** Models were tested on all combinations of the 460 target natural sounds, 5 SNRs, and 95 target locations (218,500 total stimuli) used in the human experiments. Sources were spatialized in a virtual rendering of the loudspeaker array room human listeners were evaluated in. To match the task between human and models, we restricted model localization judgments to azimuth and elevations corresponding to the 95 loudspeaker locations.

**Human-model comparison.** Human and model performance was quantified by measuring mean absolute spherical error (great circle distance), azimuth error, and elevation between the true and reported target sound location (plotted in Fig. 3d). Human-model similarity scores were the correlation (or root-mean-squared difference) between these human and model error metrics as a function of SNR.

## Localization model evaluation – psychoacoustics
We simulated an expanded version of the battery of localization experiments used in Francl and McDermott[19]. 6 of the 8 psychoacoustic experiments we used were identical to those in Francl and McDermott, using the same stimuli and analyses. The minimum audible angle and ITD lateralization experiments were the two additions, included because they seemed potentially relevant to the use of phase locking. All psychoacoustic stimuli for model localization experiments were sampled at 44.1 kHz.

## ITD / ILD cue weighting
**Human experiment.** We simulated the experiment of Macpherson and Middlebrooks (2002), which measured shifts in perceived azimuth for virtual sounds with additional ITDs and ILDs imposed[55]. In the original experiment, 13 participants (5 female) were played sounds over headphones and reported perceived azimuth by turning their head to face the virtual source. The experiment took place in an anechoic chamber and used both low-pass (0.5–2 kHz) and high-pass (4–16 kHz) 100 ms noise bursts with 1 ms squared-cosine ramps at the onset and offset.

**Model experiment.** We used identical stimuli spatialized in a virtual anechoic room at 0° elevation and 0° to 360° azimuth in steps of 5°. For each of the source locations and noise bands, we also separately created ITD- and ILD-biased versions of the stimuli. ITD-biased versions were generated by imposing additional $\pm 300\,\mu s$ and $\pm 600\,\mu s$ time delays between the two ears. ILD-biased versions analogously imposed additional $\pm 10$ and $\pm 20$ dB level differences between the two ears. We collected model azimuth predictions for each stimulus. Azimuth predictions in the rear hemifield were mapped to the frontal hemifield by reflecting across the coronal plane. We compared the model azimuth prediction for each ITD- and ILD-biased stimulus ("the biased azimuth") to the azimuth prediction for the corresponding unbiased stimulus ("the unbiased azimuth"), computing shifts in the biased azimuth relative to the unbiased azimuth. Azimuth shifts for ITD-biased stimuli were expressed in $\mu s$ by subtracting the ITD of a real source at the biased azimuth from the ITD of a real source at the unbiased azimuth. Azimuth shifts for ILD-based stimuli were expressed in dB by subtracting the ILD of a real source at the biased azimuth from the ILD of a real source at the unbiased azimuth. Expressing azimuth shifts in cue units enables calculation of a dimensionless perceptual weight by dividing the azimuth shift by the imposed cue amount. Separate ITD and ILD perceptual weights were computed for low-pass and high-pass noise by averaging across all azimuths and bias

magnitudes. For instance, an ITD perceptual weight of 1 indicates that, for a given virtual stimulus, an additional ITD of $\tau$ shifts the perceived azimuth by an angle corresponding to an ITD change of $\tau$ between two real source locations. A perceptual weight of 0 indicates that imposing additional ITDs or ILDs has no effect on the perceived azimuth. These weights are plotted for each frequency range and cue type in Fig. 3f.

**Human-model comparison.** Human-model similarity was quantified by comparing ITD and ILD perceptual weights measured for low-pass and high-pass noise between humans and models.

### Minimum audible angle vs. azimuth

**Human experiment.** Mills (1958) measured human localization acuity as a function of frequency and azimuth by playing pure tones to a blindfolded listener from a rotating boom in an anechoic chamber. Minimum audible angle thresholds were defined as the smallest change in azimuth required for the listener to discriminate whether a tone's location shifted left or right between two presentations. The key result was that localization acuity was best near the midline and degraded steadily towards the periphery. Mills (1958) only reported thresholds from a single human listener[68], but the general result is well-established and holds across different experimental paradigms[67].

**Model experiment.** We measured model thresholds by simulating a left/right lateralization experiment. Pure tones (1 s duration including 70 ms half-Hanning windows at onset and offset) were spatialized in a virtual anechoic room at 0° elevation and azimuths of -90° to +90° in steps of 0.5° (using linear interpolations of BRIRs spaced 5° apart). For each tone, we collected the model's probability distribution over locations. These distributions were then multiplied by a mask assigning zero probability to nonzero elevations and azimuths outside the frontal hemifield (intended to replicate a human participant's knowledge that the tones to be discriminated were at this subset of all possible locations). This resulted in probability distributions over predicted azimuth in the frontal hemifield for each stimulus. Left/right discrimination trials were simulated by comparing the expected value of each stimulus location under the model's output distributions for pairs of stimuli rendered at different azimuths. We used the expected value rather than the arg-max because the expected value provided a fine-grained location estimate that allowed for more precise discrimination thresholds. Trials in which the signed predicted azimuth of the second tone was larger than the signed predicted azimuth of the first were counted as rightward judgments. Minimum audible angle thresholds for different frequencies (250, 500, 750 and 1000 Hz) and reference azimuths (0°–75° in steps of 5°) were inferred from psychometric functions (proportion of rightward shifts as a function of azimuth difference) constructed from all possible trials within ±10° azimuth of the reference for each frequency. Model minimum audible angle thresholds were defined as the azimuth difference that yielded 70.7% rightward shifts (calculated by fitting Normal cumulative distribution functions to the psychometric functions).

**Human-model comparison.** We averaged human and model thresholds across pure tone frequencies of 250, 500, 750, and 1000 Hz to yield a single results plot of accuracy vs. azimuth (plotted in Fig. 3h). Human-model similarity was quantified by comparing average model thresholds with linearly interpolated human thresholds as a function of absolute azimuth between 0° and 75°.

### ITD lateralization vs. frequency

**Human experiment.** The upper frequency limit of fine structure ITD sensitivity in humans has classically been measured by asking listeners to make left/right lateralization judgments with pure tones presented over headphones. The pure tones have identical envelopes (by using a fixed window to eliminate onset ITDs) and zero ILD (identical amplitude between the two ears). Listeners hear pairs of tones with different ITDs and judge whether the second tone sounded left or right of the first. The ΔITD threshold is the smallest change in ITD between two tones that a listener can reliably discriminate in this way. We simulated the experiment of Brughera et al. (2013), who measured ΔITD thresholds of 4 young adult listeners (1 female) with 250, 500, 700, 800, 900, 1000, 1200, 1250, 1300, 1350, and 1400 Hz pure tones[44].

**Model experiment.** We simulated the experiment on models by measuring probability distributions over locations from models tested on 500 ms pure tone stimuli (including 100 ms linear onset and offset ramps). Fine structure ITDs ranged from -160 μs to 160 μs in steps of 1 μs. Frequencies ranged from 50 – 4000 Hz in steps of 50 Hz. ΔITD thresholds were inferred from model judgments using the same method as for the minimum audible angle experiment. Model predictions were compared for pairs of stimuli with different ITD, and rightward judgments were assigned to trials for which the signed azimuth prediction was larger for the second tone than the first. Psychometric functions were constructed for each frequency (proportion of rightward shifts as a function of ΔITD) and the ΔITD threshold was defined as the difference in azimuth yielding 70.7% rightward shifts. These thresholds are plotted vs. frequency in Fig. 4b and d.

**Human-model comparison.** Human-model similarity was quantified by comparing log-transformed model thresholds with linearly interpolated human thresholds as a function of frequency between 250 and 1500 Hz.

### Effect of changing ears

**Human experiment.** We simulated a change in our models' pinnae intended to be analogous to the manipulation of Hofman et al.[57]. In the original experiment, 4 participants localized white noise bursts presented in a 4-by-4 grid uniformly tiling ±20° in azimuth and ±20° elevation. Participants reported perceived locations by making eye movements to the source. After collecting baseline azimuth and elevation judgments, plastic molds were inserted in the participants ears, which altered the direction-specific filtering of their pinnae (Supplementary Fig. 3a). Participants then repeated the localization task with modified pinnae[57].

**Model experiment.** We simulated the experiment by collecting baseline model azimuth and elevation judgments with the same stimuli used in the original human experiment (500 ms noise bursts with a frequency band of 0.2 to 2 kHz) and then switching out the KEMAR HRTFs the model was trained with for 45 different sets of HRTFs from the CIPIC dataset[133]. We note the use of different real HRTFs plausibly involves less drastic changes than produced by inserting molds into the ears. Model azimuth and elevation predictions were collected for stimuli spatialized on 4-by-4 grid uniformly tiling ±30° in azimuth and 0°–30° in elevation (Supplementary Fig. 3b). Azimuths and elevations were not matched exactly to the human experiments due to constraints of the available HRTFs. We averaged model judgments across the 45 different sets of HRTFs not used to train the model to compare against human judgments with modified pinnae.

**Human-model comparison.** To summarize the effects of changing pinnae on azimuth and elevation accuracy, we computed changes in mean absolute azimuth and elevation error with the untrained pinnae relative to the trained pinnae, averaging across all 16 source locations, yielding the graph of Supplementary Fig. 3c. Human-model similarity was quantified by comparing absolute azimuth and elevation errors as a function of grid position and ear condition (as shown in Supplementary Fig. 3a, b) between humans and models.

## Effect of smoothing spectral cues

**Human experiment.** We simulated a modified version of the experiment originally conducted by Kulkarni and Colburn (1998), which measured the effect of HRTF spectral details on sound localization[56]. In the original experiment, 4 listeners were played white noise bursts in an anechoic chamber. Sounds were presented from either a physical loudspeaker in the room or virtually over open-backed headphones. The virtual sounds were spatially rendered at the loudspeaker's location using the participant's own HRTFs. Participants were tasked with reporting whether the sound came from the loudspeaker or the headphones. When the participants' full HRTFs were used, performance was at chance (50%). As the spectral details of the HRTFs were smoothed out by approximating the HRTF's discrete cosine transform with progressively fewer cosines (Supplementary Fig. 3d), performance rose above chance as participants no longer perceived the virtual stimuli at the loudspeaker's location.

**Model experiment.** We applied the same smoothing manipulation to the KEMAR HRTFs (by retaining only the first 256, 128, 64, 32, 16, 8, 2, or 1 coefficients of the HRTF's discrete cosine transform) and evaluated model performance in a virtual anechoic room using 1 s broadband (0.2–20 kHz) noise bursts. Model localization judgments were collected for each smoothing condition at 413 locations spanning 0°–60° in elevation and 0°–360° in azimuth (spacing determined by the locations of the measured KEMAR HRTFs). We computed mean absolute azimuth, elevation, and spherical errors as a function of the number of cosines used to approximate the HRTFs (plotted in Supplementary Figs. 2f and 3e).

**Human-model comparison.** Reasoning that higher absolute localization errors in the model would correspond to better performance on the human real/virtual discrimination experiment, we quantified human-model similarity by comparing model absolute spherical error and human percent correct scores as a function of the smoothing parameter (Supplementary Fig. 2f).

## Precedence effect

**Human experiment.** We simulated an experiment originally conducted by Litovsky and Godar (2010), which measured localization accuracy for 25 ms (including 2 ms cosine onset and offset ramps) pink noise bursts played at two different locations[134]. The bursts were played from two loudspeakers in an array spanning −60° to +60° in azimuth (20° spacing, 0° elevation) and were delayed relative to one another by 5, 10, 25, 50, or 100 ms. The lag click was always presented at 0° azimuth and the lead click was presented variably at one of six azimuths (±20°, ±40°, ±60°) on each trial. 10 listeners (all female) with ages between 19 and 26 were tasked with reporting whether they heard one or two sounds as well as the loudspeaker that produced each sound. Root-mean-squared azimuth errors were calculated separately for the lead and lag noise burst and reported as a function of the delay between the lead and lag bursts.

**Model experiment.** We evaluated models on the same stimuli rendered in a virtual anechoic room. Our models always reported a single location which we used to compute the root-mean-squared azimuth error relative to both the lead and lag burst. The "precedence effect" refers to several different phenomena that occur when two sounds are played in close succession from different locations[135]. The model judgments can reflect one of these (the localization dominance of the leading sound), but because the models cannot report the presence of more than one sound source location, they cannot explicitly exhibit one of the other main precedence phenomena (the perception of two distinct sources when the delay between leading and lagging clicks is large). Human and model results are plotted in Supplementary Fig. 2g.

**Human-model comparison.** Human-model similarity was quantified by comparing human and model azimuth error for both the lead and lag burst as a function of the inter-burst delay.

## Bandwidth dependence of localization

**Human experiment.** We simulated the experiment of Yost and Zhong (2014), which measured the effect of stimulus bandwidth on localization accuracy with an array of 8 loudspeakers positioned between -15° and +90° in azimuth (15° spacing) relative to the midline[77]. 33 participants (26 female) with ages between 18 and 36 were tasked with reporting which loudspeaker produced a 200 ms (including 20 ms squared cosine onset and offset ramps) sound. Stimuli were pure tones or band-pass filtered white noise bursts with bandwidths of 1/20, 1/10, 1/6, 1/3, 1, and 2 octaves. Pure tone and center frequencies were set to 250, 2000, and 4000 Hz. Human listeners made 20 localization judgments per bandwidth, center frequency, and loudspeaker position.

**Model experiment.** We evaluated our models on simulations of the stimuli from the original human experiment, rendering sounds in a virtual anechoic room at azimuths of −90° to +90° in steps of 5°. Model localization judgments were restricted to the frontal hemifield and 0° elevation. Human and model results are plotted in Supplementary Fig. 2h.

**Human-model comparison.** Human-model similarity was quantified by comparing human and model root-mean-squared error as a function of bandwidth (averaged across center frequencies).

## Median plane spectral cues

**Human experiment.** We simulated a modified version of the experiment by Hebrank and Wright (1974), which measured the accuracy of human elevation judgments as a function of the frequency content of noise bursts[78]. In the original experiment, 10 participants were played 1 s noise bursts from a vertical array of loudspeakers along the median plane spanning −30° to +210° in elevation with 30° spacing (0° is frontal). The experiment took place in an anechoic chamber and participants were tasked with reporting which loudspeaker produced the noise burst. Noise bursts were either low-pass or high-pass with varying cutoff frequencies: 3.9, 6.0, 8.0, 10.3, 12.0, 14.5 or 16.0 kHz for the low-pass noise and 3.8, 5.8, 7.5, 10.0, 13.2 or 15.3 kHz for the high-pass noise.

**Model experiment.** We evaluated our model on noise bursts with the same cutoff frequencies rendered in a virtual anechoic room at elevations of 0°, 30°, 60°, 120°, 150°, and 180° along the median plane. To match the task between human and models, we restricted model localization judgments to azimuth and elevations along the median plane (model localization judgments corresponded to the highest softmax probability location class with azimuth 0° or 180°). Human and model results are plotted in Supplementary Fig. 2i.

**Human-model comparison.** Human-model similarity was quantified by comparing human and model percent correct scores as a function of noise type and frequency cutoff.

## Speech model evaluation – word and voice recognition as a function of SNR and noise type

**Human word recognition experiment.** We measured human word recognition accuracy as a function of SNR for four different types of background noise[16] using an evaluation set of 376 unique speech excerpts (held out from the model training and validation sets). The experiment was a replication of an experiment by Kell et al.[16], modified to use words in the vocabulary of the models presented in this paper. The four types of noise were "auditory scenes", speech babble, instrumental music, and speech-shaped noise. The experiment also

included a fifth noise condition which was not analyzed here. For the first three conditions, background noise excerpts (376 per condition) were sourced from IEEE AASP CASA Challenge[136] (auditory scenes), CommonVoice[137] (8-talker speech babble), and MUSDB18[138] (instrumental music). The babble was generated by summing speech excerpts from 8 different talkers. Speech-shaped noise was synthesized for each evaluation speech clip by imposing the power spectrum of the same speech clip on white noise. Speech excerpts were combined with background noise from each condition at 6 SNRs (noiseless and -9, -6, −3, 0, +3 dB) yielding 9024 possible stimuli (376 speech excerpts × 4 noise type × 6 SNRs). Stimuli were 2 s in duration and sampled at 20 kHz. Individual participants heard only 376 stimuli (each unique speech excerpt was presented once), uniformly sampled across SNRs and noise types. To match our model word recognition task, participants were asked to report which word (from a list of 793) appeared in the middle of the utterance (defined as overlapping the 1 s mark). Participants typed responses into a textbox and, as they typed, the displayed list of 793 words was filtered to include only words that matched the entered string. Only responses from the word list could be submitted. The experiment was run online and included 44 participants (13 female, 30 male, 1 nonbinary) who self-reported normal hearing, passed a headphone check[139], completed at least 100 trials (some participants experienced connectivity issues that prevented them from completing the entire experiment, but because trials were randomly ordered their data could be included without compromising the analysis), and responded correctly to at least 85% of catch trials intended to make sure they were paying attention to the experiment (isolated words presented in silence). Here and in all other online experiments, participants heard a calibration sound and adjusted the presentation level to be comfortable prior to the start of the experiment. Participants ages were between 18 and 62 (median 36) years. We did not run an analogous voice recognition experiment as there was no way to test humans and models in the same way (because every human is familiar with an idiosyncratic set of voices).

**Model word and voice recognition experiment.** We measured model word and voice recognition accuracy as a function of SNR and noise type using the same stimuli as for the human experiment. Each of the 376 speech excerpts had a word and voice label included in the training dataset (376 unique words from 164 talkers). For the model experiment, the speech level was fixed at 60 dB SPL and noise levels were adjusted to give the desired SNRs. Models were evaluated on the full evaluation set (9024 stimuli). Results are plotted in Fig. 5b, c.

**Human-model comparison.** We quantified human-model similarity by comparing human and model word recognition accuracy as a function of SNR and noise type.

**Speech model evaluation – word and voice recognition in naturalistic auditory textures**

**Human word recognition experiment.** To probe human speech-in-noise recognition at a larger scale (to obtain a stronger model comparison test), we measured human word recognition accuracy in 43 different naturalistic auditory textures. The 376 speech excerpts from the evaluation set were embedded in 376 unique exemplars of each auditory texture. The 2 s texture exemplars were previously generated[19] to match the statistics of 43 recorded real-world textures and the success of the iterative synthesis algorithm[81] was determined both subjectively (synthesized exemplars sounded like the recorded textures) and objectively (mean-squared errors between synthetic and original texture statistics were at least 40 dB below the mean-squared texture statistics of the original recordings). The experiment was identical to the previous word recognition experiment, except that excerpts were randomly assigned to one of the 43 texture conditions and the SNR was fixed at −3 dB. The online experiment included

47 participants (24 female, 23 male) who self-reported normal hearing, passed a headphone check[139], completed at least 100 trials, and responded correctly to at least 85% of catch trials (isolated words presented in silence). Participant ages were between 23 and 59 (median 39) years.

**Model word and voice recognition experiment.** We measured model word and voice recognition accuracy for speech embedded in each of 43 auditory textures, using the same stimuli as for the human experiment. Models were evaluated on the full evaluation set (16,168 stimuli = 376 speech excerpts × 43 auditory textures). Human and model results are plotted in Fig. 5e, f.

**Human-model comparison.** We quantified human-model similarity by comparing human and model word recognition accuracy as a function of the 43 auditory textures. The noise-corrected explained variance was calculated by dividing the human-model Pearson $r^2$ by the product of the human and model split-half reliabilities (after Spearman-Brown correction).

**Word and voice recognition with F0-altered speech**

**Human voice recognition experiment.** We ran a modified version of the voice recognition experiment of McPherson and McDermott (2018), which measured human listeners' ability to recognize F0-altered voices from famous celebrities[85]. Stimuli were 4 s speech excerpts from 37 recognizable politicians, actors, singers, and television hosts. In the first block of 37 trials, each participant heard all 37 voices randomly assigned to one of 8 F0-manipulation conditions (inharmonic or shifted ±12, ±6, ±3, 0 semitones from the original F0). In the second block of 37 trials, each participant heard different excerpts of the same 37 voices with no F0 shift. Each participant's results were analyzed only for first the block, limited to just the celebrity voices they successfully recognized in the second block and identified as familiar in a pre-experiment survey (an attempt to only measure performance for familiar voices, to make for a fairer comparison with the models). All stimuli were resynthesized with the STRAIGHT algorithm[132]. Voices were made inharmonic by shifting harmonic frequency components above the fundamental by random amounts uniformly sampled between −50% and +50% of F0[85,87]. Jitter values were sampled independently for each harmonic frequency and voice clip but were constrained (via rejection sampling) such that adjacent harmonics were always separated by at least 30 Hz. The experiment was a 100-alternative forced-choice task. Participants entered responses into a textbox which filtered a displayed list of 100 celebrity names and descriptors (e.g., "Dolly Parton (country singer-songwriter)") until there was only a single match, which the participant could submit. This procedure deviated slightly from the original McPherson and McDermott experiment, which was open set, and required the experimenter to then score participant's text responses by hand. The procedure adopted here standardized responses while maintaining some of the benefits of an open set experiment (minimizing the chance that participants would artifactually boost their scores by using a process of elimination with the list of possible voice choices). The experiment included 112 participants (46 female, 65 male, 1 nonbinary) who self-reported normal hearing and passed a headphone check[139]. Participant ages were between 20 and 73 (median 39) years. Because analysis was limited to voices for which participants demonstrated familiarity and each voice could only be assigned to one F0 condition, the number of participants for each condition ranged from 87–95.

**Human word recognition experiment.** We measured human word recognition accuracy for the same 8 F0 conditions using the 376 model evaluation set speech excerpts. The experiment was identical to the word recognition in noise experiments, except that excerpts were

randomly assigned to one of 8 F0 conditions (synthesized with the same procedure used for the voice recognition experiment) and presented in quiet. The online experiment included 22 participants (8 female, 14 male) who self-reported normal hearing, passed a headphone check[139], completed at least 100 trials, and responded correctly to at least 85% of catch trials (isolated and unaltered words presented in silence). Participant ages were between 25 and 70 (median 38) years.

**Model word and voice recognition experiment.** We measured model word and voice recognition accuracy on the F0-manipulated evaluation set used in the human word recognition experiment. We collected model word and voice predictions for the 376 speech excerpts in each of 10 F0 conditions (inharmonic or shifted ±12, ±9, ±6, ±3, 0 semitones from the original F0). Human and model results are plotted in Fig. 6b, c.

**Human-model comparison.** Human-model similarity across F0 conditions was quantified with separate correlation coefficients (or root-mean-squared error) for word and voice recognition. We compared human and model word or voice recognition scores as a function of the 8 shared F0 conditions.

## Word recognition with tone-vocoded speech

**Human experiment.** We simulated an experiment originally conducted in humans by Hopkins and Moore (2009), which measured speech reception thresholds in stationary and modulated noise using progressively tone-vocoded speech[38]. In the original experiment, speech stimuli were split into frequency subbands with a 32-channel cochlear band-pass filter bank with center frequencies equally spaced on an ERB-number scale[122] between 100 and 10,000 Hz. Frequency channels above a set cutoff channel (which determined the "number of channels with intact TFS") were tone vocoded, intended to disrupt temporal fine structure. Channels were tone vocoded by imposing the temporal envelope (absolute value of the Hilbert transform) of the original speech subband on a pure tone carrier at the channel's center frequency. Tone-vocoded subbands were band-pass filtered using the corresponding filter from the cochlear filter bank and summed together with the remaining unmodified subbands. The resulting stimuli were presented to listeners in both stationary and modulated speech-shaped noise. Modulated noise was amplitude-modulated with an 8 Hz sinusoid on a decibel scale with a peak-to-valley ratio of 30 dB. Human speech reception thresholds were measured from 10 normal hearing participants using an adaptive procedure that tracked the SNR needed to achieve 50% of words correct. Hopkins and Moore (2009) reported speech reception thresholds with the cutoff channel set to 0, 8, 16, 24, and 32.

**Model experiment.** We applied the same stimulus manipulation to our 376 evaluation set speech excerpts and measured model word recognition accuracy for speech in stationary and modulated noise at SNRs of −15 to +15 dB in increments of 3 dB. We used the speech-shaped noise from the word recognition experiment described above (and shown in Fig. 5b). Amplitude-modulated noise was generated by applying the same 8 Hz sinusoidal envelope used in the human experiment to the speech-shaped noise. Speech reception thresholds were calculated for the model by fitting a sigmoid to the psychometric function (word recognition accuracy as a function of SNR) for each condition and selecting the SNR that yielded half-maximal performance. We measured model speech reception thresholds with the cutoff channel set between 0 (all channels tone vocoded) and 32 (all channels intact) in steps of 4. Because our models were trained with speech sampled at 20 kHz, the 32-channel Gammatone filter bank used to synthesize model stimuli had center frequencies equally spaced on an ERB-number scale between 80 and 8000 Hz rather than 100 to 10,000 Hz.

**Human-model comparison.** As in the original analysis by Hopkins and Moore, human and model speech reception thresholds for both noise types were expressed relative to that for speech with all channels vocoded (i.e., subtracted from the threshold with cutoff channel set to 0). Human-model similarity was quantified by comparing this "benefit from TFS" as a function of SNR and noise type between humans and models.

## Speech localization in noise and reverberation

The purpose of this experiment was to evaluate the models' performance in another setting in which phase locking has been proposed to influence speech recognition—that in which spatial attention is used to select a target talker amongst distractor talkers[39,86]. We currently lack models that can perform attentional selection tasks, such that it was not possible to conduct a model version of the published human experiments in this domain. Instead, we measured the effect of phase locking on the localization of speech, based on the logic that impaired localization of speech would translate to impaired spatial attention in cocktail party settings. We specifically tested conditions with noise and reverberation that have been found to produce individual differences in behavior that might be related to individual differences in the integrity of temporal coding in the auditory periphery.

**Model experiment.** Localization models were evaluated on the 376 speech excerpts from the evaluation set (nerve representations truncated to 1 s) spatialized in both a virtual anechoic room and a virtual reverberant room. The reverberant (RT60 = 1 s) room was a rendering of our loudspeaker array room from the localization-in-noise experiment described above (and shown in Fig. 3c, d). For the anechoic room, we changed all simulated room materials to be perfectly absorptive. Each speech clip was assigned to one of 9 loudspeaker locations (spanning −80° to +80° azimuth in steps of 20°) 2 m from the simulated listener. On each trial, threshold-equalizing noise[58] was played from the remaining 8 loudspeaker locations. The speech level was held constant at 60 dB SPL, and the total noise level was set to produce 9 different SNRs uniformly tiling −24 to +24 dB. Model localization judgments (restricted to the 9 possible locations) were collected for each of the resulting 6768 stimuli (376 speech excerpts × 2 reverberation conditions × 9 SNRs). Separate psychometric functions were constructed for each reverberation condition by calculating the proportion of correct trials as a function of SNR. Speech localization thresholds were defined as the SNR yielding 70.7% trials correct, linearly interpolated.

**Statistics.** The statistical significance of the interaction between phase locking and reverberation conditions was assessed with a permutation test analogous to a mixed model ANOVA. An F-statistic was computed from the speech localization thresholds, with phase locking cutoff as a between-subjects factor and reverberation as a within-subjects factor. The F-statistic was re-computed 10,000 times with permuted reverberation labels to assemble the null distribution used to calculate a p-value for the actual F-statistic.

## Aggregate measures of task performance in noise

To summarize model task performance in noise (Fig. 2), we averaged model results across noise conditions for each task. For sound localization, we averaged mean absolute spherical errors across the 5 SNR conditions in Fig. 3d (−13.6, −6.8, 0, +6.8, and +13.6 dB). For word and voice recognition, we averaged recognition accuracy across the 5 SNR (−9, −6, −3, 0, and +3 dB) and 4 noise type conditions in Fig. 5b, c. We calculated 95% confidence intervals for each task performance summary metric by bootstrapping the model mean (sampling 10 neural network architectures with replacement 1000 times); these confidence intervals are plotted as the error bars on task performance in Fig. 2. The statistical significance of the effects of model manipulations

(degraded phase locking or delaying interaural integration) on overall task performance in noise was assessed by comparing mean task performance metrics against bootstrapped null distributions from the 3000 Hz phase locking model task performance metrics. Two-tailed *p*-values were estimated from Gaussian fits to these null distributions (as the probability of obtaining a score more extreme than that obtained from each degraded phase locking model under the null distribution). Effect sizes are quantified by measuring differences in the mean (Fig. 2a–c) or Cohen's d (Fig. 4e) between bootstrapped distributions of human-model similarity scores from different phase locking conditions.

### Aggregate measures of human-model similarity

Human-model behavioral similarity was quantified separately for each model and experiment, first with a Pearson correlation coefficient. In each case, we compared mean model behavior (averaged across the 10 neural network architectures) with mean human behavior (averaged across experiment participants). We calculated 95% confidence intervals for each human-model similarity value by bootstrapping the model mean (sampling 10 neural network architectures with replacement 1000 times). The statistical significance of the effects of model manipulations (degraded phase locking or delaying interaural integration) on overall human-model similarity was assessed by comparing mean human-model similarity scores against bootstrapped null distributions from the 3000 Hz phase locking model human-model similarity score (same analysis as for task performance metrics). Two-tailed *p*-values are reported, and effect sizes were quantified by measuring Cohen's d between bootstrapped distributions of human-model similarity scores from different phase locking conditions.

To ensure conclusions were robust to the choice of similarity metric, we repeated human-model comparisons by measuring root-mean-squared (RMS) error between analogous human and model data points. Data were first min-max normalized within experiments (rescaling human data to range from 0 to 1 across conditions, and then applying the same scaling factors to model data) to account for different units and scales across experiments. For the three word recognition experiments that measured proportion correct in different conditions (type of background noise, SNR, or F0 manipulation), the same min/max human scores (computed across all conditions) were used to normalize data. This prevented experiments that produced null effects (i.e., the lack of an effect of F0 manipulation on human word recognition) from artificially inflating the mean RMS error.

The two human-model similarity metrics measure different things. The correlation metric assesses the similarity in relative performance across conditions, whereas the RMS error can reflect absolute differences in performance between a model and humans. A "good" model should exhibit high similarity on both metrics. A model only needs to exhibit substantially lower similarity on one metric to be ruled out. This was the scenario we found for word recognition, where models with different phase locking limits were distinguished more clearly by the correlation metric (Fig. 2) than by the RMS metric (Supplementary Fig. 1).

We note that these two types of metrics have in some cases yielded inconsistent conclusions regarding previous ideal observer models[12]. Specifically, ideal observers of frequency discrimination that use information from phase locking exhibit much better absolute performance than humans, but replicate the qualitative dependence of thresholds on frequency. By contrast, ideal observers that do not have access to phase locking exhibit absolute thresholds closer to those of humans, but do not replicate the human dependence on frequency. Here we instead found the two types of metrics to yield comparable conclusions, in that models with the lowest phase locking limits never exhibited higher human-model similarity irrespective of which metric was used. Moreover, the models with higher phase locking limits

generally matched both absolute and relative performance and thus scored relatively well with both metrics. One difference compared to previous work is that our models were optimized for real-world tasks, and evaluated in real-world conditions as well as more traditional laboratory psychoacoustic assessments. We have found (here and elsewhere[18,19]) that such models tend to produce both absolute and relative performance on par with humans. This general finding is consistent with the idea that absolute performance reflects the demands of optimization for ecologically important tasks, such that optimizing a model for such tasks produces absolute performance that is close to that of humans.

### Comparison to ideal observer models of frequency discrimination

We strived to match the conditions of Heinz et al. (2001)'s ideal observer as closely as possible. We used the same auditory nerve model[12] and generated stimuli in the same way (all stimuli had cosine phase, were 200 ms in duration padded with 50 ms of silence, and had level roving of ±3 dB). Pairs of simulated auditory nerve representations of pure tones were presented to networks as two-channel inputs with shape [60 characteristic frequencies spanning 100–10,000 Hz, 5000 timesteps sampled at 20 kHz, 2 channels for the paired inputs]. This architectural choice gave the neural network flexibility to make comparisons between the two tones at any stage of representation within the feedforward processing stream, which we thought would increase the chances of finding a model that performed the task well. Spike counts were sampled from 200 high spontaneous-rate auditory nerve fibers per characteristic frequency. Sound levels were sampled independently for each of the tones in a trial (uniformly between 37 and 43 dB SPL). As the ideal observer was separately derived for each frequency, we separately trained models for 11 different quarter-octave frequency bands with center frequencies ranging from 250 to 8000 Hz. Frequencies above 8000 Hz were not considered to avoid influence from the model's maximum characteristic frequency. Within each band, training trials were generated by randomly sampling the frequency of the first tone ($f_1$, log-uniformly within the band), the interval magnitude $I$ (log-uniformly between 1e-6 and 1e-1 octaves), and the interval direction (+ or - with equal probability). The frequency of the second tone in the trial ($f_2$) was equal to $f_2 = f_1 \times 2^{\pm I}$.

Like the localization and speech models, the frequency discrimination models were feedforward convolutional neural networks. The output layers always had a single unit with a sigmoid activation function to map outputs between 0 and 1, representing the probability that $f_2 > f_1$. The network architectures were selected in a two-stage architecture search. First, we trained the top 100 networks from a prior random architecture search (conducted in earlier pitch modeling work[18]) on the frequency discrimination task using trials from all 11 frequency bands. We selected the best-performing architecture from this set and then performed a smaller local architecture search around this architecture by making 20 altered versions of it (e.g., by adding/removing a layer and enlarging/reshaping individual convolutional kernels). Finally, we selected the 10 top-performing neural networks from this set of 20 to use for our frequency discrimination models (Supplementary Table 3). The architecture search used auditory nerve representations with a 3000 Hz IHC filter cutoff.

Models with 3000, 1000, 320, and 50 Hz IHC filter cutoffs were then separately trained on each frequency band. Each model was trained on 640,000 trials using the Adam optimizer to minimize the binary cross-entropy loss function. We used a batch size of 32 and an initial learning rate of 1e−5 that decreased by a factor of 10 every 5000 steps. We evaluated the models on 10500 within-distribution trials by keeping $f_1$ equal to the center of the frequency band and ranging the interval magnitude from 5e−7 to 5e−1 octaves. We constructed psychometric functions for each model and frequency band;

discrimination thresholds were defined as the interval magnitude yielding 75% of trials correct.

## Error bars

Except where otherwise noted in figure captions, error bars in results figures indicate ±2 standard errors of the mean across 10 neural network architectures (model results) or across experiment participants (human results). There are two exceptions. The first is in plots of aggregate measures of model task performance and human-model similarity (Fig. 2a–c, Fig. 4e, Supplementary Fig. 1a–c, and Supplementary Fig. 9b–g), where error bars indicate 95% confidence intervals bootstrapped across 10 neural network architectures. The second exception is Fig. 3h, where the human error bars indicate ±2 standard errors from 1 listener (the original experiment's only participant[68]) averaged across 4 different pure tone frequencies.

## Reporting summary

Further information on research design is available in the Nature Portfolio Reporting Summary linked to this article.

## Data availability

All data, models, and stimuli are available at https://github.com/msaddler/phaselocknet. Source data are provided with this paper.

## Code availability

Code for training and evaluating models, running experiments, analyzing results, and generating figures is available at https://github.com/msaddler/phaselocknet. Models and analyses were implemented in Python (3.11.4) using the TensorFlow (2.13.0) and PyTorch (2.2.1) deep learning libraries.

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

## Acknowledgements
We thank Bertrand Delgutte, Jim DiCarlo, Michale Fee, members of the McDermott lab, and four reviewers (Ian Bruce, Michael Heinz, David McAlpine, and Brian Moore) for their helpful feedback on the manuscript. This work was supported by National Institutes of Health grants R01DC017970 and R01DC021464 to J.H.M.

## Author contributions
M.R.S. and J.H.M. conceived the project, designed the experiments, and drafted the manuscript. M.R.S. ran the experiments, analyzed the data, and made the figures.

## Competing interests
The authors declare no competing interests.
