## [Transparent Peer Review file · Nature Communications]

Models optimized for real-world tasks reveal the task-dependent necessity of precise temporal coding in hearing

Corresponding Author: Dr Josh McDermott

Version 0:

Reviewer comments:

Reviewer #1

(Remarks to the Author)

This manuscript describes a computational modeling study, in which the output of a physiologically-detailed model of the mammalian auditory periphery provided input to machine learning models that were trained on real-world auditory tasks. Comparisons are made to the corresponding human performance on these tasks (where possible) and the effects of altering the auditory model's cutoff frequency for temporal fine structure (TFS) coding are investigated. In general, TFS cues above 1 kHz are required for human-like performance on many auditory tasks, but not for the speech-in-noise tasks examined in this study.

Overall, the study is conducted in a very thorough manner and the manuscript well written. The authors do a quite good job of explaining their study to a broad audience and provide insightful interpretation of the results. This manuscript provides an important contribution to the literature on the role of acoustic TFS on auditory perception.

I have just a few comments on sections of the manuscript where some further clarification could be provided:

- p. 3, lines 86-87: The observation that the frequency limit for binaural processing is lower than the frequency limit for phase locking in auditory nerve fibers (ANFs) is possibly due to bushy cells in the cochlear nucleus, which provide the synaptic input into the binaural circuits of the auditory brainstem, having enhanced synchrony below ~ 1 kHz but degraded synchrony (relative to ANFs) above that frequency. See Fig. 4 of Joris et al. (J. Neurophys. 1994; <https://doi.org/10.1152/jn.1994.71.3.1022>).

- p. 5, line 145: It would be perhaps helpful to clarify that for the IHC it is the combination of the IHC membrane capacitance and the ion channel conductance/resistance that produces the lowpass filtering.

- p. 13, lines 424-425: Again, the frequency limit of phase locking in bushy cells may explain this discrepancy.

- p. 19, lines 669-672: A possible explanation for this is that frequency modulations in signal components (effectively carrier frequencies) above 1 kHz can be turned into amplitude modulations by the bandpass filtering of the cochlea, which has been referred to as envelope recovery. These recovery amplitude modulations may provide additional envelope information beyond that which is directly encoded in the tone vocoder. See, for example: Fig. 10 of Heinz & Swaminathan (JARO 2009; <https://doi.org/10.1007/s10162-009-0169-8>) and Shamma & Lorenzi (JASA 2013; <https://doi.org/10.1121/1.4795783>).

- p. 23, lines 812-813: Previous studies looking at the roles of ENV and TFS cues in speech perception have also indicated that the ENV cues contribute more to speech intelligibility, and TFS cues play a secondary part and may be beneficial just at negative SNRs or if the ENV cues are very degraded, e.g., Swaminathan & Heinz (J. Neurosci. 2012; <https://doi.org/10.1523/JNEUROSCI.4493-11.2012>) and Wirtzfeld et al. (JARO 2017; <http://doi.org/10.1007/s10162-017-0627-7>). However, for many of these studies, ENV cues are considered to be amplitude modulations only up to 32 or 64 Hz, whereas the authors' findings of temporal cues being potentially useful up to 320 Hz do then include voicing pitch, and the first harmonic or two for male voices with a lower voicing pitch, and these would be considered part of the TFS cues in these previous studies. These issues are worthy of some discussion at this point in the manuscript.

- p. 35, lines 389-391: I am a bit confused by this statement, because fusion refers to when there is still a perception of one

sound, but its perceived location is based on a fusion of the binaural cues from the two acoustic sources, rather than those of the acoustic source that arrives first. So, the authors' model might still be able to provide predictions in the fusion region.

(Remarks on code availability)

Reviewer #2

(Remarks to the Author)

This is an excellent paper that fully merits publication in Nature Communications. The paper addresses the important issue of the role of neural phase locking in determining the performance of the human auditory system. The paper compares the performance of human listeners with that of machine-learning models optimized to perform real-world tasks. The upper limit of phase locking in the models is systematically varied. The results suggest that phase locking plays an important role for some tasks but not for others.

The work is highly original and newsworthy. The paper is generally very clear and well written. A huge amount of work covering a large variety of auditory tasks is presented.

One concern arises with lines 959-962. The instantaneous firing rates in the model were downsampled to 10 kHz. In other words the instantaneous firing rates were sampled every 100 microseconds. This would lead to a rather coarse representation of phase locking at 3000 Hz. In response to a 3000-Hz tone the interspike intervals cluster around integer multiples of 333 microseconds, and such intervals would be poorly represented when only sampled every 100 microseconds. I wonder if this might have contributed to the finding that there was no benefit in increasing the phase locking limit in the models from 1000 to 3000 Hz.

Otherwise, I have only minor concerns and comments, as listed below.

Throughout, put a space between numbers and units: 1 s not 1s.

Line 80. Change prosthetics to prostheses. The statement actually relates to cochlear implants, so they could just as well say cochlear implants rather than auditory prostheses.

Line 152. Change "on par with" to "similar to".

Line 216. Change transduced to conveyed.

Line 222. Insert by after increased.

Line 224. Change shows to showed.

Line 290. Perhaps a brief explanation of "texture-like" is needed here.

Line 327. Insert was before played.

Line 335. Insert comma after elevation.

Line 342. Add s to end of sound.

Line 350. Change to "One demonstration of this comes..."

Line 355. Insert specified before as.

Line 363. "is lowpass" reads strangely. Perhaps say "because envelope modulation rates are usually relatively low".

Line 365. Delete even.

Line 437. Change localizes to localized.

Lines 440-442. I could not follow "such that delaying the comparison incurs little cost."

Line 486. Change in to by.

Line 566. Add s to end of model.

Line 572. Insert "of phase locking" after benefit.

Line 629. Insert "that for" before humans.

Line 635. Change remove to reduce. Even the tone-vocoded channels convey some useful temporal fine structure information.

Line 645. Change uninformative to less informative.

Line 710. Change to "to be correlated with".

Line 714. Change report to reported.

Line 760. Change on to for.

Line 833. I was not sure what was meant by "expressive".

Line 986. Change is to was.

Line 1023. Perhaps give a brief explanation of "dropout regularization" and "softmax classification".

Line 1125. Change was to were.

Line 1190. Delete the.

Line 1194. Insert was before played.

In this paragraph and elsewhere, use "loudspeaker" rather than "speaker".

Line 1123. Change report to were.

Line 1251. Change to "A perceptual weight of 0".

Line 1309. Insert for after as.

Lines 320-328. Perhaps change ears to pinnae.

Line 1333. It should be noted that the use of different real HRTFs involves less drastic changes than produced by inserting molds into the ears.

Line 1381. Insert as at the start of the line.

Lines 1390-1391. Is there a reason why models could not be trained to report the presence of two source locations?

Line 1396. Change dependency to dependence.
Line 1400. Change the first to to and.
Line 1493. Change in to using and change using to with.
Line 1465. Insert for after as.
Line 1467. Delete held.
Line 1468. Insert give after to.
Line 1472. Delete s at end of types.
Line 1476. Insert recognition after noise.
Line 1485. Delete held.
Line 1491. Change in to for.
Lines 1506-1546. Change pitch to F0.
Line 1540. Change collect to collected.
Line 1569. Insert “for speech” after accuracy.
Line 1571. Change describe to described.
Line 1582. Insert “that for” after to.
Line 1604. Insert was before played.
Line 1615. Change was to as.
I would like my name to be revealed. Brian C.J. Moore

Reviewer #3

(Remarks to the Author)

This is potentially a very interesting and informative manuscript that takes a novel approach to assessing the importance, or otherwise, of phase-locking—action potentials whose timing is dependent on the phase of the sound stimulus cycle, here within the condensation and rarefaction period of a single cycle of a pure tone, the temporal fine structure (TFS).

It is indisputably the case that for sound source location based on the relative timing of the sound arriving at each ear, phase-locking of action potentials to the TFS is critical. Importantly, although the term TFS SHOULD exclusively be employed in this way and the way the authors do so, the term has recently been co-opted into psychoacoustic reports to accommodate spectral/frequency resolution. This is not the same thing, and the distinction made here is critical. Loss of spectral resolution in the cochlea, including from sub-clinical hearing loss (so-called ‘hidden hearing loss’, possibly the consequence of synaptopathy in one or more populations of auditory nerve fibres), for example, suggests that loss of TFS might be the factor determining poor speech understanding in noise, but no direct evidence of this exists, whilst recent evidence suggests even this form of hearing loss likely generates broader cochlear filters.

What is less clear is whether TFS is also required for so-called monaural listening tasks (i.e. when no spatial differences are explored). The current manuscript holds that this is the case, and that it relates to distinguishing different talkers from each other—based on voice pitch—but not for word discrimination. Of these two, voice pitch is evolutionarily conserved, and serves a similar purpose across species—determining source size (proxy for age, sex etc.), whilst word recognition is largely reserved for humans in whom spoken language evolved.

I have several points for the authors to consider and to which they should respond.

1. I appreciate, and consider this a strength of the manuscript, that models of performance based on real-world listening likely inform us as to the mechanisms that evolved to achieve tasks, and their bounds. This distinguishes the model from a psychoacoustic-type assessment, which is often stimulus determined. Distinguishing this model more from current/previous neural modelling studies would be important, including a more explicit statement on what makes this different. For example, it is not usual to consider binaural integration of phase-locked inputs without fully realized intermediate stages of processing that might reduce (or enhance, for frequencies below around 250 Hz) phase-locking. This could be tested. I would also like to hear from the authors as to why they think neural responses in the cochlear nucleus, at least in dedicated neurons, show hyper-phase-locking relative to the auditory nerve for these low-frequencies when they make the point about the ‘lossy’ nature of synaptic transmission in intermediate stages of processing relative to simply converging inputs from both ears onto binaural coincidence detectors.

2. Although I appreciate the need to at least acknowledge the role of pinna cues in spatial listening—specifically their role in determining elevation of sources, the use of modified pinna cues (coupled with the ability of human listeners to learn ‘new ears’—as reported by Hofman et al 1998; cited by the authors) I am not convinced of the need to do more than acknowledge this in the current manuscript. Currently, this section in the Results reads as a ‘strawman’ (and see Figure 5). The collapse of elevation sensitivity when pinna cues are modified leaves sensitivity to azimuth unchanged (Figure 5e top panel) other than for the very-low pass IHC temporal filter. However, I don’t see the point of Figure 5e bottom panel, not in terms of what it shows, but why the authors consider this relevant. Absent correct pinna cues I would expect only ITD based horizontal localisation, and therefore the localisation deficit in this bottom panel is ‘simply’ the result of instantiating the low-pass IHC filter that was used to assess source localisation the previous sections. It certainly doesn’t interact with the smoothing of the spectral shape either, so I don’t see what else this is adding. If anything was to be removed or downplayed in this fairly length manuscript, it would be this. It might provide for completeness, but it is not surprising in the least empirically or theoretically. Removing it to supplemental would save some space in the manuscript.

3. The perspective that voice pitch might limit the upper bound of monaural phase locking that can be extracted ignores distinguishing higher pitches from other sources. This seems a very human-centric perspective (I don't necessarily disagree; voice pitch is determined by vocal tract length which provides for the same purpose of distinguishing size of an individual as it does in other species) but what about higher pitches? Is distinguishing voice pitches 'special' in which case this should be evident in the relative ability (in some way) to process pitch? It is also worth noting that phase-locking to upper harmonics in the envelope would be pass-band limited, with cochlea filters of around 4 kHz having bandwidths of approx. 350 Hz (i.e. just less than 1/10 of the centre frequency). Is it fortuitous or otherwise that this limit seems to hold? 4kHz seems to mark a transition in pitch processing, whether spectral or temporal.

4. Cochlear implants are the true human perspective for life without TFS phase-locking. Implant users are usually now well trained on, and highly capable of parsing, speech in quiet, but they struggle in noise and in terms of sound localisation. In the latter, pitting interaural level or timing cues against each other they consistently adopt the level cue, ignoring the usually more potent timing cue. They also struggle to resolve pitch per se., including voice pitch. The manuscript could usefully discuss the data with respect to this unique population of human listeners. This is especially so, given the section on tone vocoding (implant users' peripheral auditory nerve fibres are acting likely similar to speech/noise vocoders).

(Remarks on code availability)

Reviewer #4

(Remarks to the Author)

This manuscript describes an extremely thorough and rigorous comparison of modeling analyses and human performance (some collected here, but mostly from literature) that address a fundamental question in auditory neuroscience that has been debated for many decades – whether fine temporal coding is relied upon by human listeners for perception. This question is highly significant in the field of auditory neuroscience as it has major theoretical and translation implications, e.g., for the design/re-design of auditory prostheses such as cochlear implants.

This work is highly significant in that it uses modern machine learning techniques to allow this question to be evaluated for real-world listening tasks that are highly relevant to the daily lives of human listeners (e.g., complex sound localization, word and voice recognition) and as such are likely to be what the human brain trained on in learning to use the most useful auditory cues that enter the auditory system. As such, this work advances the field in important and significant ways by showing specific tasks (e.g., sound localization, voice recognition, but not word recognition) for which fine temporal cues are needed to explain human perception.

The work (both modeling and experimental) appears to be rigorously carried out in almost all regards (see one concern below) and is extremely thorough in its breadth of experimental conditions and stimuli considered. As such, the conclusions are well justified. Limitations in the approach are well considered as well in the discussion section, adding to the rigor of this work. However, several issues exist that can be addressed to improve the presentation of this work.

Furthermore, although the thoroughness is admirable, the downside is that the manuscript is dense and long with many details that only auditory perceptual scientists will fully appreciate. As such I worry about the accessibility of the big-picture take-home points to the broad audience that will surely be interested in this modern approach to a classic question. I don't have exact suggestions to cure this as the thoroughness is what makes this work so rigorous, but I encourage the authors to do as much as they can (beyond what they have already done in many ways with nice overview paragraphs) to make this long manuscript accessible to a broad audience. It is not that the details are not able to be followed, in fact they are well written, it is just that I worry that the general reader will struggle to separate the main broad findings from the necessary details that may be out of their expertise. It would be a shame to lose the forest for the trees.

General Comments

- While this paper (in full detail) does a good job of highlighting that certain tasks may require fine timing, but others do not, the title suffers a bit from suggesting that fine timing is necessary in all conditions (as if there is one answer to this question for all tasks). I recognize the title is limited in space and not able to fit in qualifications, but there is other recent work in the field suggesting the opposite conclusion (e.g., Whiteford and Oxenham – not currently cited) and so it is important to make clear (as soon as there is space to do so) that there is not necessarily one answer to this question for all tasks and as such different studies can get different answers but not be inconsistent with each other. In this regard, some discussion of the apparently opposing strong results from this study and strong results from the Whiteford and Oxenham work would help the reader to reconcile opposing answers they will find in the literature (and in fact in the same study, such as this one, where the answer depends on the task).

- Much nice discussion is provided about the differences between optimal observer approaches, where the decision process can be proven mathematically to be optimal but are limited to simple tasks, and ML approaches that are able to be applied to real-world complex tasks (by optimizing parameters based on empirical data). But this leads to confusing wording (often in same sentence or adjacent sentences), because different definitions of optimal & optimization are being used, when at face value they would be assumed to be related. E.g., lines 47-51: "Although not provably optimal, ... human-like behavior can emerge in deep artificial neural networks optimized for such tasks." Please work to clarify this important but subtle difference where optimizing a system does not mean making it optimal.

- This is brought up a bit in the Discussion section limitations (lines 798-805; 831 onward), but defining optimal and optimizing earlier would help as it comes up in the intro. In fact, the discussion on lines 798-800 addresses this a bit but not entirely correctly – The classical ideal observer theory is not limited to simple "stimuli", only simple "tasks" – e.g., a simple one-parameter discrimination task is tractable for optimality with arbitrarily complex stimuli (e.g., Heinz et al 2001) – so I suggest removing "stimuli" from line 799. To address the question of whether ML approaches are in fact close to optimal (lines 831 onward), it would be nice to add in a condition where both the ML and optimal detector approaches are viable (e.g., Maximum-likelihood estimators and CR bounds can be derived) and see if they agree (e.g., pure-tone frequency discrimination limens (FDL) that Siebert and Heinz et al analyzed (analytically and computational, respectively). This would provide a nice result to cite that the same ML approaches used for the complex real-world tasks are in fact close to optimal (e.g., able to use fine timing to perform several orders of magnitude better than humans) when we know mathematically what optimal performance actually is.
- A related comment – I think it should be made more clear that the ML models are not in fact fit to the human data at all, but rather are just training to do the task as well as possible. I think this is made more clear in previous publications from this group, but it is a critical point for the arguments being made. In fact, if the FDL experiment could be added and turned out as I expect it would (i.e., that the ML models greatly outperformed humans), this (and the ITD with 3000-Hz cutoff condition) can be cited to illustrate this point and allow us to think of these as likely near optimal models.
- One methodological question that arose that also makes it important to compare this approach to Siebert's approach is the question of stochasticity in neural spike trains. It is well known that auditory nerve fiber spike times are well described as a Poisson process with refractory effects. My understanding from the methods (e.g. line 978) is that each time bin is treated independently and as binomial distribution. The independence contradicts refractory effects, and it is not clear how similar the neural statistics here are to the known Poisson statistics of AN fibers. Some discussion and/or simulations will help to be sure this is not a significant deviation from known AN statistics (I doubt it is, but needs to be checked). The simulations suggested above to see if the ML approach matches Siebert's predictions will also help to address this. The implications of getting AN-fiber statistics correct for these real-world predictions should be discussed (e.g., refractory effects on the usefulness of high-frequency fine-time information). The Poisson statistics have been shown by Siebert and others to be important for intensity discrimination tasks, but it would be interesting to discuss whether they are critical for real-world tasks.
- Similarity metrics – More guidance is needed on why two similarity metrics are used and how to interpret them, and this should be tied into the decades long interpretation issue with respect to the questions being addressed here: is it the absolute similarity between model and human performance that matters (has been used to suggest rate code for FDL predictions), or the pattern that matters (has been used to suggest timing code in FDL predictions). It seems like the two metrics being used (RMS and correlation) are trying to get at this issue, but this is not really discussed well enough in the general framework to make that issue clear to the reader. Perhaps it is because the two are more similar in these real-world tasks (but not always!), but the current predictions and use of these similarity metrics should be tied into this discussion with respect to this debate in the past literature addressing these questions.
- Overall it could be made more clear where the human data being compared to comes from (i.e., measured as part of this study, other studies from the authors' group, or from others in the literature). A lot of this is in fact in the manuscript in detailed descriptions, but given length of the manuscript this gets lost at times and overall. I think it would benefit the presentation of this work to be sure it is explicit in each section/figure caption, as well as provide some sense in abstract or more likely (given space constraints) in intro a sense that most of the data is actually from established data from the literature, not collected here (but that some human data were collected by the authors).

Specific Comments

- Line 16. "is uncertain" does not do justice to the breadth of work already done to explore this question – suggest "remains uncertain".
- Line 22 "ecological task demands" and "neural implementation constraints" are quite vague here in the abstract and only become clear (mostly) well into the manuscript. Consider being more specific in describing the actual results (e.g., that binaural sluggishness was needed to explain human behavior in localization). May be hard given space constraints in abstract, but will help because as it is now it is hard to understand what is meant from just the abstract.
- Lines 23-24. This is an interesting point, but is not clear as written what exactly is meant ("in principle" is pretty vague). Prosthetics are briefly discussed in the discussion, but not in this light – i.e., not about trying to overcome the failure to provide fine temporal coding and whether that would restore near-normal hearing. Is there evidence that prostheses that fail to provide high-fidelity temporal cues lead to performance in line with the 50- or 320-Hz model predictions here? None of this is discussed anywhere in the manuscript, so this final sentence seems to be an overstep as written.
- The ITD condition where central limitations were needed to account for human data that degrades above ~1500 Hz, but the monaural ML model degrades above 3000 Hz is a nice example suggesting central limits may be needed, but begs the question whether a model without the central limits but with 1500-Hz IHC cutoff could explain these data just as well. If so, what else in the broad set of predictions here would fail with 1500-Hz cutoff? Since we don't actually know human cutoff, what if human PL cutoff is actually 1500 Hz, not 3000 Hz, as others have proposed (e.g., in some sections of Verschooten et al 2019). This possibility should be discussed.
- Line 948. "S" fiber types; line 978 "F" fibers – seems like S is used throughout, so not sure why F used here.
- Fig. 3f – clarify noise is signal here, not masker.
- Fig. 3d – mention inverted axis, such that down is more error, up is better.
- Line 408 Section Title – would be nice to be more specific – "delayed binaural integration"
- Fig.5. stimuli should be stated in caption (unless I missed it)

- lines 480-481 Section heading. It is hard to see this result, since data supporting it are indirect. Title may not be place to clarify exact evidence for this conclusion, but fig caption perhaps or explicitly stating argument explicitly in text to get to this conclusion.
- Fig. 8e – mention somewhere why no human data are shown here, in contrast to almost all other panels in the paper.
- Line 785. Some explanation of why “limited benefit” above 1000 Hz would help (even if not known for sure). E.g., perhaps at higher frequencies head size limits usefulness of phase comparisons (that become ambiguous) when wavelengths get short re: head size.
- Line 849 – what does “spike sampling” mean? Is this just another way to say stochastic nature of spike trains, which is how the randomness has been discussed in the past.
- Line 921. Why “apparent” stochasticity? This has been shown to exist many times and is well characterized.
- Supp Fig 8 (and 9) caption – mention Fig. 2 (and 3) as comparison with full AN model.

(Remarks on code availability)

N/A

Version 1:

Reviewer comments:

Reviewer #1

(Remarks to the Author)

The authors have done an excellent job at responding to my comments on the original manuscript.

(Remarks on code availability)

Reviewer #2

(Remarks to the Author)

The authors have responded comprehensively and appropriately to the comments of the reviewers. I have only minor suggestions for further changes. It should not be necessary for me to review the paper again.

Line 103. Change in to using.

Line 121. I am not sure if this is the journal style, but “Data are...” would be more correct.

Lines 125-126 partly repeat lines 123-124.

Lines 269-270 and elsewhere (e.g. lines 278-279, 458, 829). This implies that phase locking above 1000 Hz is not used at all. But an IHC cutoff frequency of 1000 Hz only means that the precision of phase locking in the model decreases above 1000 Hz. Usable phase locking may occur at higher frequencies, consistent with evidence that TFS plays a role in sound localization for frequencies up to 1500 Hz.

Lines 568, 572, 582, 592, 593, 594, 596, Fig. 6b, 601, 705, 787, 842, 1262, 1611, and elsewhere. Use the term “pitch” to refer to the subjective attribute and use F0 (consistently) to refer to the physical fundamental frequency.

Figure 7c. The human data show a marked increase as the number of channels with TFS increases from 0 to 8, while none of the models shows a marked increase. This merits a small comment.

Line 774. Insert “performance for” before “impaired”.

Line 928. Change “fundamental frequency” to F0.

Lines 979-980. Change “fundamental frequencies” to F0s.

Lines 1082-1083. This reads a bit strangely.

Line 1798. Change idea to ideal.

I would like my name to be revealed. Brian C.J. Moore

(Remarks on code availability)

The code appears to be valid and correct.

Reviewer #3

(Remarks to the Author)

I am very pleased with the edits the authors have made for this manuscript, especially concerning the perspective on spatial hearing and the removal of the section concerning head-related transfer functions to supplemental information.

(Remarks on code availability)

Reviewer #4

(Remarks to the Author)

Thanks for your thorough responses to my comments and those of the other reviewers. All of my concerns were addressed well in the revisions. This rigorous and thorough manuscript is now even stronger, and will make a very nice contribution to the literature.

Mike Heinz

(Remarks on code availability)

Saddler and McDermott - Response to Reviews

Please note that all line numbers quoted in this response document refer to the pdf that has tracked changes.

Reviewer #1 (Remarks to the Author):

This manuscript describes a computational modeling study, in which the output of a physiologically-detailed model of the mammalian auditory periphery provided input to machine learning models that were trained on real-world auditory tasks. Comparisons are made to the corresponding human performance on these tasks (where possible) and the effects of altering the auditory model's cutoff frequency for temporal fine structure (TFS) coding are investigated. In general, TFS cues above 1 kHz are required for human-like performance on many auditory tasks, but not for the speech-in-noise tasks examined in this study.

Overall, the study is conducted in a very thorough manner and the manuscript well written. The authors do a quite good job of explaining their study to a broad audience and provide insightful interpretation of the results. This manuscript provides an important contribution to the literature on the role of acoustic TFS on auditory perception.

Thank you.

I have just a few comments on sections of the manuscript where some further clarification could be provided:

- p. 3, lines 86-87: The observation that the frequency limit for binaural processing is lower than the frequency limit for phase locking in auditory nerve fibers (ANFs) is possibly due to bushy cells in the cochlear nucleus, which provide the synaptic input into the binaural circuits of the auditory brainstem, having enhanced synchrony below ~ 1 kHz but degraded synchrony (relative to ANFs) above that frequency. See Fig. 4 of Joris et al. (J. Neurophys. 1994; <https://doi.org/10.1152/jn.1994.71.3.1022>).

We have noted this in the revised manuscript:

“Although there are physiological correlates of this limit (cells that provide input to brainstem binaural circuits exhibit degraded synchrony above 1 kHz)⁴⁶, it is not well understood in normative terms” (lines 116-118)

- p. 5, line 145: It would be perhaps helpful to clarify that for the IHC it is the combination of the IHC membrane capacitance and the ion channel conductance/resistance that produces the lowpass filtering.

We have clarified this in the revised manuscript:

“The fidelity of temporal coding in the mammalian ear is limited by the capacitance and ion channel properties of the hair cell membrane³⁰ as well as the hair-cell-to-nerve-fiber synapse⁴⁹, both of which act as low-pass filters.” (lines 177-179)

- p. 13, lines 424-425: Again, the frequency limit of phase locking in bushy cells may explain this discrepancy.

We have rephrased the description of this finding in the revised manuscript, to make the likely relationship between known binaural anatomy/physiology clearer:

“This discrepancy with humans is consistent with the known anatomy of the binaural system... The lower limit of ITD sensitivity in humans is plausibly due to anatomical constraints that force information from each ear to pass through additional synapses before being compared, with some loss of temporal precision at these synapses⁴⁶.” (lines 515-520)

- p. 19, lines 669-672: A possible explanation for this is that frequency modulations in signal components (effectively carrier frequencies) above 1 kHz can be turned into amplitude modulations by the bandpass filtering of the cochlea, which has been referred to as envelope recovery. These recovery amplitude modulations may provide additional envelope information beyond that which is directly encoded in the tone vocoder. See, for example: Fig. 10 of Heinz & Swaminathan (JARO 2009; <https://doi.org/10.1007/s10162-009-0169-8>) and Shamma & Lorenzi (JASA 2013; <https://doi.org/10.1121/1.4795783>).

We have added a discussion section to the revised manuscript that addresses this and other issues associated with vocoder manipulations:

“A long tradition of psychophysical research has also attempted to test the role of phase locking in perception^{37-39,36,96,41}. Such studies have typically used stimuli intended to isolate or remove information conveyed by phase locking, often by measuring the envelope and fine structure from the output of a set of auditory filters, and then generating stimuli in which either the envelope or fine structure are rendered uninformative or otherwise altered. One challenge for these “vocoder” approaches is that if the resulting stimuli are analyzed with a filter bank that is distinct from the one used for stimulus generation, information that was limited to one stimulus component (e.g. the fine structure) during stimulus generation can appear in a different stimulus component (e.g. the envelopes) of the analysis filter bank^{97,98}. It is thus difficult to know whether a stimulus that is intended to remove a particular type of information actually succeeds in doing so once the stimulus is represented in the ears of a listener.” (lines 1071-1097)

- p. 23, lines 812-813: Previous studies looking at the roles of ENV and TFS cues in speech perception have also indicated that the ENV cues contribute more to speech intelligibility, and TFS cues play a secondary part and may be beneficial just at negative SNRs or if the ENV cues are very degraded, e.g., Swaminathan & Heinz (J. Neurosci. 2012;

<https://doi.org/10.1523/JNEUROSCI.4493-11.2012>) and Wirtzfeld et al. (JARO 2017; <http://doi.org/10.1007/s10162-017-0627-7>). However, for many of these studies, ENV cues are considered to be amplitude modulations only up to 32 or 64 Hz, whereas the authors' findings of temporal cues being potentially useful up to 320 Hz do then include voicing pitch, and the first harmonic or two for male voices with a lower voicing pitch, and these would be considered part of the TFS cues in these previous studies. These issues are worthy of some discussion at this point in the manuscript.

We have clarified in the Results section that any manipulation to the auditory nerve will affect the coding of both TFS and envelope:

“The lowest cutoff eliminates essentially all phase locking to temporal fine structure in natural sounds (as well as to envelope modulations above the cutoff).” (lines 190-191)

We have added the following sentences to the Discussion section:

“Another challenge for these approaches is conceptual. The signal processing distinction between envelope and fine structure is well-defined at the stage of stimulus generation, but is lost at the auditory nerve, which converts the entirety of the stimulus into a single representation of spiking activity. Degradations of phase locking thus potentially affect the encoding of both the envelope and fine structure of a stimulus. For instance, the difference in performance that we observed between models with phase locking cutoffs of 50 and 320 Hz could partially reflect degradation of what would traditionally be considered envelope cues.” (lines 1082-1097)

- p. 35, lines 389-391: I am a bit confused by this statement, because fusion refers to when there is still a perception of one sound, but its perceived location is based on a fusion of the binaural cues from the two acoustic sources, rather than those of the acoustic source that arrives first. So, the authors' model might still be able to provide predictions in the fusion region.

We have clarified that the model in its current form cannot predict whether there are one or two sounds. The model assumes there is one sound and provides a location estimate for that one sound.

“The model judgments can reflect one of these (the localization dominance of the leading sound), but because the models cannot report the presence of more than one sound source location, they cannot explicitly exhibit one of the other main precedence phenomena (the perception of two distinct sources when the delay between leading and lagging clicks is large).” (lines 1783-1794)

Reviewer #2 (Remarks to the Author):

This is an excellent paper that fully merits publication in Nature Communications. The paper addresses the important issue of the role of neural phase locking in determining the performance of the human auditory system. The paper compares the performance of human listeners with that of machine-learning models optimized to perform real-world tasks. The upper limit of phase locking in the models is systematically varied. The results suggest that phase locking plays an important role for some tasks but not for others.

The work is highly original and newsworthy. The paper is generally very clear and well written. A huge amount of work covering a large variety of auditory tasks is presented.

Thank you.

One concern arises with lines 959-962. The instantaneous firing rates in the model were downsampled to 10 kHz. In other words the instantaneous firing rates were sampled every 100 microseconds. This would lead to a rather coarse representation of phase locking at 3000 Hz. In response to a 3000-Hz tone the interspike intervals cluster around integer multiples of 333 microseconds, and such intervals would be poorly represented when only sampled every 100 microseconds. I wonder if this might have contributed to the finding that there was no benefit in increasing the phase locking limit in the models from 1000 to 3000 Hz.

To test for this possibility, we re-ran all of the experiments with the 3000 Hz phase locking models using auditory nerve representations sampled at 20 kHz rather than 10 kHz. This increase by a factor of two had a negligible effect on the results, indicating that the sampling rate does not explain the similar results for 1000 Hz and 3000 Hz phase locking limits. We note that this conclusion is also supported by considerations of the sampling theorem (a sampling rate of 10 kHz should allow the representation of signal frequencies up to the Nyquist limit of 5 kHz), which is relevant in a setting where responses for multiple nerve fibers are being summed.

This control experiment is described in the revised text:

“The high sampling rate of the model auditory nerve responses was intended to ensure that the information in phase locking up to 3000 Hz could be faithfully represented (the Nyquist limit for a 10 kHz sampling rate is 5 kHz, well above the highest limit used in our models). However, the discretization of time that results from this representation causes inter-spike intervals to be quantized, which might be expected to result in some loss of information, particularly at frequencies close to the upper limit of 3000 Hz. To test whether the downsampling of firing rates to 10 kHz could have limited the benefit of high-frequency phase locking, we repeated all experiments on the models with a 3000 Hz phase locking limit, instead using an auditory nerve sampling rate of 20 kHz. To keep model sizes and architectures similar after doubling the input time dimension, we modified the first two stages of each neural network to reflect the higher sampling rate.

Specifically, the kernels in the first two convolutional stages in each model had twice as many taps along the time axis and the extent of temporal pooling in the second pooling stage was doubled. We reasoned that these modifications would give the models the best chance to extract information from high-frequency phase locking without doubling the number of learnable parameters in the final fully-connected layers (which would plausibly create a confound that seemed better to avoid). Despite roughly preserving the number of learnable parameters, the GPU memory footprint of these higher-sampling-rate models is considerably larger (because the output of the convolution operation contains more activations). We were able to train the models by halving the batch sizes and training for twice as many steps, thus keeping the total number of training examples constant across the two sampling rates.

The 20 kHz sampling rate models produced extremely similar results to the default 10 kHz sampling rate models (Supplementary Fig. 12, 13, and 14). For the sound localization and voice recognition models there were no statistically significant differences between the 20 kHz and 10 kHz models in overall task performance ($p > 0.07$) or human-model similarity ($p > 0.35$). For the word recognition model, there was a very small increase in overall task performance (48.0% correct compared to the 47.3%, 47.5% and 46.8% correct for the 10 kHz sampling rate models with 320, 1000, and 3000 Hz phase locking limits, $p < 0.01$) but no increase in human-model similarity ($p = 0.29$). These results suggest the 10 kHz auditory nerve sampling rate did not contribute to the lack of benefit observed for phase locking above 1000 Hz.” (lines 1335-1343)

a. Naturalistic task performance in noise

b. Human-model similarity (Pearson correlation)

c. Human-model similarity (RMS error)

Otherwise, I have only minor concerns and comments, as listed below.

Throughout, put a space between numbers and units: 1 s not 1s.

Done

Line 80. Change prosthetics to prostheses. The statement actually relates to cochlear implants, so they could just as well say cochlear implants rather than auditory prostheses.

Changed to “cochlear implants”

Line 152. Change “on par with” to “similar to”.

Done

Line 216. Change transduced to conveyed.

Done

Line 222. Insert by after increased.

Done

Line 224. Change shows to showed.

Done

Line 290. Perhaps a brief explanation of “texture-like” is needed here.

We added a clarifying sentence:

“Background noises were selected to be more temporally homogeneous than the targets (e.g., the sound of running water rather than a single splash; see Methods) to ensure the task was well-defined ” (lines 346-348)

Line 327. Insert was before played.

Done

Line 335. Insert comma after elevation.

Done

Line 342. Add s to end of sound.

Done

Line 350. Change to “One demonstration of this comes...”

Done

Line 355. Insert specified before as.

Done

Line 363. “is lowpass” reads strangely. Perhaps say “because envelope modulation rates are usually relatively low”.

Done

Line 365. Delete even.

Done

Line 437. Change localizes to localized.

Done

Lines 440-442. I could not follow “such that delaying the comparison incurs little cost.

We meant to convey that because there is little benefit from temporal coding above 1000 Hz, a system without access to higher frequency phase locking (i.e., due to delayed interaural comparison) would be no worse off. This point is better made at the end of the following paragraph, so we deleted the confusing phrase here.

Line 486. Change in to by.

Done

Line 566. Add s to end of model.

Done

Line 572. Insert “of phase locking” after benefit.

Done

Line 629. Insert “that for” before humans.

Done

Line 635. Change remove to reduce. Even the tone-vocoded channels convey some useful temporal fine structure information.

Done

Line 645. Change uninformative to less informative.

Done

Line 710. Change to “to be correlated with”.

Done

Line 714. Change report to reported.

Done

Line 760. Change on to for.

Done

Line 833. I was not sure what was meant by “expressive”.

We replaced this with “alternative”

Line 986. Change is to was.

Done

Line 1023. Perhaps give a brief explanation of “dropout regularization” and “softmax classification”.

We added a clarification:

“Dropout regularization helps prevent overfitting by randomly silencing network units during training, preventing learned solutions from being overly dependent on any individual unit. Softmax classification re-scales network output representations so they can be interpreted as probability distributions over output classes (the output representation for each stimulus is a non-negative vector that sums to one).” (lines 1394-1397)

Line 1125. Change was to were.

Done

Line 1190. Delete the.

Done

Line 1194. Insert was before played.

Done

In this paragraph and elsewhere, use “loudspeaker” rather than “speaker”.

Done

Line 1123. Change report to were.

Done

Line 1251. Change to “A perceptual weight of 0”.

Done

Line 1309. Insert for after as.

Done

Lines 320-328. Perhaps change ears to pinnae.

Done

Line 1333. It should be noted that the use of different real HRTFs involves less drastic changes than produced by inserting molds into the ears.

We added a note to this effect: "We note the use of different real HRTFs plausibly involves less drastic changes than produced by inserting molds into the ears" (lines 1719-1720)

Line 1381. Insert as at the start of the line.

Done

Lines 1390-1391. Is there a reason why models could not be trained to report the presence of two source locations?

We are working on this. It is a harder problem than single-source localization, and has not been studied nearly as thoroughly in humans, so it is nontrivial both to build models and to validate them.

Line 1396. Change dependency to dependence.

Done

Line 1400. Change the first to to and.

Done

Line 1493. Change in to using and change using to with.

We rewrote this sentence.

Line 1465. Insert for after as.

Done

Line 1467. Delete held.

Done

Line 1468. Insert give after to.

Done

Line 1472. Delete s at end of types.

Done

Line 1476. Insert recognition after noise.

Done

Line 1485. Delete held.

Done

Line 1491. Change in to for.

We rewrote this sentence.

Lines 1506-1546. Change pitch to F0.

Done

Line 1540. Change collect to collected.

Done

Line 1569. Insert "for speech" after accuracy.

Done

Line 1571. Change describe to described.

Done

Line 1582. Insert “that for” after to.

Done

Line 1604. Insert was before played.

Done

Line 1615. Change was to as.

Done

I would like my name to be revealed. Brian C.J. Moore

Thank you.

Reviewer #3 (Remarks to the Author):

This is potentially a very interesting and informative manuscript that takes a novel approach to assessing the importance, or otherwise, of phase-locking—action potentials whose timing is dependent on the phase of the sound stimulus cycle, here within the condensation and rarefaction period of a single cycle of a pure tone, the temporal fine structure (TFS).

It is indisputably the case that for sound source location based on the relative timing of the sound arriving at each ear, phase-locking of action potentials to the TFS is critical. Importantly, although the term TFS SHOULD exclusively be employed in this way and the way the authors do so, the term has recently been co-opted into psychoacoustic reports to accommodate spectral/frequency resolution. This is not the same thing, and the distinction made here is critical. Loss of spectral resolution in the cochlea, including from sub-clinical hearing loss (so-called ‘hidden hearing loss’, possibly the consequence of synaptopathy in one or more populations of auditory nerve fibres), for example, suggests that loss of TFS might be the factor determining poor speech understanding in noise, but no direct evidence of this exists, whilst recent evidence suggests even this form of hearing loss likely generates broader cochlear filters.

Thank you. We agree, as we hope is clear from our paper.

What is less clear is whether TFS is also required for so-called monaural listening tasks (i.e. when no spatial differences are explored). The current manuscript holds that this is the case, and that it relates to distinguishing different talkers from each other—based on voice pitch—but not for word discrimination. Of these two, voice pitch is evolutionarily conserved, and serves a similar purpose across species—determining source size (proxy for age, sex etc.), whilst word recognition is largely reserved for humans in whom spoken language evolved.

I have several points for the authors to consider and to which they should respond.

1. I appreciate, and consider this a strength of the manuscript, that models of performance based on real-world listening likely inform us as to the mechanisms that evolved to achieve tasks, and their bounds. This distinguishes the model from a psychoacoustic-type assessment, which is often stimulus determined. Distinguishing this model more from current/previous neural modelling studies would be important, including a more explicit statement on what makes this different. For example, it is not usual to consider binaural integration of phase-locked inputs without fully realized intermediate stages of processing that might reduce (or enhance, for frequencies below around 250 Hz) phase-locking. This could be tested. I would also like to hear from the authors as to why they think neural responses in the cochlear nucleus, at least in dedicated neurons, show hyper-phase-locking relative to the auditory nerve for these low-frequencies when they make the point about the ‘lossy’ nature of synaptic transmission in intermediate stages of processing relative to simply converging inputs from both ears onto binaural coincidence detectors.

In the revised manuscript we have clarified the distinction between our modeling approach and that of conventional models of binaural processing that typically attempt to incorporate detailed observations of binaural physiology. Our approach was instead to assess whether a minimal constraint inspired by biology would suffice to produce better matches to behavior. We have clarified this distinction in the revised results section, also noting the sharpening that occurs in the brainstem for low frequencies:

“Reasoning that synapses introduce temporal jitter that effectively imposes low-pass filtering⁷³, interaural integration was only allowed to occur in the models after early temporal pooling layers (see Methods). We note that this is a relatively weak biological constraint in the context of the detailed models of binaural processing stages^{71,72,74,75} that are used elsewhere in our field. We note also that there is evidence for enhancement of the precision of low-frequency phase locking in the brainstem⁴⁶, in addition to the loss of higher frequency phase locking that we modeled here. It nonetheless seemed useful to assess whether a minimalistic biologically inspired constraint would be sufficient to replicate human behavior.” (lines 535-542)

We have also noted that cochlear nucleus cells show enhanced phase locking to low frequencies, and discuss how this could potentially be understood in normative terms using models like ours that have somewhat more biologically plausible components (our suspicion is that the sharpening can be viewed as an algorithmic strategy to aid the extraction of timing information):

“The absence of realistic neural components in the models presented here limits the relevance of such analyses to hypotheses for actual neural circuits for extracting temporal information. However, future models with more biological constraints have exciting potential to make progress on these questions. For instance, one finding that at present lacks a normative explanation is the “sharpening” of phase locking that occurs in some neurons in the cochlear nucleus despite having a lower overall upper limit of phase locking⁴⁶. Task-optimized models could help evaluate the hypothesis that this sharpening aids the extraction of information, for instance by revealing whether sharpened timing emerges in intermediate stages prior to interaural comparisons.” (lines 1174-1186)

We know of no models of the cochlear nucleus at present that could be plugged into our model in the way that we have integrated models of the auditory nerve. But we have noted that this is an exciting direction for the future (and one we hope to pursue):

“Machine learning could also be combined with specific mechanistic proposals for how brainstem circuitry may extract task-relevant cues^{11,103}. Such proposals could be built into a machine learning model as an additional stage that is either fully fixed, or that has a small number of tunable parameters. Asking if the resulting model better accounts for behavior could help test mechanistic hypotheses.” (lines 1186-1192)

2. Although I appreciate the need to at least acknowledge the role of pinna cues in spatial listening—specifically their role in determining elevation of sources, the use of modified pinna cues (coupled with the ability of human listeners to learn ‘new ears’—as reported by Hofman et al 1998; cited by the authors) I am not convinced of the need to do more than acknowledge this in the current manuscript. Currently, this section in the Results reads as a ‘strawman’ (and see Figure 5). The collapse of elevation sensitivity when pinna cues are modified leaves sensitivity to azimuth unchanged (Figure 5e top panel) other than for the very-low pass IHC temporal filter. However, I don’t see the point of Figure 5e bottom panel, not in terms of what it shows, but why the authors consider this relevant. Absent correct pinna cues I would expect only ITD based horizontal localisation, and therefore the localisation deficit in this bottom panel is ‘simply’ the result of instantiating the low-pass IHC filter that was used to assess source localisation the previous sections. It certainly doesn’t interact with the smoothing of the spectral shape either, so I don’t see what else this is adding. If anything was to be removed or downplayed in this fairly length manuscript, it would be this. It might provide for completeness, but it is not surprising in the least empirically or theoretically. Removing it to supplemental would save some space in the manuscript.

Thanks for the feedback - we agree. We moved this section and figure to the supplemental materials, and now only briefly describe it in the main text:

“... we did find that models without access to phase locking became abnormally dependent on spectral cues for azimuthal localization (Supplementary Fig. 3), evidently to make up for the impaired binaural information that results from impaired phase locking. This latter result provides further evidence for the importance of phase locking to human spatial hearing.” (lines 580-584)

3. The perspective that voice pitch might limit the upper bound of monaural phase locking that can be extracted ignores distinguishing higher pitches from other sources. This seems a very human-centric perspective (I don’t necessarily disagree; voice pitch is determined by vocal tract length which provides for the same purpose of distinguishing size of an individual as it does in other species) but what about higher pitches? Is distinguishing voice pitches ‘special’ in which case this should be evident in the relative ability (in some way) to process pitch? It is also worth noting that phase-locking to upper harmonics in the envelope would be pass-band limited, with cochlea filters of around 4 kHz having bandwidths of approx. 350 Hz (i.e. just less than 1/10 of the centre frequency). Is it fortuitous or otherwise that this limit seems to hold? 4kHz seems to mark a transition in pitch processing, whether spectral or temporal.

We didn’t mean to argue that monaural processing of phase locking is entirely driven by what is needed for voice recognition. We can see how this confusion might have resulted from the original submission, where we suggested that the phase locking dependence evident for word recognition must have been driven by the importance of phase locking in other tasks, and gave voice recognition as an example. However, during the revision process we decided that this argument required more nuance given there is a significant effect of the phase locking limit on word recognition performance, that is moderately

substantial for some types of noise (we now point this out explicitly in the text). There is nonetheless also evidence that voice recognition increases the dependence of word recognition on phase locking, in that the tone vocoding effects are stronger in the models trained on both tasks. So we have added some discussion of this along with a new supplementary figure that makes it easy to compare the tone vocoding effect in the two training conditions (word recognition alone vs. word + voice recognition).

Regardless, we view voice recognition as just one candidate among potentially many tasks that might play a role in this way. It just happens to be one that we were able to study in the context of this project. It seems equally plausible to us that music-related tasks, or environmental sound recognition, might also affect the incorporation of phase locking into word recognition. We have clarified this issue in the revised text:

In the Results section:

“These results suggest phase-locked spike timing is needed to comprehensively account for human word recognition behavior. The phase-locking-dependent effects of tone vocoding were present even in models that were only optimized for word recognition (Supplementary Fig. 4g). This suggests that the modest benefit of phase locking on word recognition task performance (Fig. 2a, 5, and Supplementary Fig. 6) is enough to produce a strategy that incorporates phase locking to some extent. However, the magnitude of the tone vocoding effect was somewhat larger in models that were jointly optimized for word and voice recognition (5.8 dB compared to 4.1 dB for models optimized only for word recognition; Supplementary Fig. 8). This raises the possibility that the dependence of human-like word recognition on phase locking is partly a consequence of sharing machinery with tasks that benefit more from phase locking (voice recognition being one candidate).” (lines 860-869)

The new Supplementary Figure:

We also note that our results are by and large consistent with the idea that each domain uses what it needs. We see no reason that there should be a single upper limit of monaural processing of phase locking. For instance, in pitch processing, analogous models provide evidence that phase locking above 1 kHz is used (Saddler et al. 2021). As we develop models of music perception tasks in the future, we expect to be able to test this more definitively. We now more explicitly point out the task dependence (also in response to Reviewer 4), at the start of the Discussion:

“This finding suggests that different domains likely use phase locking to different extents depending on its utility for natural behavior.” (lines 992-993)

We also note that the results leave open the possibility that other species might use phase locking up to higher limits, for the representation of vocalizations or other sounds. We have added a note to this effect in the revised discussion:

“The higher fundamental frequencies of animal vocalizations compared to human speech also raises the possibility that non-human animal analogues of voice recognition could utilize phase locking up to higher frequencies than we found implicated for human voice recognition, which could in principle also be investigated with our modeling framework.” (lines 1217-1221)

4. Cochlear implants are the true human perspective for life without TFS phase-locking. Implant users are usually now well trained on, and highly capable of parsing, speech in quiet, but they struggle in noise and in terms of sound localisation. In the latter, pitting interaural level or timing cues against each other they consistently adopt the level cue, ignoring the usually more potent timing cue. They also struggle to resolve pitch per se., including voice pitch. The manuscript could usefully discuss the data with respect to this unique population of human listeners. This is especially so, given the section on tone vocoding (implant users' peripheral auditory nerve fibres are acting likely similar to speech/noise vocoders).

We agree this is an important issue. We have added the following paragraph to the Discussion section:

“A similar approach could be applied to cochlear implants, by substituting simulations of electrically stimulated nerve fibers¹⁰⁹ for the nerve model used here. Most current cochlear implant processing strategies discard phase locking to the fine structure, but also induce a number of other differences in auditory nerve responses compared to those produced by a normal ear¹¹⁰⁻¹¹³. It is thus not clear how much of the difficulties experienced by cochlear implant listeners (e.g. impaired sound localization, pitch perception, and speech intelligibility in noise) are primarily due to the loss of fine structure rather than to other factors. Models optimized with different types of cochlear implant processing strategies could provide insight into these issues, and into the potential for alternative strategies.” (lines 1203-1210)

Reviewer #4 (Remarks to the Author):

This manuscript describes an extremely thorough and rigorous comparison of modeling analyses and human performance (some collected here, but mostly from literature) that address a fundamental question in auditory neuroscience that has been debated for many decades – whether fine temporal coding is relied upon by human listeners for perception. This question is highly significant in the field of auditory neuroscience as it has major theoretical and translation implications, e.g., for the design/re-design of auditory prostheses such as cochlear implants.

This work is highly significant in that it uses modern machine learning techniques to allow this question to be evaluated for real-world listening tasks that are highly relevant to the daily lives of human listeners (e.g., complex sound localization, word and voice recognition) and as such are likely to be what the human brain trained on in learning to use the most useful auditory cues that enter the auditory system. As such, this work advances the field in important and significant ways by showing specific tasks (e.g., sound localization, voice recognition, but not word recognition) for which fine temporal cues are needed to explain human perception.

The work (both modeling and experimental) appears to be rigorously carried out in almost all regards (see one concern below) and is extremely thorough in its breadth of experimental conditions and stimuli considered. As such, the conclusions are well justified. Limitations in the approach are well considered as well in the discussion section, adding to the rigor of this work. However, several issues exist that can be addressed to improve the presentation of this work.

Furthermore, although the thoroughness is admirable, the downside is that the manuscript is dense and long with many details that only auditory perceptual scientists will fully appreciate. As such I worry about the accessibility of the big-picture take-home points to the broad audience that will surely be interested in this modern approach to a classic question. I don't have exact suggestions to cure this as the thoroughness is what makes this work so rigorous, but I encourage the authors to do as much as they can (beyond what they have already done in many ways with nice overview paragraphs) to make this long manuscript accessible to a broad audience. It is not that the details are not able to be followed, in fact they are well written, it is just that I worry that the general reader will struggle to separate the main broad findings from the necessary details that may be out of their expertise. It would be a shame to lose the forest for the trees.

Thank you. We have moved one section and figure to the supplemental materials to streamline the revised manuscript. We also removed some methods details from the results section that seemed non-essential for the general reader, and have trimmed a few other bits of the main text that seemed non-essential. The reviews inevitably required some additions to the text, but we have kept these as concise as possible.

General Comments

- While this paper (in full detail) does a good job of highlighting that certain tasks may require fine timing, but others do not, the title suffers a bit from suggesting that fine timing is necessary

in all conditions (as if there is one answer to this question for all tasks). I recognize the title is limited in space and not able to fit in qualifications, but there is other recent work in the field suggesting the opposite conclusion (e.g., Whiteford and Oxenham – not currently cited) and so it is important to make clear (as soon as there is space to do so) that there is not necessarily one answer to this question for all tasks and as such different studies can get different answers but not be inconsistent with each other. In this regard, some discussion of the apparently opposing strong results from this study and strong results from the Whiteford and Oxenham work would help the reader to reconcile opposing answers they will find in the literature (and in fact in the same study, such as this one, where the answer depends on the task).

We have revised the title and the abstract to make it clearer that results varied across tasks. The revised title and abstract:

Models optimized for real-world tasks reveal the task-dependent necessity of precise temporal coding in hearing

Neurons encode information in the timing of their spikes in addition to their firing rates. Spike timing is particularly precise in the auditory nerve, where action potentials phase lock to sound with sub-millisecond precision, but its behavioral relevance remains uncertain. We optimized machine learning models to perform real-world hearing tasks with simulated cochlear input, assessing the precision of auditory nerve spike timing needed to reproduce human behavior. Models with high-fidelity phase locking exhibited more human-like sound localization and speech perception than models without, consistent with an essential role in human hearing. However, the temporal precision needed to reproduce human-like behavior varied across tasks, as did the precision that benefited real-world task performance. These effects suggest that perceptual domains incorporate phase locking to different extents depending on the demands of real-world hearing. The results illustrate how optimizing models for realistic tasks can clarify the role of candidate neural codes in perception.

We have added a reference to the Whiteford paper, noting that our results make it plausible that different tasks could yield different results:

“Our results also indicate that tasks could vary in the extent to which they require fine timing, such that the conclusions derived for one task may not generalize to others⁹⁹.”
(lines 1096-1097)

- Much nice discussion is provided about the differences between optimal observer approaches, where the decision process can be proven mathematically to be optimal but are limited to simple tasks, and ML approaches that are able to be applied to real-world complex tasks (by optimizing parameters based on empirical data). But this leads to confusing wording (often in same sentence or adjacent sentences), because different definitions of optimal & optimization are being used, when at face value they would be assume to be related. E.g., lines 47-51: “Although not provably optimal, ... human-like behavior can emerge in deep artificial neural networks

optimized for such tasks.” Please work to clarify this important but subtle difference where optimizing a system does not mean making it optimal.

We have clarified this distinction in the Introduction:

“In contrast to analytically derived optimal solutions, the solutions found via an optimization process are not guaranteed to be optimal (for instance, the optimization procedure could get stuck in local optima, and/or the model class being optimized could be suboptimal for the problem). However, optimization drives a model towards better performance, such that the resulting model may nonetheless reveal the characteristics of a system optimized for a problem under particular constraints. In this way, machine learning offers an alternative to the traditional ideal observer approach for real-world perception problems that can only be specified empirically.” (lines 63-69)

- This is brought up a bit in the Discussion section limitations (lines 798-805; 831 onward), but defining optimal and optimizing earlier would help as it comes up in the intro. In fact, the discussion on lines 798-800 addresses this a bit but not entirely correctly – The classical ideal observer theory is not limited to simple "stimuli", only simple "tasks" – e.g., a simple one-parameter discrimination task is tractable for optimality with arbitrarily complex stimuli (e.g., Heinz et al 2001) – so I suggest removing “stimuli” from line 799.

Done.

To address the question of whether ML approaches are in fact close to optimal (lines 831 onward), it would be nice to add in a condition where both the ML and optimal detector approaches are viable (e.g., Maximum-likelihood estimators and CR bounds can be derived) and see if they agree (e.g., pure-tone frequency discrimination limens (FDL) that Siebert and Heinz et al analyzed (analytically and computational, respectively). This would provide a nice result to cite that the same ML approaches used for the complex real-world tasks are in fact close to optimal (e.g., able to use fine timing to perform several orders of magnitude better than humans) when we know mathematically what optimal performance actually is.

Thank you for the suggestion. We ran the suggested experiment, and it worked out as expected, making for a nice addition to the paper.

To test whether machine learning models can attain near-optimal performance on a task for which optimal performance has previously been derived, we trained deep neural networks to make pure tone frequency discrimination judgments. Neural networks operated on pairs of simulated auditory nerve representations of 200 ms pure tones and were trained to report which tone had the higher frequency. To optimize the neural network architecture, we trained 120 different convolutional neural network architectures on this task and selected the 10 top-performing networks to use as our frequency discrimination model. We separately trained the model with different phase locking limits by setting the IHC filter cutoff to either 3000, 1000, 320, or 50 Hz. We then measured the

model pure tone discrimination thresholds and compared them to the ideal observers from Siebert (1970) and Heinz et al. (2001). As can be seen in the figure below, our neural network model with the lowest phase locking limit (50 Hz) closely approximates the rate-place ideal observer, whereas the model with the highest phase locking limit (3000 Hz) closely approximates the all-information ideal observer. This is exactly what would be expected if the neural network models were approaching the actual ideal observers in each case. These results bolster the argument that the machine learning approach can achieve results that are close to optimal. We have added this result to the paper.

Here is the figure that has been added to the Results section:

And the new section of the Results text that describes the experiment:

“The approach taken in this paper is predicated on the idea that an optimized machine learning model can approach the characteristics of an ideal observer. To test the plausibility of this assumption, we trained neural network models on a task for which provably optimal observers can be derived: frequency discrimination. We trained 120 different convolutional neural network architectures on the task and selected the 10 top-performing architectures. Simulated auditory nerve representations of the two stimuli (200 ms pure tones of different frequencies) were supplied to the models as different input channels (Fig. 8a), with models separately optimized for the four phase locking cutoffs used elsewhere in this paper. We measured discrimination thresholds from psychometric functions generated from the model judgments (Fig. 8b), using the same stimulus conditions with which previously published ideal observers for this task were evaluated. As shown in Fig. 8c, the optimized neural network model with the lowest phase locking cutoff (50 Hz) closely approximated the “rate-place” ideal observer that

operates exclusively on firing rates. By contrast, the model with the highest phase locking cutoff (3000 Hz) closely approximates the ideal observer that uses spike timing in addition to firing rates. The two intermediate phase locking cutoffs produce results intermediate between the two ideal observers. This result shows that machine learning models of the sort used in this paper can achieve results that are close to optimal for simple tasks, bolstering the idea that the results shown here for more complex tasks may also be indicative of characteristics of ideal observers.” (lines 953-969)

- A related comment – I think it should be made more clear that the ML models are not in fact fit to the human data at all, but rather are just training to do the task as well as possible. I think this is made more clear in previous publications from this group, but it is a critical point for the arguments being made. In fact, if the FDL experiment could be added and turned out as I expect it would (i.e., that the ML models greatly outperformed humans), this (and the ITD with 3000-Hz cutoff condition) can be cited to illustrate this point and allow us to think of these as likely near optimal models.

We have emphasized this point at the beginning of the Results section:

“We emphasize that the models were not fit to match human data, and were optimized only for task performance. Any similarity to human behavior is thus a consequence of optimization for the task given the constraints of the simulated auditory nerve input and model architecture.” (lines 235-238)

- One methodological question that arose that also makes it important to compare this approach to Siebert’s approach is the question of stochasticity in neural spike trains. It is well known that auditory nerve fiber spike times are well described as a Poisson process with refractory effects. My understanding from the methods (e.g. line 978) is that each time bin is treated independently and as binomial distribution. The independence contradicts refractory effects, and it is not clear how similar the neural statistics here are to the known Poisson statistics of AN fibers. Some discussion and/or simulations will help to be sure this is not a significant deviation from known AN statistics (I doubt it is, but needs to be checked). The simulations suggested above to see if the ML approach matches Siebert’s predictions will also help to address this. The implications of getting AN-fiber statistics correct for these real-world predictions should be discussed (e.g., refractory effects on the usefulness of high-frequency fine-time information). The Poisson statistics have been shown by Siebert and others to be important for intensity discrimination tasks, but it would be interesting to discuss whether they are critical for real-world tasks.

We modeled spike trains from individual nerve fibers as a time series of independent Bernoulli random variables (i.e., a Poisson process in the limit of infinitely small time bins). We described the spike sampling as “binomial” because we summed the spike trains from all nerve fibers with the same CF and spontaneous rate, which is equivalent to sampling a binomial variable at each timestep. We neglected to model refractoriness based on the assumption that its effects would be negligible in this setting, in which the

volley principle should cause the response summed across fibers to be robust to refractoriness in individual fibers.

In response to your comment we tested this assumption. To examine the effect of incorporating 1ms absolute refractory periods, we measured the statistics of the spike counts in our simulated nerve response when each individual nerve fiber's firing rate was set to zero for 1ms after each spike was sampled. Consistent with our assumption, this manipulation had very little effect on the summed spiking trains used as input to the neural network stages of our models. Summed spiking representations with refractoriness had ~1% fewer spikes and were highly correlated with equivalent representations without refractoriness ($r > 0.99$). From unpublished work in our group modeling auditory nerve fiber loss, we are confident such small differences in the input representations would not produce discernable differences in model behavior.

The figure below shows example nerve responses with and without simulated refractoriness (note that the blue curves are slightly below the red curves, but otherwise very similar):

Effect of neglecting refractoriness on summed auditory nerve spike counts. We simulated the spiking responses of 32000 auditory nerve fibers to a 2 s speech utterance, once neglecting refractory effects (red lines) and once incorporating a 1 ms absolute refractory period (blue lines). Spike counts from 19200 high spontaneous-rate fibers (a.), 8000 medium spontaneous-rate fibers (b.), and 4800 low spontaneous-rate fibers (c.) as a function of time, summed across 50 characteristic frequencies spanning 125 to 8000 Hz. Panels d, e, and f zoom in on the middle 50 ms of a, b, and c, respectively. The red and blue lines are partially transparent such that overlapping regions appear purple.

We also note that as far as we can tell from going through the papers, both Siebert and Heinz similarly ignored refractoriness. However, when we trained DNNs to make frequency discrimination judgments using the auditory nerve representation from Heinz, we obtained thresholds similar to those reported by Siebert and Heinz.

We have added a note to the methods section to address this issue and describe this analysis:

“We did not attempt to model refractoriness in nerve fiber responses on the grounds that summing across fibers should minimize effects of refractoriness. To test this assumption, we generated examples of an alternative set of nerve responses in which each individual nerve fiber’s firing rate was set to zero for 1 ms after each spike was sampled. This resulted in a small reduction in the overall number of spikes, but otherwise produced very similar responses (the summed spike trains used as inputs to the neural networks were highly correlated to those obtained without modeling refractoriness; $r > 0.99$).” (lines 1301-1313)

- Similarity metrics – More guidance is needed on why two similarity metrics are used and how to interpret them, and this should be tied into the decades long interpretation issue with respect to the questions being addressed here: is it the absolute similarity between model and human performance that matters (has been used to suggest rate code for FDL predictions), or the pattern that matters (has been used to suggest timing code in FDL predictions). It seems like the two metrics being used (RMS and correlation) are trying to get at this issue, but this is not really discussed well enough in the general framework to make that issue clear to the reader. Perhaps it is because the two are more similar in these real-world tasks (but not always!), but the current predictions and use of these similarity metrics should be tied into this discussion with respect to this debate in the past literature addressing these questions.

Our motivation for using two metrics was just to provide a stronger test of the models. Our perspective on this is that the right model should look good under both a mean-squared-error metric and a correlation metric. So if one model is substantially worse than other models under either type of metric, it is ruled out as an account of perception. We had not made the link to the interpretational issues associated with the FDL ideal observers, but we can see how some readers would.

We have clarified these issues in several places. In the Methods:

“The two human-model similarity metrics measure different things. The correlation metric assesses the similarity in relative performance across conditions, whereas the RMS error can reflect absolute differences in performance between a model and humans. A “good” model should exhibit high similarity on both metrics. A model only needs to exhibit substantially lower similarity on one metric to be ruled out. This was the scenario we found for word recognition, where models with the different phase locking limits were

distinguished more clearly by the correlation metric (Fig. 2) than by the RMS metric (Supplementary Fig. 1).” (lines 2097-2103)

And in the caption for Supplementary Fig. 1, which shows the aggregate RMSE results:

“All models reproduced human word recognition fairly well according to this alternative metric, but the 50 Hz model was still worst overall, and the change in human-model similarity, while modest, was largest between the 50 Hz and 320 Hz models than between the other phase locking limits. We note that a model only needs to appear worse than others according to one metric to be ruled out, and the correlation metric was more diagnostic in this case. This is because the 50 Hz model exhibits a qualitative discrepancy for one experiment (Fig. 7a-c), and this is revealed most clearly with a correlation metric.” (lines 2529-2533)

We have also added some discussion of the issues that were present in earlier modeling work:

“We note that these two types of metrics have in some cases yielded inconsistent conclusions regarding previous ideal observer models¹². Specifically, ideal observers of frequency discrimination that use information from phase locking exhibit much better absolute performance than humans, but replicate the qualitative dependence of thresholds on frequency. By contrast, ideal observers that do not have access to phase locking exhibit absolute thresholds closer to those of humans, but do not replicate the human dependence on frequency. Here we instead found the two types of metrics to yield comparable conclusions, in that models with the lowest phase locking limits never exhibited higher human-model similarity irrespective of which metric was used. Moreover, the models with higher phase locking limits generally matched both absolute and relative performance and thus scored relatively well with both metrics. One difference compared to previous work is that our models were optimized for real-world tasks, and evaluated in real-world conditions as well as more traditional laboratory psychoacoustic assessments. We have found (here and elsewhere^{19,20}) that such models tend to produce both absolute and relative performance on par with humans. This general finding is consistent with the idea that absolute performance reflects the demands of optimization for ecologically important tasks, such that optimizing a model for such tasks produces absolute performance that is close to that of humans.” (lines 2105-2119)

We also note that while going over these results during the revision process we found a suboptimality in the way the RMS similarity metric was previously being calculated, which we have corrected in the revised manuscript. Previously we were normalizing model results separately for each experiment. This was intended to deal with the fact that different experiments have different measurement units, but had the consequence of equating the variance in experiment results across experiments, which is inappropriate for this type of metric. We instead switched to min-max normalizing model results to

re-scale human data across conditions between 0 and 1, but now apply the same normalization to all experiments with the same measurement units. The results are a bit different as a consequence, but the conclusions are not affected. The main difference is that all the word recognition models have fairly low RMSE (though the 50 Hz model remains the worst). This is because the main discrepancy between the 50 Hz model and humans is the qualitative form of the result for tone vocoding, and this shows up most clearly with a correlation metric.

The updated metric is described in the Methods section:

“Data were first min-max normalized within experiments (rescaling human data to range from 0 to 1 across conditions) to account for different units and scales across experiments. For the three word recognition experiments that measured proportion correct in different conditions (type of background noise, SNR, or F0 manipulation), the same min/max human scores (computed across all conditions) were used to normalize data. This prevented experiments that produced null effects (i.e., the lack of an effect of F0 manipulation on human word recognition) from artificially inflating the mean RMS error.” (lines 2090-2095)

Here is the updated version of Supplementary Fig. 1:

With Figure 2 for comparison:

- Overall it could be made more clear where the human data being compared to comes from (i.e., measured as part of this study, other studies from the authors' group, or from others in the literature). A lot of this is in fact in the manuscript in detailed descriptions, but given length of the manuscript this gets lost at times and overall. I think it would benefit the presentation of this work to be sure it is explicit in each section/figure caption, as well as provide some sense in abstract or more likely (given space constraints) in intro a sense that most of the data is actually from established data from the literature, not collected here (but that some human data were collected by the authors).

We have tried to clarify in each figure panel and caption where human data is from the literature.

Specific Comments

- Line 16. "is uncertain" does not do justice to the breadth of work already done to explore this question – suggest "remains uncertain".

Done.

- Line 22 "ecological task demands" and "neural implementation constraints" are quite vague here in the abstract and only become clear (mostly) well into the manuscript. Consider being more specific in describing the actual results (e.g., that binaural sluggishness was needed to explain human behavior in localization). May be hard given space constraints in abstract, but will help because as it is now it is hard to understand what is meant from just the abstract.

Thank you for the feedback. We have rewritten the abstract to be more explicit about the results and conclusions.

- Lines 23-24. This is an interesting point, but is not clear as written what exactly is meant ("in principle" is pretty vague). Prosthetics are briefly discussed in the discussion, but not in this light

– i.e., not about trying to overcome the failure to provide fine temporal coding and whether that would restore near-normal hearing. Is there evidence that prostheses that fail to provide high-fidelity temporal cues lead to performance in line with the 50- or 320-Hz model predictions here? None of this is discussed anywhere in the manuscript, so this final sentence seems to be an overstep as written.

We have removed this from the abstract, and instead have added a paragraph to the discussion discussing applications to cochlear implants.

- The ITD condition where central limitations were needed to account for human data that degrades above ~1500 Hz, but the monaural ML model degrades above 3000 Hz is a nice example suggesting central limits may be needed, but begs the question whether a model without the central limits but with 1500-Hz IHC cutoff could explain these data just as well. If so, what else in the broad set of predictions here would fail with 1500-Hz cutoff? Since we don't actually know human cutoff, what if human PL cutoff is actually 1500 Hz, not 3000 Hz, as others have proposed (e.g., in some sections of Verschooten et al 2019). This possibility should be discussed.

We have no way to exclude this possibility apart from finding other tasks that show evidence of incorporating phase locking at higher cutoffs. We have in fact seen evidence from pitch models that pitch behavior is more human-like with phase locking above 1 kHz (Saddler et al., 2021), but that is at present the only relevant piece of evidence that we know of. We have added a note to the results section that the results are equally consistent with there being a phase locking cutoff in human that is lower than 3000 Hz:

“We note that the results are equally consistent with the possibility that the cutoff of phase locking in humans is substantially lower than 3000 Hz (i.e., lower than in other mammals, as some have argued³³), and would also provide a normative justification for such a lower cutoff from the standpoint of sound localization.” (lines 567-570)

- Line 948. “S” fiber types; line 978 “F” fibers – seems like S is used throughout, so not sure why F used here.

Corrected.

- Fig. 3f – clarify noise is signal here, not masker.

Done.

- Fig. 3d – mention inverted axis, such that down is more error, up is better.

Done

- Line 408 Section Title – would be nice to be more specific – “delayed binaural integration”

We understand the sentiment. In this case we think the suggested change will make the general idea less clear to a general audience, and so have opted to stick with the original heading.

- Fig.5. stimuli should be stated in caption (unless I missed it)

Done.

- lines 480-481 Section heading. It is hard to see this result, since data supporting it are indirect. Title may not be place to clarify exact evidence for this conclusion, but fig caption perhaps or explicitly stating argument explicitly in text to get to this conclusion.

We have moved this section to a supplementary figure, and removed the section heading as a result.

- Fig. 8e – mention somewhere why no human data are shown here, in contrast to almost all other panels in the paper.

We have added an explanation to the figure caption:

“Although the qualitative effects shown here have been documented in humans, the experiment we used to measure the effects in our model had not been conducted in human listeners, and so we do not have an explicit comparison to human data.” (lines 846-847)

- Line 785. Some explanation of why “limited benefit” above 1000 Hz would help (even if not known for sure). E.g., perhaps at higher frequencies head size limits usefulness of phase comparisons (that become ambiguous) when wavelengths get short re: head size.

We have added this speculative explanation:

“(perhaps because time differences become ambiguous when wavelengths are short relative to head size)” (lines 1021-1022)

- Line 849 – what does “spike sampling” mean? Is this just another way to say stochastic nature of spike trains, which is how the randomness has been discussed in the past.

This was just a reference to what actually happens in the model, where spikes are sampled from a time-varying firing rate. We have added a reference to the Methods to refer the reader to the underlying details.

- Line 921. Why “apparent” stochasticity? This has been shown to exist many times and is well characterized.

We were alluding to the possibility that some of the trial-to-trial variability documented in the central nervous system could be a function of internal state, and thus might not be

stochastic from the perspective of the observer. We have removed this paragraph to help streamline the manuscript.

- Supp Fig 8 (and 9) caption – mention Fig. 2 (and 3) as comparison with full AN model.

Done.

Responses to Round 2 of Reviews – Saddler and McDermott

Reviewer #1 (Remarks to the Author):

The authors have done an excellent job at responding to my comments on the original manuscript.

Thank you.

Reviewer #2 (Remarks to the Author):

The authors have responded comprehensively and appropriately to the comments of the reviewers. I have only minor suggestions for further changes. It should not be necessary for me to review the paper again.

Thank you.

Line 103. Change in to using.

Done

Line 121. I am not sure if this is the journal style, but “Data are...” would be more correct.

Done

Lines 125-126 partly repeat lines 123-124.

Fixed

Lines 269-270 and elsewhere (e.g. lines 278-279, 458, 829). This implies that phase locking above 1000 Hz is not used at all. But an IHC cutoff frequency of 1000 Hz only means that the precision of phase locking in the model decreases above 1000 Hz. Usable phase locking may occur at higher frequencies, consistent with evidence that TFS plays a role in sound localization for frequencies up to 1500 Hz.

We have added a note to highlight this issue:

“We note that the cutoff determines the frequency at which phase locking precision rolls off, not the upper limit of all detectable phase locking, which could be slightly higher. For simplicity, we refer to different models by their cutoff frequencies (e.g., the 3000 Hz phase locking model).”

Lines 568, 572, 582, 592, 593, 594, 596, Fig. 6b, 601, 705, 787, 842, 1262, 1611, and elsewhere. Use the term “pitch” to refer to the subjective attribute and use F0 (consistently) to refer to the physical fundamental frequency.

Corrected

Figure 7c. The human data show a marked increase as the number of channels with TFS increases from 0 to 8, while none of the models shows a marked increase. This merits a small comment.

We added a comment to note this discrepancy:

“We note that even the 3000 Hz phase locking model showed a smaller benefit than humans for 8 vs. 0 channels, possibly because the filter bank used to vocode stimuli in the model experiment differed slightly from that used in the human experiment (see Methods).”

Line 774. Insert “performance for” before “impaired”.

Done

Line 928. Change “fundamental frequency” to F0.

Done

Lines 979-980. Change “fundamental frequencies” to F0s.

Done

Lines 1082-1083. This reads a bit strangely.

We revised this to make it clearer:

“We modified the upper frequency limit of phase locking in the auditory nerve model by adjusting the cutoff frequency of the IHC low-pass filter. In the unmodified auditory nerve model, the low-pass characteristics of the IHC membrane potential were modeled as a 7th order filter with a cutoff frequency of 3000 Hz.”

Line 1798. Change idea to ideal.

Done

I would like my name to be revealed. Brian C.J. Moore

Reviewer #2 (Remarks on code availability):

The code appears to be valid and correct.

Thank you.

Reviewer #3 (Remarks to the Author):

I am very pleased with the edits the authors have made for this manuscript, especially concerning the perspective on spatial hearing and the removal of the section concerning head-related transfer functions to supplemental information.

Thank you.

Reviewer #4 (Remarks to the Author):

Thanks for your thorough responses to my comments and those of the other reviewers. All of my concerns were addressed well in the revisions. This rigorous and thorough manuscript is now even stronger, and will make a very nice contribution to the literature.

Mike Heinz

Thank you.